# Birth mode is associated with earliest strain-conferred gut microbiome functions and immunostimulatory potential

Linda Wampach[1,9], Anna Heintz-Buschart [1,2,3], Joëlle V. Fritz[1,4], Javier Ramiro-Garcia[1], Janine Habier[1], Malte Herold[1], Shaman Narayanasamy[1,5], Anne Kaysen[1,4], Angela H. Hogan[6], Lutz Bindl [4], Jean Bottu[4], Rashi Halder [1], Conny Sjöqvist[7,8], Patrick May [1], Anders F. Andersson [7], Carine de Beaufort[4] & Paul Wilmes[1]

The rate of caesarean section delivery (CSD) is increasing worldwide. It remains unclear whether disruption of mother-to-neonate transmission of microbiota through CSD occurs and whether it affects human physiology. Here we perform metagenomic analysis of earliest gut microbial community structures and functions. We identify differences in encoded functions between microbiomes of vaginally delivered (VD) and CSD neonates. Several functional pathways are over-represented in VD neonates, including lipopolysaccharide (LPS) bio-synthesis. We link these enriched functions to individual-specific strains, which are trans-mitted from mothers to neonates in case of VD. The stimulation of primary human immune cells with LPS isolated from early stool samples of VD neonates results in higher levels of tumour necrosis factor (TNF-α) and interleukin 18 (IL-18). Accordingly, the observed levels of TNF-α and IL-18 in neonatal blood plasma are higher after VD. Taken together, our results support that CSD disrupts mother-to-neonate transmission of specific microbial strains, linked functional repertoires and immune-stimulatory potential during a critical window for neonatal immune system priming.

---

[1] Luxembourg Centre for Systems Biomedicine, University of Luxembourg, avenue des Hauts-Fourneaux 7, 4362 Esch-sur-Alzette, Luxembourg. [2] German Centre for Integrative Biodiversity Research (iDiv) Halle-Jena-Leipzig, Deutscher Platz 5e, 04103 Leipzig, Germany. [3] Helmholtz Centre for Environmental Research GmbH – UFZ, Theodor-Lieser-Str. 4, 06120 Halle (Saale), Germany. [4] Centre Hospitalier de Luxembourg, rue Nicolas Ernest Barblé 4, 1210 Luxembourg, Luxembourg. [5] Megeno S.A., avenue des Hauts-Fourneaux 9, 4362 Esch-sur-Alzette, Luxembourg. [6] Integrated BioBank of Luxembourg, rue Louis Rech 1, 3555 Dudelange, Luxembourg. [7] KTH Royal Institute of Technology, Science for Life Laboratory, School of Biotechnology, Division of Gene Technology, Tomtebodavägen 23a, 17165 Solna, Sweden. [8] Environmental and Marine Biology, Åbo Akademi University, Tykistökatu 6, 20520 Turku, Finland. [9] Present address: Laboratoire National de Santé, rue Louis Rech 1, 3555 Dudelange, Luxembourg. These authors contributed equally: Linda Wampach, Anna Heintz-Buschart, Joëlle V. Fritz. Correspondence and requests for materials should be addressed to P.W. (email: paul.wilmes@uni.lu)

The past decades have witnessed steadily increasing rates in caesarean section deliveries (CSD) performed largely in the absence of medical necessity and reaching proportions of 19.1% worldwide and 25% in Europe[1,2]. During vaginal birth, specific bacterial strains are transmitted from mothers to infants[3–6] and differences in microbial colonization in neonates born by CSD have been identified[7–10] as early as 3 days postpartum[7,10]. However, due to conflicting results, which principally imply a negligible impact of delivery mode on the colonizing neonatal microbiome in the gut[11], it remains unclear whether disruption of mother-to-infant transmission of microbiota through CSD occurs and whether it affects human physiology early on, with potentially persistent effects in later life. As the first few days after birth represent a 'critical window' in neonatal health and development[12–14], there is growing concern that disruption of microbial transmission from mother to neonate is linked to conditions more frequently observed in CSD-born individuals, including allergies[15], chronic immune disorders[16] and metabolic disorders[17]. To address these concerns, it is essential to determine if there are differences in the functional complement conferred by the earliest colonizing microbiota in relation to CSD, if any differences result from changes in the transmission of strains from mothers to neonates, and if these impact neonatal physiology.

While the majority of studies so far indicate that delivery mode is the strongest factor determining early neonatal gut microbiome colonization[3,7–10,18], these effects are either extenuated or largely absent in other studies[11,19]. In this context, it is important to consider that CSD may be performed as a result of underlying maternal or foetal medical conditions (e.g., multiple gestation, foetal malpresentation or suspected foetal macrosomia)[20] and can co-occur with other microbiome-influencing factors. More specifically, CSD is most often accompanied by the administration of antibiotics to mothers due to local health regulations or hospital practices (e.g., in case of a positive screening of the mother for group B *Streptococcus*)[21]. Being born small for gestational age (SGA) frequently coincides with CSD as well (i.e., more than 50% of all SGA neonates)[22]. SGA neonates have an elevated propensity for developing metabolic disorders during childhood or adulthood, which has been associated with alterations to the gut microbiome[23], and may be linked to the elevated rate of CSD in this population.

Apart from confounding factors, the methods and study designs employed over the past years may in part explain some of the conflicting results regarding the effect of delivery mode on the early gut microbiome. Notably, taxonomic profiling based on 16S rRNA gene amplicon sequencing does not offer sufficient resolution to assess the direct effect of the delivery mode at the level of strain transmission, which is expected to be a determinant of succession. Although recent studies have focused on mother-to-neonate strain transfer and have shown that maternal strains do colonize the neonatal gut, non-vaginal delivery was not assessed comprehensively[4–6]. In addition, although single nucleotide variants (SNVs) have been tracked over time, no such studies have so far covered the earliest time points after delivery (days 0–5) in well-matched mother–neonate pairs in relation to a direct comparison of delivery modes. Consequently, there is a strong need for adequate high-resolution metagenomic analyses capable of resolving the vertical transmission of individual-specific strains and encoded functions from mothers to neonates on an individual basis, while also supplementing observed in silico findings with further in vitro validation experiments.

Independent of whether prenatal colonization of the foetus takes place or not[24], delivery marks the moment of extensive exposure to microbial communities of faecal, vaginal, skin and environmental origins and this event thereby has a profound impact on the colonization of the neonatal gut[9,25]. The initial low microbial biomass in the earliest neonatal stool samples[7] makes the sequencing data prone to the over-representation of putative artefactual reads[24,26] and may contribute to the generation of inconsistent results. Therefore, the removal of any artefactual sequences is essential to ensure unambiguous, high-resolution overviews of the earliest microbial colonization of the neonatal gut.

Here, we performed a detailed analysis of the earliest microbial colonization of the neonatal gut using a combination of 16S rRNA gene amplicon sequencing and high-resolution metagenomics. Our results highlight differences in gut microbiome composition according to delivery mode as well as concurrent differences in the encoded functional potential, which in turn are linked to differences in the transfer of strains from mother to neonate. Based on the enrichment of the LPS biosynthesis pathway in VD neonates, we performed LPS extractions from neonatal stool samples, in vitro immune stimulation assays as well as extensive assessments of LPS purity. The stimulation of primary human immune cells with purified LPS from the faeces of VD neonates collected at day 3 postpartum resulted in the production of higher levels of TNF-α and IL-18. In accordance with these results, the levels of TNF-α and IL-18 in neonatal blood plasma were also higher in VD neonates when compared to CSD. Taken together, we observe a microbiome-driven relationship between delivery mode and endotoxin-induced immune system priming with the potential for lasting effects in later life.

## Results

**Study design and cohort characteristics**. To characterize the temporal patterns of earliest microbial colonization in relation to delivery mode, we recruited and sampled a total of 33 neonates (Supplementary Data 1). The neonatal gut microbiome of some of these neonates had previously been characterized using a combination of 16S rRNA gene amplicon sequencing and quantitative real-time PCR[7]. For a subset of neonates, well-matched neonatal and maternal samples were subjected to high-resolution metagenomic analyses. To differentiate between potential effects of CSD and/or SGA, neonates born by CSD and neonates born by CSD and being SGA were included in the cohort and analysed separately. For each mother–neonate pair, we sampled microbiomes of maternal body sites, which are indicated to be important in relation to neonatal gut colonization (collection of stool and vaginal swabs; Methods) less than 24 h before delivery. Additionally, earliest neonatal stool samples were collected at ≤ 24 h, 3 days and 5 days postpartum (63 samples; Supplementary Data 1). Extracted genomic DNA from all samples was subjected to 16S rRNA gene amplicon sequencing and extracted DNA from the samples of the subset of mother–neonate pairs were subjected to random shotgun sequencing. The 16S rRNA gene amplicon sequencing data were processed using NG-Tax[27], while the resulting metagenomic data were processed using a reproducible, reference-independent bioinformatic pipeline[28].

**Removal of artefactual sequences**. Neonatal stool samples from days 1–5 postpartum contain limited amounts of microbial DNA[7], and low-biomass samples are prone to over-representation of artefactual DNA that is introduced during the extraction procedure or preparation of sequencing libraries[24,26]. For the 16S rRNA gene amplicon sequencing data, any possible effect from putative artefactual reads was restricted by applying the methodology previously described in Wampach et al[7]. To account for the presence of artefactual sequences in the metagenomic data, we devised an additional, combined in vitro and in silico strategy to identify and remove artefactual sequences from

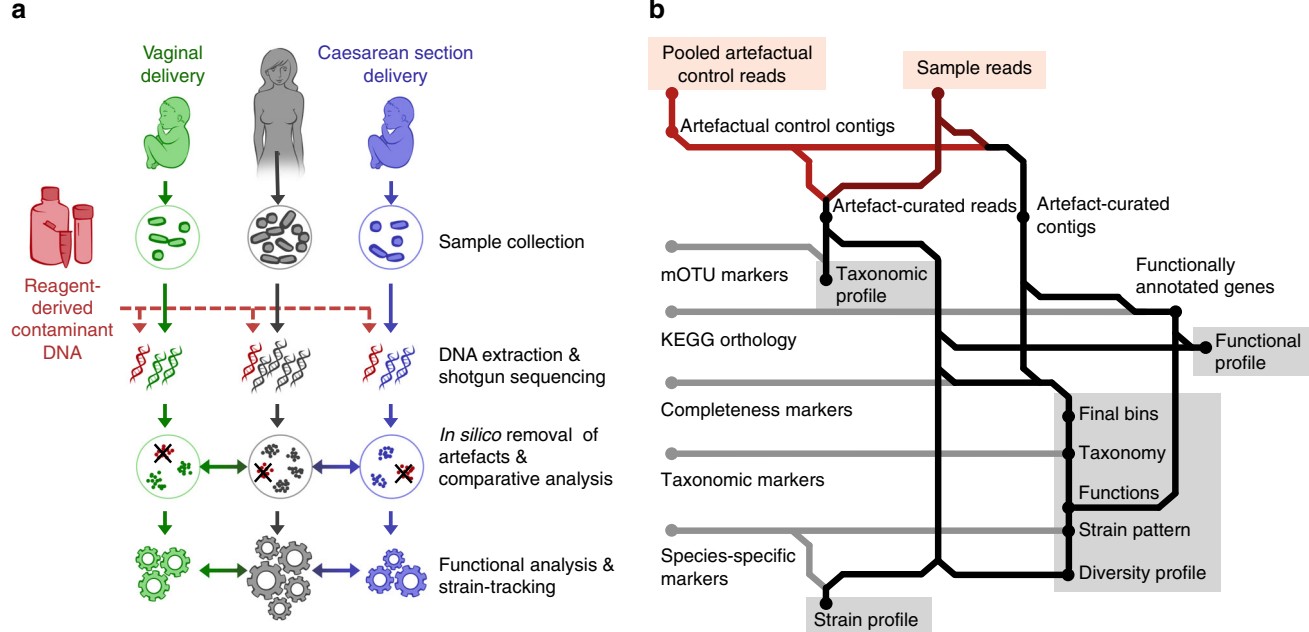

**Fig. 1** Curation of metagenomic data. **a** Schematic representation of the workflow for removal of artefacts introduced during genomic extraction or preparation of sequence libraries in the low-biomass neonatal samples. **b** Sample-wise bioinformatic workflow for removal of artefactual sequences from metagenomic data, extraction of taxonomic and functional profiles, and reconstruction of genomes and strain-resolved analyses. The resulting data sets used for inter-sample comparisons are highlighted in grey. mOTU, metagenomic operational taxonomic unit

the metagenomic data (Fig. 1a). For the in vitro part, DNA was extracted from a human gut epithelial cell line using the same procedure as for the neonatal stool samples and diluted to the levels of DNA extractable from the collected low-biomass samples (Methods). The choice of human DNA as a negative control was based on the following criteria: (i) the inability to generate a sequencing library from blank water control samples due to the inherent very low amounts of DNA (these are typically below the threshold for library construction); (ii) the ability to clearly differentiate signal (in the titration series: human sequences) from artefacts (non-human sequences); microbial DNA was not chosen as the homology between contaminant and bona fide sequences may have confounded delineation; (iii) the removal of human sequences is common practice when performing metagenomic analyses on human samples and appropriate methods exist to distinguish between human and microbial sequences in silico; (iv) the blinding of the variability originating from the laboratory environment or sequencing facility due to the nature of the samples (i.e., human control samples were treated with the exact same reagents as the faecal study samples). Our in silico workflow for the identification and removal of artefacts from metagenomic data (Fig. 1b) first clusters[29] contigs from the artefact control samples and the study samples together (Supplementary Fig. 1a). It subsequently removes contigs from study samples that cluster with the artefactual contigs, i.e., that fall into the same bin (Supplementary Note 1). After subsequent filtering steps and the successful removal of artefactual contigs from all study samples, we observed differences in the number of removed reads according to sample type (Supplementary Fig. 1b; Supplementary Data 2). On the basis of this essential data curation step, sequences from *Achromobacter xylosoxidans* or *Burkholderia* spp. taxa were for example identified and subsequently eliminated from the bona fide metagenomic data. Using the curated metagenomic data, we obtained taxonomic profiles (Supplementary Data 3), functionally annotated gene sets (Supplementary Data 4), reconstructed genomes following binning[30] and strain-determining variant patterns[31] (Supplementary Data 5).

**Earliest microbial taxonomic profiles**. The 16S rRNA gene amplicon and the metagenomic sequencing data, which were generated for a subset of mother–neonate pairs, showed highly similar succession trends in terms of diversity, evenness and richness measures (Supplementary Fig. 2a & b; Supplementary Note 2). The taxonomic profiles derived from the 16S rRNA gene amplicon and metagenomic sequencing were highly correlated (Supplementary Fig. 3a). The differences in taxonomic profiles according to delivery mode reflected results from previous studies, notably the higher relative abundance in *Bacteroides* and *Parabacteroides* and lower levels in *Staphylococcus* in VD neonates at days 3 and 5 postpartum[7,10] (Supplementary Data 6 to 8; Supplementary Note 3). In order to resolve the effect of delivery mode in relation to other potentially contributing factors such as maternal antibiotic intake prior to delivery, gestational age, feeding regime and sampling time point, differentially abundant taxa for both 16S rRNA gene amplicon and metagenomic sequencing data were determined separately using a multivariate additive general model approach (MaAsLin[32]). Taking into account the effects of the above-mentioned factors, delivery mode was found to be the dominant driver of neonatal gut microbiome colonization, with other measured factors having considerably less of an effect (Supplementary Note 4; Supplementary Data 9).

**Earliest functional differences according to delivery mode**. To assess whether the apparent taxonomic differences between the gut microbiomes of VD and CSD neonates are reflected at the level of functional potential, we used the metagenomic sequencing data to calculate Jensen-Shannon divergences for all samples (Supplementary Fig. 4a). Overall, comparison of the functional profiles of all neonates to the gut microbial potential of their respective mothers highlighted that the neonatal gut microbiota were more divergent from the maternal vaginal microbiota than the corresponding gut microbiota (Supplementary Fig. 4a, b & c). We also compared the CSD (±SGA) gut microbiota at day 3 and day 5 postpartum to those of VD neonates (Fig. 2a). CSD (±SGA)

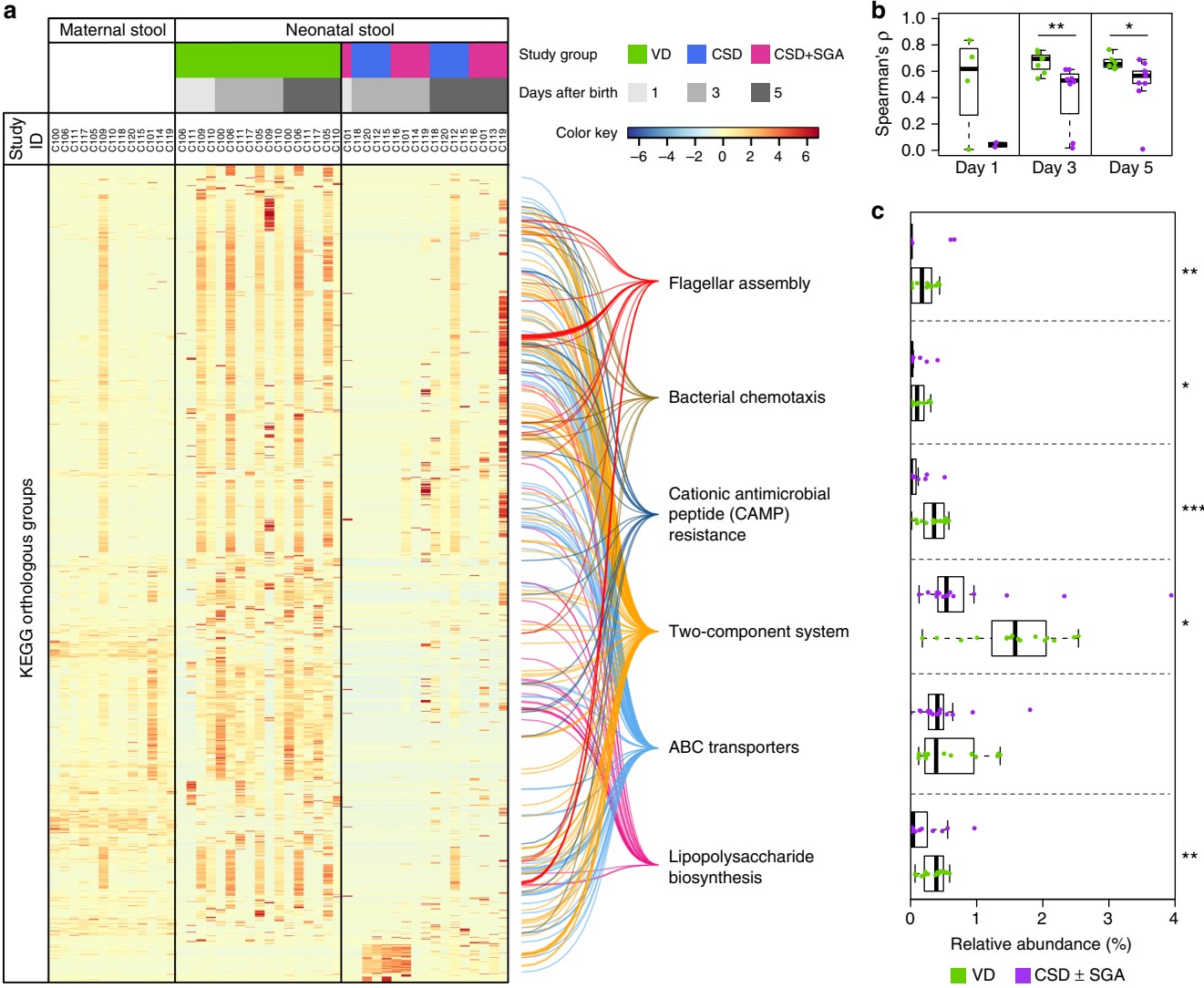

**Fig. 2** Maternal and neonatal gut microbiome functional profiles. **a** Heatmap of relative abundance of gut microbial orthologous gene groups with significant differential abundances in neonates born by vaginal delivery (VD) compared to either caesarean section delivery (CSD) or CSD with small for gestational age (SGA) status (CSD + SGA) groups and having the same direction of $\log_2$ fold change (calculated with the R package DESeq2[33] ; false-discovery-rate (FDR)-adjusted $P < 0.05$). Colour key indicates row-wise z-scores. The six significantly enriched pathways are indicated (FDR-adjusted $P < 0.05$). **b** Spearman correlation coefficients of functional profiles of neonatal and maternal gut microbiomes in VD and CSD (±SGA) neonates. **c** Cumulative relative abundance of enriched pathways indicated in **a** for VD and CSD (±SGA) groups. **b**, **c** Comparison by Wilcoxon rank-sum test for two-group comparisons with multiple testing adjustment; *FDR-adjusted $P < 0.05$, **FDR-adjusted $P < 0.01$, ***FDR-adjusted $P < 0.001$; Boxplots: centre line – median, bounds – first and third quartile, whiskers <= 1.5 × interquartile range

neonates lacked most functions at day 3 compared to VD neonates (Supplementary Fig. 5–10), while some appeared at day 5. Notably, neonatal-maternal correlations between community-wide functional potentials of the gut microbiomes at days 1, 3 and 5 postpartum were higher for VD than for CSD (±SGA) (Fig. 2b; Wilcoxon rank-sum test, FDR-adjusted $P = 6.0 \times 10^{-3}$ for day 3 and $P = 1.8 \times 10^{-2}$ for day 5).

We detected a total of 1,697 functional categories from the Kyoto Encyclopedia of Genes and Genomes (KEGG) Orthology (KO) database that were differentially abundant in the comparisons of the gut microbiome of CSD or CSD + SGA neonates to VD neonates. These presented the same directionality of change using the R package DESeq2[33] with a linear model considering the different collection time points containing at least 1,000 KOs (days 3 and 5) as covariates (Fig. 2a; Supplementary Data 10). Among the differentially abundant genes, there was an enrichment in genes involved in LPS biosynthesis (Fig. 2a; hypergeometric test,

false discovery rate (FDR)-adjusted $P = 1.5 \times 10^{-9}$), and the proportion of reads mapping to genes involved in this pathway was larger in VD neonates compared to both CSD groups neonate (Fig. 2c; Wilcoxon rank-sum test, FDR-adjusted $P = 9.6 \times 10^{-3}$). Other important microbial metabolic pathways, which were enriched with differentially abundant genes between VD and CSD (±SGA), included flagellar assembly (Fig. 2a; hypergeometric test, FDR-adjusted $P = 4.9 \times 10^{-12}$), bacterial chemotaxis (Fig. 2a; hypergeometric test, FDR-adjusted $P = 1.5 \times 10^{-2}$), cationic antimicrobial peptide (CAMP) resistance (Fig. 2a; hypergeometric test, FDR-adjusted $P = 4.0 \times 10^{-3}$), two-component system (Fig. 2a; hypergeometric test, FDR-adjusted $P = 2.5 \times 10^{-5}$) and ABC transporters (Fig. 2a; hypergeometric test, FDR-adjusted $P = 1.3 \times 10^{-4}$). As comparisons between VD and CSD as well as VD and CSD + SGA were largely matching independent of SGA status, we combined both groups (CSD and CSD + SGA) to increase statistical power (CSD ± SGA). Notably,

all pathways also showed higher relative gene abundances in VD compared to CSD (±SGA) neonates except for the ABC transporter pathway (Fig. 2c; Wilcoxon rank-sum test, FDR-adjusted $P = 4.1 \times 10^{-3}$, $3.8 \times 10^{-2}$, $2.2 \times 10^{-4}$, $2.1 \times 10^{-2}$, respectively).

To corroborate the apparent higher propensity of the VD microbiome for LPS biosynthesis, we annotated the OTUs resulting from the 16S rRNA gene amplicon sequencing data according to their attributed Gram staining information. Hereby, we observed that the gut microbiomes of VD neonates harboured significantly higher relative abundances of Gram-negative bacteria at days 3 and 5 compared to CSD (±SGA) neonates (Wilcoxon rank-sum test, FDR-adjusted $P = 1.7 \times 10^{-3}$ and $P = 4.0 \times 10^{-3}$ for day 3 and 5 respectively; Supplementary Fig. 3b). Additionally, the relative abundances of 7,000 KO functional categories were predicted using PanFP[34] based on the extensive 16S rRNA gene amplicon data (Supplementary Data 11). A multivariate analysis (MaAsLin[32]) was performed to compare the functional profiles of CSD (±SGA) to VD neonates and for both generated data sets (i.e., predicted KO functional categories based on 16S rRNA gene amplicon sequencing data and annotated KOs based on metagenomic sequencing data). Results from the multivariate analyses demonstrated that delivery mode was the strongest determining factor in both data sets (i.e., predicted and metagenomics-based KOs) for explaining the differentially abundant genes (Supplementary Data 9). Whilst not statistically significant, the trends for the predicted microbial pathways obtained with PanFP were largely concordant with the enriched pathways in VD neonates based on the differential analysis of the metagenomic data. Nevertheless, predictions of functional potentials based on 16S rRNA gene amplicon sequencing data are likely unreliable as a significant fraction of the gut microbiome (i.e., up to 40%) is represented by microorganisms without a sequenced isolate genome[35]. In contrast, the metagenomic data, through resolving the actual functional gene complement, allows a detailed comparison of the functional potential of the earliest gut microbiomes, as well as the tracking of individual-specific single-nucleotide variants (SNVs).

**Vertical transfer of enteric strains from mothers to neonates**. To determine if the observed differences in microbial functions were encoded by specific strains that were vertically transferred from the mother to the neonate, we mined the metagenomic sequencing data to identify microbial taxa and strains that both members of any of the 16 maternal–neonatal pairs, for which we had generated metagenomic data, had in common. We devised an ensemble approach to link reconstructed genomes on the basis of taxonomic annotations[30], similarity of phylogenetic marker genes and the presence of SNVs[31] (Methods). This enabled the tracking of specific strains from mothers to neonates (Supplementary Data 5). Given the high degree of specificity, the presence of transferred strains is highly relevant on a pair-by-pair, individual basis to assess mother-to-neonate transfer. This is all the more important given the extensive inter-individual variability of the neonatal gut microbiome (Supplementary Fig. 3c). Mother-to-neonate transfer differed between VD and CSD (±SGA), with significantly more maternal strains being shared by VD neonates than CSD (±SGA) neonates (Fig. 3a; Wilcoxon rank sum test: $P = 3 \times 10^{-3}$). While the reconstructed genomes of 25 taxa belonging to the phyla Proteobacteria and Firmicutes were identified in maternal–neonatal pairs for all birth modes (mostly the skin-derived and upper-gastrointestinal tract-inhabiting genera *Streptococcus* and *Staphylococcus* spp.), the reconstructed genomes of 23 enteric taxa belonging to the phyla Bacteroidetes and Actinobacteria (notably *Bacteroides* and *Bifidobacterium*

spp.) were exclusively observed in VD pairs. Notably, in the case of vaginal delivery, multiple strains of Gram-positive bacteria (e.g., *Bifidobacterium*) were transferred from mother to neonate (Fig. 3a; transmission in 71% of all VD neonates, 0% in CSD ± SGA on days 3 and 5), as well as Gram-negative bacteria (e.g., Bacteroidetes; Fig. 3a; transmission in 79% of all VD neonates, 0% in CSD and 20% in CSD + SGA on days 3 and 5).

**Linking differentially abundant functions to transferred strains**. To compare the levels of genetic divergence between early colonizing microbial populations over time, we calculated fixation indices ($F_{ST}$) and intra-population diversity ($\pi$), on the basis of the resolved SNVs. Our results reflected a shift in population structures during the transfer from mothers to neonates. We observed higher fixation indices (Wilcoxon signed-rank test: $P < 4 \times 10^{-3}$) between maternal and neonates' strains (M:3 and M:5; Fig. 3b; Supplementary Fig. 11a) compared to the same transferred strains observed at different times within the neonates (i.e., 3:5). Moreover, intra-population diversity tended to increase during the first days of life for strains that were not transmitted from the mother and belonged to the phylum Firmicutes, as seen in CSD compared to VD neonates (Wilcoxon rank sum test for day 5 versus day 3: $P = 1 \times 10^{-2}$; Fig. 3c; Supplementary Fig. 11b), suggesting that new strains invaded the neonatal gut during this short time period. The differences in relative abundance of taxa corresponded to the inferred routes of transmission linked to birth mode. For example, the metagenomic operational taxonomic unit (mOTU) *Bacteroides dorei/vulgatus* was more abundant in VD neonates, whereas *Staphylococcus epidermidis* was more abundant in CSD (±SGA) neonates (Fig. 3d, e). While the same strain of *B. vulgatus* was present in paired maternal and neonatal samples in the VD group (Fig. 3f), *S. epidermidis* strains were observed only in CSD neonates (Fig. 3g), suggesting an origin outside the maternal gut or vaginal environment. Taken together, these results are consistent with the transmission of strains from the maternal gut microbiome during vaginal delivery, resulting in relatively stable colonization of the neonatal gut during the earliest days, in contrast to CSD neonates.

To assess whether the transferred strains conferred specific functional traits to the neonate or not, we assessed the genomic complements of the earliest microbiota. Analysis of reconstructed genomes that were linked to maternal metagenomes showed that vertically transmitted strains were more likely to be enriched in functions that were depleted in CSD neonates (odds ratio (OR) 5.0, Fisher's exact test $P = 2.4 \times 10^{-11}$). Among these strains, Bacteroidetes (*B. vulgatus* and other *Bacteroides* species) and *Clostridium* spp. were common. Most strikingly, a *B. vulgatus* genome, which shared the majority of SNVs with the corresponding maternal reconstructed genome, was enriched in functions that were significantly more abundant in VD neonates compared to CSD (±SGA) neonates (OR = 3.7, FDR-adjusted $P = 3.0 \times 10^{-186}$; Fig. 3h). By contrast, strains of *Staphylococcus aureus*, an uncharacterized Actinobacterium and *S. epidermidis* encoded functions that were more prevalent in association with CSD. The reconstructed genome of *S. epidermidis* (Fig. 3i) was enriched (OR = 4.1, FDR-adjusted $P = 5.9 \times 10^{-57}$) in functions with higher relative abundances in CSD (±SGA) neonates, and considerably fewer SNVs were shared with the respective mother. These results indicate that vaginal delivery not only favours the vertical transfer of enteric strains from mother to neonate, but also results in the transfer of specific functional traits to the neonate, which are involved in important microbial pathways such as LPS biosynthesis and may be relevant in stimulating the developing immune system during the first days of life.

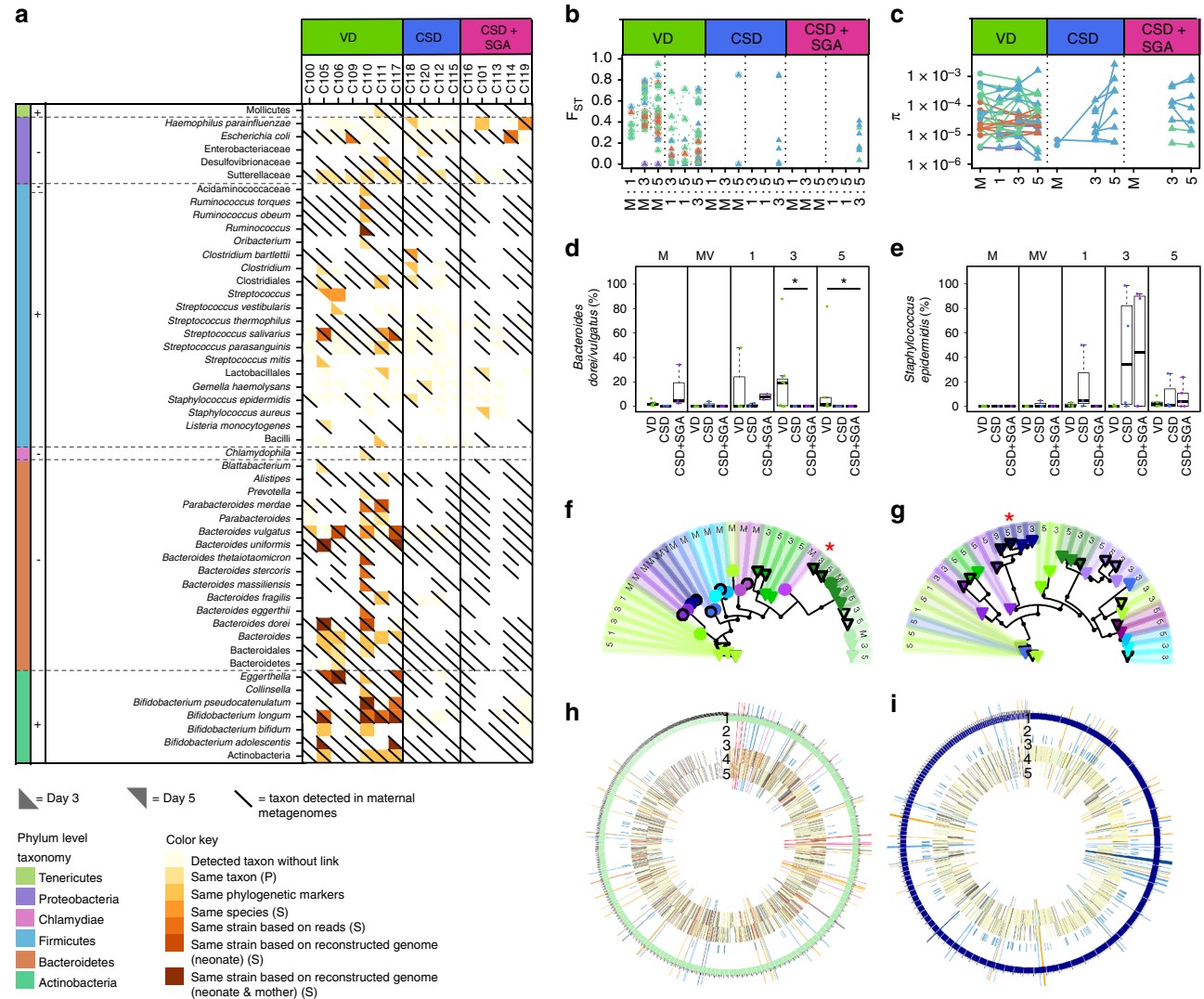

**Fig. 3** Transmission of functions by distinct microbial strains. **a** Taxa which were detected in gut microbiomes of mothers (diagonal line) and neonates (on postnatal day 3 (below the line) and/or day 5 (above the line), indicated by shading) in vaginal delivery (VD), caesarean section delivery (CSD) and CSD with small for gestational age (SGA) status (CSD + SGA) groups. The level of evidence of transmission is indicated by the shading colour, with darker shading for stronger evidence. A taxon without link describes a taxon that was found in the maternal samples, but not shared between mother and neonate. P based on PhyloPhlAn; S based on StrainPhlAn. Neonates C115 and C116 are twins. **b** Inter-population fixation indices ($F_{ST}$) comparing maternal (M) and neonatal (days 1, 3, 5) faecal samples. Phylum-level colour key is given in **a**. Encircled symbols highlight strains that are shared with the respective mother. **c** Intra-population diversity index ($\pi$). Circles and triangles represent maternal and neonatal faecal samples, respectively. **d**, **e** Relative abundance of the metagenomic operational taxonomic units (mOTU) belonging to *Bacteroides dorei/vulgatus* (**d**) and *Staphylococcus epidermidis* (**e**) in maternal faecal (M), maternal vaginal (MV) and neonatal faecal (days 1, 3, 5) samples from VD, CSD and CSD + SGA groups; *false discovery rate (FDR)-adjusted $P < 0.05$ in Wilcoxon rank-sum test for two-group comparisons; boxplots: centre line – median, bounds – first and third quartile, whiskers <= 1.5 x interquartile range. **f**, **g** Strain-level phylogenetic trees of *B. vulgatus* (**f**) and *S. epidermidis* (**g**); black bordered and borderless symbols represent genome reconstructions and read-based strain-level identity, respectively; genome reconstructions marked with a red asterisk are represented in **h** and **i**. **h**, **i** Genome reconstructions of *B. vulgatus* from neonatal faeces (C117; VD; day 3; bin P2.2.1) (**h**) and *S. epidermidis* from neonatal faeces (C112; CSD; day 5; bin P2.2) (**i**). Circular tracks represent: assembled contigs (1), single-nucleotide variants (black), shared between mother and neonate (red) (2), positions of strain markers (3), abundance fold-changes between VD and CSD (±SGA) neonates for functionally annotated genes in forward (4) and reverse directions (5); long spokes highlight genes affiliated with enriched pathways as depicted and colour-coded in Fig. 2a

**Immunostimulatory potential of the earliest gut microbiome.** As LPS forms part of the outer membrane of Gram-negative bacteria, the attributed Gram staining information of microorganisms directly corresponds to their propensity to synthetize LPS. Importantly, LPS is a highly potent innate immune activator that is recognized by the Toll-like receptor (TLR) 4. The earliest VD gut microbiome exhibited an enrichment in the microbial LPS biosynthesis pathway (Figs. 2a and 3h) as well as in Gram-negative taxa, which were frequently

transmitted from the mother (Fig. 3a). This observation is supported by the 16S rRNA gene amplicon sequencing data (Supplementary Fig. 3b). Consequently, an apparent higher microbial synthesis of LPS likely results in an increased immunostimulatory potential of the developing gut microbiome. To test whether the VD-associated colonizing gut microbiota, which encode a specific functional complement (including an enrichment in genes involved in LPS biosynthesis), drives early physiological differences in VD neonates, we focussed on

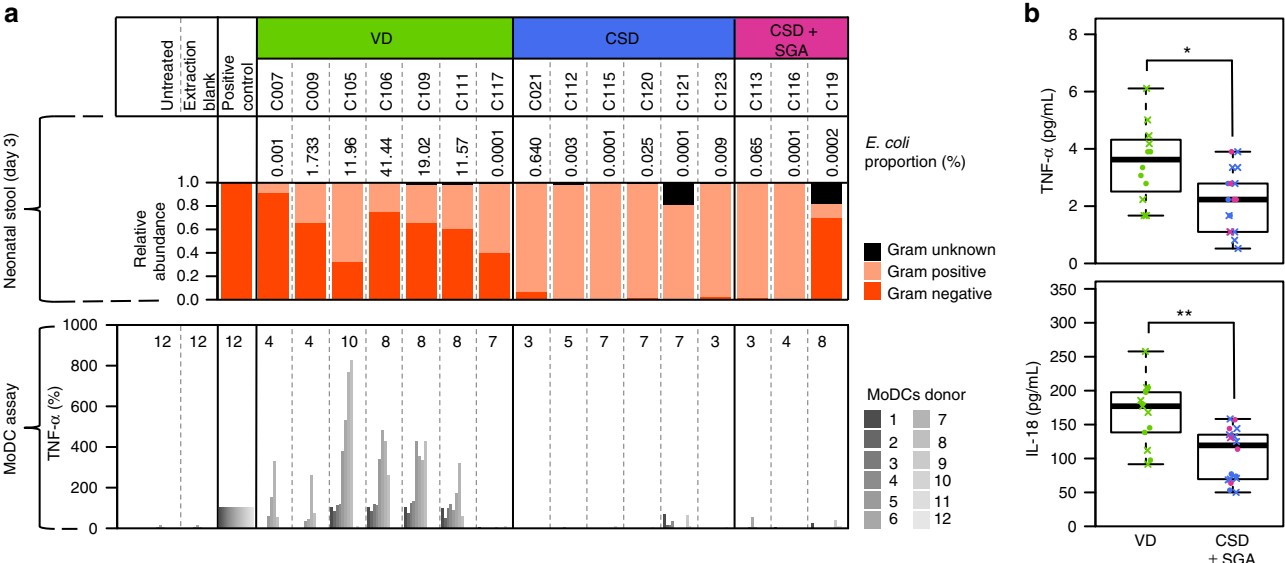

**Fig. 4** Cytokines of monocyte-derived dendritic cells after stimulation with LPS from neonatal stool and in neonatal plasma. **a** Lipopolysaccharide (LPS) was isolated from faecal samples collected on day 3 postpartum from neonates in groups of vaginal delivery (VD), caesarean section delivery (CSD) and CSD with small for gestational age (SGA) status (CSD + SGA), and incubated for 24 h with human monocyte-derived dendritic cells (MoDCs) isolated from a total of 12 adult donors. MoDCs were stimulated with the exact same LPS volume that was extractable from the same initial amount of faecal material from each neonate sample (Methods). Exact numbers of donors used per sample are given in the plot. Positive control: LPS isolated from *E. coli* overnight culture. Neonates C115 and C116 are twins. **b** Plasma levels of TNF-α and IL-18 in samples collected at day 3 after birth from VD and CSD (±SGA) neonates. Comparison by Wilcoxon rank-sum test with multiple testing adjustment; *false discovery rate (FDR)-adjusted $P < 0.05$ and **FDR-adjusted $P < 0.01$. Circles correspond to neonates with metagenomic data, crosses represent neonates without metagenomic data. Boxplots: centre line – median, bounds – first and third quartile, whiskers <= 1.5 × interquartile range

the early immunostimulatory potential of LPS from the neonatal gut.

Based on our data, the microbial composition differed most strongly in VD and CSD neonates on day 3 postpartum and thereby may critically affect the developing immune system at this time[12]. We isolated LPS from faecal samples from day 3 with sufficient biomass from 16 neonates (7 VD, 7 CSD, 2 CSD + SGA; Supplementary Data 12; Methods). We subsequently used several approaches to assess the purity of the isolated LPS fractions (Supplementary Note 5; Methods). Using agarose and polyacrylamide gel electrophoresis, we successfully visualized the isolated LPS and did not find traces of protein contamination but observed minor traces of fragmented DNA. However, this DNA did not contribute considerably to the immunostimulatory effect of the LPS fractions (Supplementary Fig. 12). No contamination with peptidoglycan or other bacterial molecules containing the D-glutamyl-meso-diaminopimelic acid moiety were detected in the isolated LPS fractions using the highly sensitive HEK-Blue™ reporter cells overexpressing the receptors hTLR2, hNOD1 and hNOD2, respectively (Supplementary Fig. 13; Methods). Some microbial products detected by hTLR2 (e.g., lipoteichoic acid, lipoprotein from Gram-positive bacteria, lipoarabinomannan from Mycobacteria or zymosan from yeast cell walls) were likely present in LPS samples for which high amounts of LPS were obtained from faecal samples. Conclusively, the isolated LPS fractions were assessed to be of high purity based on the HEK-Blue™ cell assays, although some unknown microbial products may play a stimulatory role in the high-yield LPS fractions (1 ng of standard LPS and an average of 2.9 ng of LPS isolated from VD neonates; Supplementary Fig. 13b). Consequently, the composition of the different isolated LPS fractions played a role in the subsequently triggered immune response.

To assess the stimulatory effect at the interface between the neonatal innate and adaptive immune systems[36], we stimulated

monocyte-derived dendritic cells (MoDCs) from 12 adult donors with neonatal LPS extracts (Methods). In order to reflect the in vivo situation as closely as possible, we stimulated the MoDCs with the exact same LPS volume that was obtainable from the same initial amount of faecal material from each neonate sample and subsequently measured levels of the LPS-inducible cytokine TNF-α in the supernatants using an ELISA assay (Fig. 4a; Supplementary Data 13). In parallel, a panel of additional cytokines was measured using an approach for quantifying and normalizing the employed LPS fractions (Methods). This was based on a maximum stimulation of MoDCs with 100 Endotoxin Units (EU) of LPS in order to mimic the amount of LPS an immune cell may encounter within a given neonatal sample (Supplementary Figure 14; Supplementary Data 14 and 15). The levels of all measured cytokines, and especially of TNF-α and IL-18, were higher in culture supernatants from MoDCs treated with LPS from VD neonates (Supplementary Fig. 14; Supplementary Data 16; Supplementary Note 6). Based on the outcome of all applied methods, we observed that the nature and composition of the different LPS subtypes contributed to the level of immune activation triggered by the LPS fractions from the different samples. Taken together, these results demonstrate higher immunostimulatory potentials of the earliest gut microbiome in neonates born by VD compared to CSD.

To test whether potential effects of differential immunostimulation are apparent early on in vivo, we assessed cytokine levels in plasma samples from a total of 31 neonates (Supplementary Data 1). Plasma samples were collected on the same day as the neonatal faecal samples from which LPS was isolated, i.e., day 3 postpartum. Levels of IL-18 were significantly higher in the VD group compared to the CSD group (Wilcoxon rank-sum test, FDR-adjusted $P = 2.4 \times 10^{-3}$; Fig. 4b), as were the levels of TNF-α (FDR-adjusted $P = 3.0 \times 10^{-2}$; Fig. 4b, Supplementary Data 17; Supplementary Note 7), which is consistent with previous results

in neonates at a similar age[37] and in healthy infants aged 0–3 months[38].

## Discussion

Here we employed high-resolution, artefact-curated metagenomic analyses of paired, high-quality samples from mothers and neonates to resolve the neonatal gut microbiome over the first few days of life. While previous studies have used analogous analytical approaches (16S rRNA gene amplicon sequencing and metagenomics) to resolve the early neonatal gut microbiome, these studies did not involve the systematic collection and appropriate preservation of paired mother–neonate samples[11], they did not specifically track vertical strain transfer[11,18], they did not include provisions for the removal of artefactual sequences[3,5,6,11,18], they did not focus on the earliest time points after delivery[3,18], nor did they resolve differences in functional potential according to delivery mode[3,5,6,11,18]. However, consideration of these factors is essential to assess the effect of delivery mode on the earliest transfer of community structure and function, subsequent microbiome colonization patterns and the resulting effects on neonatal physiology. Our cohort included paired sample sets from mothers (i.e., vaginal swabs and stool collected prior to delivery) and their respective neonates (i.e., stool collected at days 1, 3 and 5 and blood plasma at day 3) born by distinct delivery modes. All microbiome samples yielded high-resolution, artefact-curated sequencing data, which was analysed at the strain level.

As earliest neonatal gut microbiome samples are naturally of low biomass, the accurate identification and curation of potential artefactual sequences is essential. In the absence of appropriate controls, sequences derived from contaminant taxa in reagents may be relatively prominent, thereby masking actual signals and confounding results regarding in particular the transfer of taxa and functions from mothers to neonates. In our study, adequate controls were included and putative artefactual reads removed based on a combined in vitro and in silico workflow. In order to reach the required specificity to unambiguously address the question of vertical transmission of microbial community structure and function from mother to neonate, the use of curated, high-resolution metagenomic sequencing data (rather than solely performing 16S rRNA gene amplicon sequencing) is imperative. More specifically, the applied methodological approach allows the highly specific tracking of individual microbial functions and strains from mother to neonates on a case-by-case basis. Our results based on both 16S rRNA gene amplicon and metagenomic sequencing, and supported by multivariate analyses, demonstrate that early differences exist in the gut microbiomes of neonates and that these differences are predominantly driven by the mode of delivery. Our data agrees with previously reported differences in microbial composition related to birth mode, notably the increased relative abundance of *Bacteroides* and *Parabacteroides* in VD neonates as well as OTUs assigned to *Staphylococcus* being enriched in CSD neonates[3,4,7,39]. No fundamental differences in taxonomical compositions nor functional potentials were apparent when comparing CSD and CSD + SGA neonates, indicating that delivery mode is a stronger determinant for neonatal gut microbial colonization than SGA status.

In line with the broad taxonomic differences between VD and CSD, our findings demonstrate that CSD significantly affects the functional gene complement of the earliest neonatal gut microbiome by impeding the vertical transfer of specific bacterial strains from the maternal gut microbiome to the neonate. Consequently, the gut of CSD neonates is most likely colonized by strains derived from other sources, such as breast milk, skin or saliva, as suggested in previous studies[40–42]. Notably, a selection of enteric strains from the mother was found to be exclusively

transferred to the VD neonate (e.g., Bacteroidetes). When measuring population differentiation ($F_{ST}$), we found evidence that strains acquired from the mother are capable of quickly adapting to the new environment, as observed previously[5]. In contrast, vaginal strains harboured a low potential to stably colonize the neonatal gut thereby further adding to recently published data demonstrating that vaginal taxa do not play a prominent role in the initial colonization of the neonatal gut[5]. This may in part be explained by the distinct niches of the anaerobic gastrointestinal environment and the microaerophilic vaginal environment. With respect to ongoing clinical interventions aimed at restoring the earliest neonatal gut microbiome in the case of caesarean section[43], our findings raise questions over the expected efficiency of microbiota engraftment from a purely vaginal source and suggest that gut-derived strains may be more efficacious.

Independent of the precise mechanism of strain transfer, we observed that several functional pathways were significantly under-represented in CSD neonates, while these were in turn enriched in VD neonates and linked to vertically transmitted strains, in particular the LPS biosynthesis pathway (Fig. 2a). LPS, an outer surface membrane component of Gram-negative bacteria, promotes the secretion of pro-inflammatory cytokines and thereby sits at the interface of the earliest gut microbiome colonization and neonatal immune system priming. Following the apparent enrichments in LPS biosynthesis in VD neonates due to higher amounts of Gram-negative bacteria, the subsequent extraction and quantification of LPS from neonatal stool and stimulation of primary human immune cells therewith demonstrated a reduced immunostimulatory potential of the earliest gut microbiome in CSD neonates. The differences in earliest immune system priming may result in persistent effects on human physiology in later life, which has also been recently suggested based on work in a murine model[44]. On the basis of the observed immunogenicity of the purified LPS fractions, it has not escaped our attention that other factors, including the actual LPS composition, may additionally contribute to the difference in the immunostimulatory potential of the colonizing gut microbiome (Fig. 4a; Supplementary Note 6). Furthermore, other bacterial products, triggering for example TLR2, may contribute towards the observed higher immunostimulatory potential of faecal LPS from VD neonates (Supplementary Fig. 13; Supplementary Note 5). Considering the potential repercussions on neonatal physiology, the detailed elucidation of these additional factors will be the subject of future work.

Our study highlights differences in immunostimulatory potential of the earliest gut microbiome according to delivery mode. This occurs during a critical window of immune system priming. Notably, alterations to early immune system stimulation may be linked to the higher propensity of CSD infants to develop chronic diseases in later life[2]. For example, previous studies focusing on environmental exposure in early life have suggested that the exposure to Gram-negative bacteria and/or environmental endotoxins (such as LPS) could confer protective effects towards allergy development[45,46]. In this context, LPS is likely closely involved in the priming of the neonatal immune system and the subsequent tolerance towards the colonizing gut microbiome during a most critical window in early neonatal life[12–14]. Using a mouse model, it has been shown that strongly immunostimulatory LPS can contribute to the protection from immune-mediated diseases such as diabetes[47] and that disruption of host-commensal interactions in early-life can lead to persistent defects in the development of specific immune subsets[12]. On the basis of additional cytokine measurements in neonatal plasma, VD neonates displayed higher levels of IL-18 and TNF-α, thereby indicating a link between the immunostimulatory potential of microbial LPS in the gut and the overall immune status of the

neonatal host early on. Investigations of the longer-term consequences of these differences between CSD and VD neonates will be necessary to assess their possible impact on the development of chronic diseases in later life.

Apart from LPS biosynthesis, other pathways that were significantly enriched in the gut microbiome of VD neonates included genes involved in membrane transport, i.e., ATP-binding cassette (ABC) transporters. On the one hand this may reflect the adaptation of the colonizing microbiome of VD neonates to the gut environment through enhanced nutrient intake. On the other hand, associated ABC transporter proteins for both Gram-positive and Gram-negative bacteria have previously been shown to be immunogenic[48], which may suggest that they play a role in the activation of the neonatal immune system. Additionally, enrichments in pathways relating to bacterial motility were observed. These included the two-component system pathway, which is an important mediator of signal transduction, flagellar assembly and bacterial chemotaxis. These pathways are essential for bacterial motility in response to external stimuli and consequently also for competition with other members of the gut microbiome[49]. Additionally, flagellin, the main structural component of the flagellum, is an effective stimulator of innate immunity[50] and promotes mucosal immunity through the activation of TLR5[51]. Another functional pathway that is potentially interacting with the human immune system early on is the resistance to cationic antimicrobial peptides (CAMP). While this resistance has been observed in all major commensal phyla and across all members of the phylum Bacteroidetes, this pathway is essential to evade detection by the human immune system through the modification of the microbial LPS structure[52]. In the context of our study, an enrichment in CAMP resistance may therefore prevent the dominant colonizers (i.e., Bacteroidetes) from being recognized by the immune system and subsequently removed from the VD neonatal gut. Future studies are needed to assess whether the gut microbiome of VD neonates harbours more modified LPS moieties linked to CAMP resistance and which potential effects the altered LPS structures may have on the neonatal immune system. In accordance with the observation of an apparent enrichment in flagellar biosynthesis, bacterial chemotaxis and CAMP resistance, other microbiota-derived molecular factors, apart from LPS, may be involved in early immune system priming.

Our results imply that a more comprehensive understanding of the effect of the earliest microbial exposure on innate and adaptive immune responses and the different molecular factors involved in neonatal immune system priming is necessary. Future long-term follow-up studies based on larger cohorts, high-resolution multi-omic analyses, detailed immunological screening and tracking of health status will be essential to unravel the interdependencies between mode of delivery, other potential confounding factors, mother-to-neonate transmission, microbiome colonization, exposure to microbial factors, immune system priming and long-term health status. Furthermore, additional sources of maternal strains of importance in relation to microbiome-conferred molecular factors, besides the maternal vagina and gut, have to be considered to assess their relative importance in relation to their impact on neonate physiology. For this, additional samples may be obtained from maternal milk, skin, the oral cavity, and the hospital environment[5,11,40–42]. An additional focus should be placed on uncovering the source and mode of transfer of gut strains from mothers to neonates. Such mechanistic understanding will be important for devising future clinical interventions principally aimed at restoring a VD-like pioneering microbiota in the case of CSD. An alternative approach may consist of ensuring appropriate early priming of the neonatal immune system by the controlled provision of microbial antigens. Both avenues may provide the basis for the development of preventative strategies for adverse health effects in CSD neonates in the future.

## Methods

**Ethics.** Written informed consent was obtained before specimen collection from all enrolled mothers after a detailed consultation. All aspects of recruitment as well as collection, handling, processing and storing of samples and data were approved by the Luxembourg ethics board, the Comité national d'éthique de recherche, under reference number 201110/06 and by the Luxembourg National Commission for Data Protection under reference number A005335/R000058.

**Clinical metadata.** All study participants were enrolled and gave birth at the Centre Hospitalier de Luxembourg (CHL). Exclusion criteria for mother–neonate pairs included the administration of antibiotics to neonates immediately post-partum, birth prior to 34 weeks of gestation, and maternal gestational diabetes. Clinical metadata for all the analysed time points (days 1, 3 and 5 postpartum) are listed in Supplementary Data 1. Recorded metadata include information on the delivery mode, classification of caesarean section as elective or emergent, birth weight, gestational age, identification of the neonate as small for gestational age (SGA status) where relevant, gender, body length, weight and feeding regime. If a neonate received formula milk at any collection time point, the neonate was considered having received combined feeding for the remainder of the study, as even short-term formula feeding has been shown to cause profound and long-lasting shifts in the gastrointestinal microbiome composition[53]. Enrolled pairs of mothers and neonates ($n = 16$ pairs) included one twin birth (C115 (CSD) and C116 (CSD + SGA)).

**Sample collection.** Neonatal faecal samples were collected during the first 24 h as well as at days 3 and 5 after birth. Samples and data were collected at the CHL until day 3 after birth; subsequent samples were collected at home by trained study nurses. From the 33 neonates that were recruited into the study, the gut microbiome of 15 (Supplementary Data 1) had previously been characterized using a combination of 16S rRNA gene amplicon sequencing and quantitative real-time PCR[7]. For a subset of neonates, the mother was sampled additionally. Maternal samples (vaginal swabs and faeces) were collected less than 24 h before delivery. Samples were collected into sterile plastic tubes, immediately flash-frozen in liquid nitrogen and stored at −80 °C until further processing. Neonatal blood was collected by capillary or venous sampling, and plasma was isolated and stored at −80 °C from 31 healthy neonates (13 VD, 13 CSD, five CSD + SGA) at day 3 (28 samples) or day 5 (three samples) after birth, including 15 of the 16 neonates for whom metagenomic data were analysed (six VD, four CSD, five CSD + SGA), two neonates for whom no metagenomic but 16S rRNA gene amplicon sequencing data were available (two CSD) and 14 neonates (seven VD, seven CSD) that were sampled under the same conditions[7] (Supplementary Data 1). Clinical data were stored on secure servers at the Luxembourg Centre for Systems Biomedicine (LCSB), and biological samples were stored until further processing at the Integrated BioBank of Luxembourg (IBBL), which is NF S96-900:2011 certified.

**Sample processing and extraction of nucleic acids.** Genomic DNA was isolated from vaginal swabs with the PowerSoil DNA isolation kit (MO BIO Laboratories; Antwerp, Belgium) with an additional step to increase extraction yield involving the incubation of the samples in PowerSoil tubes with solution C1 at 65 °C for 10 min prior to homogenization for 5 min at 20 Hz in an Oscillating Mill MM 400 (Retsch, Haan, Germany). DNA was subsequently extracted following the manufacturer's instructions.

Faecal samples and cell-culture pellets were processed with the Powerlyzer PowerSoil DNA isolation kit (MO BIO Laboratories), optimized for low-yield samples. Bead solution (500 µl), C1 (60 µl), UltraPure™ Phenol:Chloroform:Isoamyl Alcohol (25:24:1, v/v; Invitrogen, Aalst, Belgium; 200 µl) and 50 mg neonatal stool or 150 mg maternal stool were added to a dry glass bead tube, incubated at 65 °C for 10 min, and homogenized by milling for 45 s at 4 m s$^{-1}$ in a FastPrep-24 5 G (MP Biomedicals, Illkirch-Graffenstaden, France). Samples were centrifuged for 1 min at 12,000 g. Solutions C2 (250 µl) and C3 (100 µl) were added to the supernatant and incubated at 4 °C for 5 min, centrifuged for 1 min at 12,000 g, then 700 µl of solution C4 and 600 µl of 100% ethanol were added to the supernatant and mixed. 650 µl were loaded onto a Spin Filter and centrifuged at 10,000 × g for 1 min. This step was repeated until all lysate had passed through the filter. For the higher input-mass maternal faecal samples, the same isolation procedure was followed except that the filters were washed with a mix of 300 µl solution C4 and 370 µl 100% ethanol, with centrifugation at 10,000 × g for 1 min. This latter step was omitted for the low input neonatal samples. All filters were washed with 650 µl 100% ethanol, then 500 µl solution C5. After drying, 60 µl solution C6 was added to the centre of the filter and incubated at room temperature for 5 min. DNA was eluted by centrifugation at 10,000 × g for 30 s. RNase A (100 µg ml$^{-1}$, 2 µl) was added and incubated at 37 °C for ≥ 30 min. Then, one-tenth volume 3 M sodium acetate (pH 6.8) and two volumes isopropanol were added to precipitate the DNA on ice prior to centrifugation. The pellet was washed with 150 µl 70% ethanol,

before the dried DNA was dissolved in 50 µl (neonatal faecal samples) or 100 µl (maternal faecal samples) RNase-free water. To obtain an artefact control sample, DNA was extracted from 800,000 trypsinized Caco-2 cells/ml. Caco-2 cells were grown in Dulbecco's Modified Eagle's Medium (Thermo Fisher Scientific, Ghent, Belgium) containing 20% v/v foetal bovine serum and 1% penicillin–streptomycin (Invitrogen) to prevent microbial growth. DNA was extracted with the low-biomass protocol described above, subsequently titrated and samples with 480, 240, 120, 60 and 30 ng total mass were sequenced. DNA integrity and quantity were determined for extracted samples of all origins on 1% agarose gels and in a Qubit 2.0 fluorometer (Thermo Fisher Scientific). Extracted DNA was stored at −80 °C until further use.

**DNA sequencing.** All DNA samples (along with 8 controls) underwent standard amplicon sequencing of the V4 region of 16S rRNA genes using primers 515F- 5′-GTGBCAGCMGCCGCGGTAA-3′ and 805R- 5′-GACTACHVGGGTATCTAA TCC-3′ at the Center for Analytical Research and Technology–Groupe Inter-disciplinaire de Génoprotéomique Appliquée (CART-GIGA; Liège, Belgium). Selected DNA samples of maternal (vaginal and faecal extracts), neonatal (faecal extracts at days 1, 3 and 5) and cell-culture origins were subjected to random shotgun sequencing (Supplementary Data 1). Metagenomic libraries were constructed with an optimized low-quantity DNA library preparation kit and sequenced on a HiSeq 2500 platform (Illumina) at GATC Biotech (Konstanz, Germany). For neonatal samples collected from C105, C109, C110 and C119 metagenomic libraries were prepared using TruSeq DNA Nano kit (Illumina) and sequenced on a NextSeq 500 platform (Illumina) at LCSB Sequencing Platform. A total of 84% of the study samples (63 of 75) collected from the mother–neonate pairs yielded sufficient DNA for metagenomic sequencing and sufficient artefact-curated metagenomic data for subsequent analyses.

**Metagenomic data processing.** Metagenomic data sets were processed with the Integrated Meta-omic Pipeline (IMP; version 1.3), which performs pre-processing, assembly, functional annotation of predicted genes and downstream analyses of Illumina next-generation sequencing metagenomic data in a single, reproducible workflow[28]. Illumina TruSeq3-PE-2 adapter sequences were trimmed from the reads in the pre-processing step (including the removal of human reads), and the de novo assembly step used the MEGAHIT[54] metagenome assembler. The IMP parameters were customized for different sample types: default parameters were retained for maternal faecal samples; for low-biomass samples (maternal vaginal swabs, neonatal faecal samples from days 1, 3 and 5 and cell culture sample), the integrated VizBin[29] sequence cut-off length was set to 1.

**Curation of metagenomic data from artefacts.** To identify and exclude artefactual sequences in the low biomass samples, contigs were assembled from the sequencing reads obtained from the DNA extracts of the Caco-2 cells after the removal of human reads. Given that the Caco-2 cells were cultured in the presence of 1% penicillin–streptomycin, that the routine surveys for *Mycoplasma* were negative, and that the metagenomic sequencing data did not include any *Myco-plasma* sequences, any bacterial contamination of the mammalian cell culture could be confidently excluded. Then, metagenomic reads from each study sample were mapped against these contigs using Bowtie 2[55] (version 2.0.2). Matching sequences were excluded prior to taxonomic profiling of metagenomic reads by phylogenetic markers[35]. As the artefactual sequences identified in the control samples did not represent full genomes, we further used a binning-based approach to identify additional potential artefactual sequences of the same organism among the de novo assembled contigs of the study samples. After removing the rRNA sequences from the contigs[56], we performed joint binning of control cell-culture contigs with each of the samples' contigs individually using VizBin[29] without any length cut-off. Bins were identified based on VizBin embeddings[56] using density-based spatial clustering of applications with noise (DBSCAN), without correction for the depth of coverage and completeness. All distinct bins (total length < 10 Mbp) that contained > 0.01% of the total contig length of the cell-culture control sample were considered putative reconstructed genomes of artefactual DNA, and the corresponding contigs were removed from the study samples in silico.

**Functional profiling.** Genes were predicted from contigs assembled with IMP and, after removing artefactual contigs, these genes were functionally annotated with hidden Markov models (HMMs)[56] trained for all KO[57] groups. The functional KO HMMs were aligned using HMMER 3.1[58,59]. The best hit KO (if multiple KOs could be assigned to a gene, the KO with the highest bit score was chosen) for every gene was assigned if the bit score was higher than the binary logarithm of the number of target genes. The FeatureCounts[60] tool with arguments –p and –O was used to extract the number of reads per KO (Supplementary Data 4; representing mean ± standard deviation 77 ± 13 % of all mapping reads).

**Linking genome reconstructions by marker gene sequence homology.** The curated contigs were binned based on the VizBin embeddings using DBSCAN as well as correction by the depth of coverage and completeness[56]. The reconstructed genomes of all samples belonging to a mother–neonate pair were merged into a union set. For each sample set, predicted amino acid sequences were searched

against and annotated using a defined set of essential marker genes[61] using HMMER 3.1[58]. Protein sequences assigned to 35 specific marker genes that form the cross-section of previously suggested sets of phylogenetic marker genes[61,62] were selected. These marker amino acid sequences were clustered with CD-HIT[63] at 97.5% identity. The frequencies of genes from different genome reconstructions co-occurring in the same clusters were determined. A simple graph network representation was constructed with the reconstructed genomes as nodes and counts of co-occurrences between two reconstructed genomes as weighted, undirected edges. Highly interlinked sub-networks, representing related reconstructed genomes, were detected with the cluster_fast_greedy algorithm[64] implemented in the R package igraph (v.1.0.1). The resulting reconstructed genomes from a given sub-network were manually inspected, and the taxonomy of reconstructed genomes was assigned using PhyloPhlAn[30] (Supplementary Data 5).

**Strain-level analysis.** Strains that occurred in multiple samples were determined with StrainPhlAn[31], using the pre-processed sequencing read data and reconstructed genomes. For each sample, taxonomic profiles were generated from pre-processed reads with MetaPhlAn2[65] using default settings. Strain reconstructions were extracted with the sample2markers.py script in StrainPhlAn with default arguments. StrainPhlAn was used to extract the clades detected in all samples and to construct reference databases for each clade. The sample-based strain reconstructions and reference databases of each clade and all reconstructed genomes were analysed with StrainPhlAn to build multiple sequence alignments and phylogenetic trees. The neonatal samples were considered to share strains with maternal samples if the cophenetic distance between the neonatal microbiome read-based or reconstructed genome-based markers and the maternal markers was less than the distance to the markers of any other individual. Trees were visualized with GraPhlAn (https://bitbucket.org/nsegata/graphlan/wiki/Home). To visualize the positions of markers in genome reconstructions, the reference markers of the species assigned to the reconstructed genomes in StrainPhlAn were aligned to the genome reconstructions post hoc, using blastn and an $E$ value cut-off of $1 \times 10^{-10}$, as in StrainPhlAn.

**Fixation index and intra-population diversity calculation.** For all neonatal reconstructed genomes that were estimated to be > 65% complete and linked to at least one other sample of the same neonate or their mother, the fixation index ($F_{ST}$) and the intra-population diversity ($\pi$) were assessed by the presence of SNVs. Metagenomic sequencing reads were mapped against the reconstructed genomes using MOSAIK[66] (version 2.2), with default parameters. A minimum alignment identity of 95% was applied to restrict the mapping to reads of the same species[67]. Genome–sample combinations generating alignments with a median coverage < 20X and/or a breadth < 40% were not included in downstream analyses. To reduce bias stemming from variation in coverage, alignments were down-sampled to a median coverage of 20X using Picard tools (version 1.85; http://broadinstitute.github.io/picard/). SNV calling was performed with FreeBayes[68] (version 1.1.0) using the -pooled-continuous option on the merged alignment files containing all samples for the same genome. Potential SNVs were required to be supported by four or more reads and to have an allele frequency ≥ 1%.

The output from FreeBayes (VCF-file) was used as input for POGENOM (https://github.com/EnvGen/POGENOM), a Perl-based tool that enables population-genomic analysis of metagenome samples. POGENOM was used to calculate the intra-population nucleotide diversity ($\pi$), which is defined as the average number of nucleotide differences per site between any two sequence reads chosen randomly from the sample population ($0 \le \pi < 1$). When reads of two or more samples mapped with sufficient coverage to the same genome, the fixation index ($F_{ST}$) was calculated, reflecting the population differentiation between a pair of samples. $F_{ST}$ is defined as one minus the average intra-population diversity of the samples divided by the nucleotide diversity between the samples (inter-population diversity). POGENOM was tuned to include only the loci recovered in all samples mapped to the same reference genome, assuring a valid comparison of the intra-species variation.

**Processing of amplicon sequencing data.** Analysis of the 16S rRNA gene amplicon sequences was performed with NG-Tax[27], with default parameters. Operational taxonomic units (OTUs) were assigned to the taxonomy in an open reference approach, using USEARCH[69] against the SILVA[70] 16S rRNA gene amplicon reference database (version 128; Supplementary Data 6). To exclude sequencing artefacts, only dominant phylotypes were examined by removing OTUs that were represented by fewer than 10 reads in the study samples.

**Analyses of taxonomic profiles.** To determine the Gram staining of the bacteria, we used the NCBI microbial attributes, which can be downloaded from http://www-ab2.informatik.uni-tuebingen.de/megan/taxonomy/microbialattributes.zip. Final Gram staining was assessed by main staining trends per genus and manually curated at the family and order levels. Functional community profiles were predicted based on OTU abundances using PanFP[34].

**Statistical data analysis.** The R statistical software package (version 3.3.3) was used for statistical analyses and visualization of the taxonomic profiles derived

from metagenomic and amplicon sequencing. Sum normalization and calculations of taxon richness (number of metagenomic OTUs (mOTUs) for metagenomic data or OTUs for amplicon sequencing data), diversity (Shannon), evenness (Pielou) indices and Spearman correlation coefficients were performed using the vegan R package. To discover differences in the data sets between the birth modes at the different collection time points postpartum, Wilcoxon rank-sum tests were applied, with FDR multiple-testing adjustment if applicable. Differential taxonomic abundances (according to delivery mode) were also calculated using ANCOM[71] with Benjamini-Hochberg multiple testing correction at 0.05 false-discovery rate. To determine the effect of the variables within the metadata, differentially abundant taxa were also determined using MaAslin[32] with default parameters and a $q < 0.05$ threshold for multi-testing correction. The model used was genus ~ sampling day + maternal antibiotic intake + feeding regime + gestational age. Differential analysis of KO abundance, comparing VD to CSD and VD to CSD + SGA with a linear model, which considered the different collection time points containing at least 1000 KOs (days 3 and 5) as covariates, was performed with the R package DESeq2 version 1.10.1[33]. KOs were considered significantly differentially abundant in VD and CSD (±SGA) if the FDR-adjusted $P$ value of the Wald test was < 0.05 for at least one comparison (CSD vs. VD or CSD + SGA versus VD) and the directionality of change in both comparisons was the same. Principal coordinate analysis (PCoA) graphs were generated using Jensen-Shannon distances as implemented in the R package phyloseq[72]. Differentially abundant pathways were detected through pathway enrichment analysis using a custom R script[56]. Tests for the enrichment of reconstructed genomes with differentially abundant KOs were performed using Fisher's exact test and FDR-adjustment for multiple testing in R.

**LPS isolation from neonatal faecal samples.** LPS was isolated from 16 selected neonatal faecal samples on the basis of availability of sufficient material. Samples (7 VD, 7 CSD, 2 CSD + SGA; Supplementary Data 12) were collected on day 3 after birth, and from overnight cultures of *Escherichia coli* strain K-12 (sub-strain MG1655). To maximise yields, LPS was purified from three aliquots of 50 mg of each neonatal faecal sample using the hot phenol–water method[73] and further purification was performed using a modified phenol re-extraction protocol[74]. For the *E. coli* control samples, three 5 ml overnight cultures were diluted to an optical density (600 nm) of 0.5 and centrifuged. LPS was isolated from cell pellets by the same protocol as above. LPS for each individual was pooled and quantified using an ELISA-based endotoxin detection assay (Endolisa; # 609033, Hyglos GmbH, Germany). From the 16 neonatal faecal samples, 11 produced measurable amounts of LPS, whereas 5 were under the detection limit (Supplementary Data 12). An extraction blank was generated using the same LPS isolation protocol.

**Quantitative real-time PCR to determine bacterial loads.** DNA from all neonatal faecal samples used for LPS isolation was diluted (when applicable) to a concentration of 1 ng l⁻¹ and amplified in duplicates with universal prokaryotic 16S rRNA gene primers 926F and 1062R[75] and with specific *Escherichia coli* primers Ec461F and Ec780R[76]. Primer sequences, annealing temperatures and cycle details are specified in Supplementary Data 14. Genomic DNA isolated from *Salmonella* Typhimurium LT2 and *E. coli* strain K-12 (sub-strain MG1655) was used to prepare standard curves for universal prokaryotic and specific *E. coli* primers, respectively. Reaction mixture, measurements and calculations of bacterial load (nanograms bacterial DNA per milligram stool and nanograms *E. coli* DNA per milligram stool) were performed as previously described[7] (Supplementary Data 14). The proportion of *E. coli* DNA in comparison to total bacterial DNA was subsequently calculated.

**In vitro immunostimulation using LPS from neonatal faecal samples.** Primary human monocytes were isolated from blood samples obtained from the Luxembourg Red Cross originating from twelve healthy adult donors. Human neonatal dendritic cells (DCs) were previously shown to be competent in MHC class I antigen processing and presentation to the same extent than adult DCs[77]. Most importantly, the NF-κB-dependent pathway in TLR-4 signalling is intact in neonatal MoDCs as they produce pro-inflammatory cytokines upon LPS stimulation, while adult and neonatal DCs are both able to produce comparable levels of TNF-α, IL-6 and IL-8 in response to LPS[78]. Isolated monocytes were differentiated into dendritic cells (MoDCs) in 12-well plates for 5 days in RPMI 1640 medium (Thermo Fisher Scientific) supplemented with 10% foetal bovine serum (Thermo Fisher Scientific), 20 ng ml⁻¹ each of granulocyte-macrophage colony-stimulating factor (Peprotech, London, UK), 20 ng ml⁻¹ IL-4 (Peprotech) and 1% penicillin–streptomycin (Invitrogen). To assess the immune stimulatory potential of isolated LPS, we treated MoDCs for 24 h with LPS extracted from VD or CSD (±SGA) neonatal faecal samples using two different methods; one based on LPS volume and one based on the normalization of LPS concentration with the bacterial load (see below for more information).

As we started from the same amount of material for all the neonatal stool samples and used the exact same extraction protocol to isolate all LPS fractions for all samples, we assumed that if we treated MoDCs from the same donor with the exact same volume of yielded LPS (independent of the concentration of LPS present), we would realistically emulate the microbial LPS load which immune cells

would be exposed to in vivo and thus be representative of the immunostimulatory potential of a given sample at 3 days postpartum. To stimulate MoDCs, 7.5 µl of LPS extract per $10^5$ MoDCs was added per well. For the negative control, MoDCs were incubated with 7.5 µl of LPS extraction blank, and for the positive control, MoDCs were treated with 15 endotoxin units (EU) LPS isolated from *E. coli* cultures. MoDCs were treated for 24 h to assess the immunostimulatory potential of the isolated LPS. Treatments were performed in duplicates and tested on at least three different donors. Culture supernatants from stimulated MoDCs were diluted 1/10 (or 1/50, if above standard curve range) and analysed for the presence of TNF-α using a commercial ELISA reagent set (Human TNF alpha uncoated ELISA, Life Technologies, Belgium) and a microplate reader (Biotek instruments, Germany).

For the second method, we verified our results using the bacterial load for normalizing LPS concentration values. Naturally, all faecal samples have a different bacterial load within the 150 mg of starting material that is used to isolate LPS. In order to assess if the differences we observed before with equal volumes of LPS (see above) were due to the fact that some samples have a much lower bacterial load or if also the bacterial composition (and proportion of Gram-negative bacteria) plays a role in the immunostimulation, we normalized the amount of LPS used to stimulate MoDCs with the bacterial load. For example, the bacterial load was highest for VD neonate C105 (Supplementary Data 15; Supplementary Fig. 14), and the corresponding bacterial load was 51.5 µg DNA per 150 mg stool. Therefore, this load was divided by the load in each other sample to yield a normalization factor. To stimulate MoDCs with 100 EU of LPS, 2.51 µl C105 LPS was added. For other samples, the LPS load was calculated by multiplying 2.51 µl by the previously determined bacterial normalization factor. For the negative control, $2 \times 10^5$ MoDCs were incubated with 15 µl of LPS extraction blank, and for the positive control $2 \times 10^5$ MoDCs were treated with 100 EU LPS isolated from *E. coli* cultures. Treatments were performed on cells from four distinct MoDCs donors ($2 \times 10^5$ MoDCs/donor), except for LPS isolated from C120, which was only sufficient to stimulate donor 4's MoDCs in duplicate. MoDCs and isolated LPS samples were incubated for 24 h. Culture supernatants from stimulated MoDCs were diluted twofold and analysed for the presence of seven cytokines (CXCL8/IL-8, IL-1β, IL-6, IL-10, IL-12p70, IL-18 and TNF-α) using a Human Premixed Multi-Analyte Kit (R&D Systems Europe; UK) and a MagPix multiplex reader (Luminex, Netherlands), according to the manufacturers' instructions (Supplementary Data 16). Statistical significance between the different cohorts was determined using the Wilcoxon rank-sum test.

**Coomassie blue and silver staining of LPS extracts.** On the basis of availability of sufficient extracted LPS material, 0.5 µg of extracted LPS from the stool samples of two VD neonates (C007 and C111) collected on day 3 after birth, were prepared with Laemmli sample buffer (Bio-Rad, Belgium), heated for 5 min at 95 °C and separated on 12 % Bis-Tris precast gel (Bio-Rad, Belgium) at 200 V for 45 min. As positive controls, 0.5 µg, 1 µg and 10 µg of commercially available LPS (*Escherichia coli* O55:B5, gel-filtration chromatography; Sigma-Aldrich, Belgium) and 10 µg of *E. coli* protein extract were used. A precast gel was loaded with the LPS samples and stained with Coomassie (Imperial protein stain, ThermoFisher, Belgium) to check for protein contaminations. Silver staining of the gel was performed using a corresponding kit (SilverQuest, ThermoFisher, Belgium) according to the manufacturer's instructions.

**Ethidium bromide staining of LPS extracts.** To check if LPS extracts were contaminated with immunostimulatory nucleic acids, 0.5 µg of extracted LPS from the stool samples of two VD neonates (C007 and C111), which presented highly concentrated LPS fractions that could be visualised on agarose gel, were prepared with DNA loading dye (ThermoFisher, Belgium) and loaded onto a 1% agarose gel. In addition, 0.5, 1 and 10 µg of commercially available LPS (*Escherichia coli* O55: B5, gel-filtration chromatography; Sigma-Aldrich, Belgium) were used to compare the purity of the LPS samples. As a positive control, 100 ng of *E. coli* DNA extract was used. The gel was stained with ethidium bromide, separated at 100 V for 50 min and analysed using a BioDocAnalyse system (Biometra, Germany). To check if nucleic acid contaminations could be identified in isolated LPS samples and would result in a TNF-α response (Supplementary Fig. 12) and purified using NucleoSpin Gel and PCR Clean-up kit (Macherey-Nagel, France). As controls, bands of *E. coli* DNA and commercially available LPS (10 µg) were cut out and purified. In addition, a purification blank was generated. The purified DNA fractions were used to stimulate MoDCs following the same protocol as for the stimulation with extracted LPS.

**HEK-Blue™ cell assay.** In order to verify the purity of the extracted LPS fractions, HEK-Blue™ reporter cell lines overexpressing one of the receptors hTLR2, hTLR4, NOD1 or NOD2 (InvivoGen, France), were stimulated with LPS extracted from five selected neonatal faecal samples (three VD and two CSD), which presented sufficient amounts of extractable LPS. HEK-Blue™ TLR and NOD cells are designed to detect stimulants of the human receptors by induction of secreted embryonic alkaline phosphatase (SEAP). For all the cell lines, the levels of SEAP were determined with HEK-Blue™ Detection (InvivoGen, France), a cell culture medium that allows for real-time detection of SEAP.

While the hTLR4 receptor only recognizes LPS, hTLR2 recognizes peptidoglycan, lipoteichoic acid and lipoprotein from gram-positive bacteria, lipoarabinomannan from mycobacteria, and zymosan from the yeast cell wall, the receptor NOD1 binds to bacterial molecules containing the D-glutamyl-meso-diaminopimelic acid (iE-DAP) moiety and NOD2 recognizes bacterial molecules (peptidoglycans) and stimulates an immune reaction. HEK-Blue™ cells were grown and maintained in DMEM ($4.5\,g\,l^{-1}$ glucose, L-glutamine, Sigma-Aldrich, Belgium), supplemented with 10% foetal bovine serum (Thermo Fisher Scientific), 1% penicillin–streptomycin (Sigma-Aldrich, Belgium), $100\,\mu g\,ml^{-1}$ Normocin (InvivoGen, France) and respective selective antibiotics according to the user's manual.

To monitor the activation of NF-κB, HEK-Blue™ cells were seeded according to the user's manual in HEK-Blue™ Detection medium (InvivoGen, France), in flat-bottom 96-well plates and stimulated for 22 h with LPS samples. We used two conditions: first, using the same concentration of LPS, where 1 µl of extracted LPS ($0.01\,ng\,\mu l^{-1}$) was added per well, and second, using the same volume of LPS, where 7.5 µl extracted LPS was added to $10^5$ HEK-Blue™ cells. To convert endotoxin activity (EU) into mass (ng), we considered that around 10 EU are equivalent to 1 ng endotoxin[79]. For positive controls, HEK-Blue™ NOD1 cells were stimulated with 1 µl TriDAP ($10\,\mu g\,\mu l^{-1}$, InvivoGen, France), HEK-Blue™ NOD2 cells with 1 µl Murabutide ($10\,\mu g\,\mu l^{-1}$, InvivoGen, France), HEK-Blue™ hTLR2 cells with 1 µl of Pam3CSK4 ($1\,\mu g\,\mu l^{-1}$; InvivoGen, France) and HEK-Blue™ hTLR4 cells with 1 µl ultrapure LPS ($5\,\mu g\,\mu l^{-1}$, source strain: ATCC 12014; CDC 5624-50 [NCTC 9701], InvivoGen, France). In addition, all cell lines were treated with 1 µl ultrapure LPS ($5\,\mu g\,\mu l^{-1}$, InvivoGen, France) and 1 µl ultrapure LPS ($0.01\,ng\,\mu l^{-1}$, InvivoGen, France) as well as with commercially available LPS (standard LPS; *Escherichia coli* O55:B5, gel-filtration chromatography; Sigma-Aldrich, Belgium): 1 µl of $5\,\mu g\,\mu l^{-1}$ and 1 µl of $0.01\,ng\,\mu l^{-1}$. For the negative control, HEK-Blue™ cells were incubated with 1 µl of endotoxin-free $H_2O$ (InvivoGen, France). All conditions were performed in duplicates and SEAP expression was monitored using a microplate reader at 655 nm (Biotek instruments, Germany) except for LPS isolated from C117 where only 7.5 µl extracted LPS/$10^5$ HEK-Blue™ cells was added to the cells and tested in duplicates.

**Cytokine profiling of neonatal plasma samples.** Plasma samples ($n = 31$) collected 3 or 5 days postpartum (13 VD, 13 CSD and five CSD + SGA; 28 samples collected at day 3, 3 samples collected at day 5 postpartum; Supplementary Data 1) were diluted twofold and analysed for 18 cytokines using a Human Premixed Multi-Analyte Kit (R&D Systems Europe) and a Bio-Plex analyser multiplex reader (Bio-Rad, Belgium), according to the manufacturers' instructions. The kit is able to detect CXCL8/IL-8, IL-1β, IL-6, IL-10, IL-12/23 p40, IFN-β, IL-15, IL-21, IL-5, Galectin-1, IFN-γ, IL-18, IL-27, Granzyme B, IL-13, IL-2, IL-4 and TNF-α. Of these cytokines, 11 were above the detection limit (CXCL8/IL-8, IL-6, IL-10, IL-15, IL-21, Galectin-1, IL-18, IL-13, IL-2, IL-4 and TNF-α; Supplementary Data 17).

**Code availability.** All custom scripts written for this study are available online at https://git-r3lab.uni.lu/Cosmic/Earliest.

## Data availability
The pre-processed, non-human metagenomic sequencing data and the amplicon sequencing data generated during the current study are available from NCBI under bioproject accession number PRJNA379120. A reporting summary for this Article is available as a Supplementary Information file.

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

## Acknowledgements

In silico analyses presented in this paper were carried out using the HPC facilities of the University of Luxembourg[80]. We are grateful to all the parents and neonates who participated in the study. We thank the dedicated clinical staff and neonatologists of the paediatric clinic and gynaecologists at the CHL for participant recruitment and sample collection, especially Alain Noirhomme and all involved study nurses of the Clinical and Epidemiological Investigation Center (CIEC) who performed sample and data collection at the CHL and at home as well as the scientific staff of the IBBL for sample storage. We are thankful for the assistance of Audrey Frachet and Janine Habier (LCSB) for laboratory support, Jochen Schneider (LCSB) for discussions, and Lola Kourouma for metadata entry and cross-checking of medical data. Lisa Morgenstern from GATC Biotech AG in Constance is thanked for her work and discussion on random shotgun sequencing and Wouter Coppieters and Latifa Karim from CART-GIGA for their work on 16S rRNA gene amplicon sequencing. We thank the MO BIO technical support team for recommendations on extraction kit optimizations. The present work was partially financed by the Fondation André et Henriette Losch. It was further supported by an ATTRACT programme grant (ATTRACT/A09/03) and CORE programme grants (CORE/15/BM/104040 and CORE/C15/SR/10404839) to P.W. and (CORE Junior/14/BM/8066232) to J.V.F., Aide à la Formation Recherche grants to L.W. (AFR PHD-2013-5824125) and S.N. (AFR PHD-2014-1/7934898), all funded by the Luxembourg National Research Fund (FNR). A.K. was funded by the University of Luxembourg (ImMicroDyn1). Sample collection, processing and storage were co-funded by the Integrated BioBank of Luxembourg under the Personalised Medicine Consortium Diabetes programme.

## Author contributions

L.W. carried out the biomolecular extractions, sequence annotation, did the comparative analyses of metagenomic and 16S rRNA gene amplicon sequencing data. A.H-B. coordinated the metagenomic measurements, carried out the strain-level analysis and performed genome reconstructions. Both L.W. and A.H-B. curated the metagenomic data

from artefactual sequences, performed binning, annotated called genes and interpreted the data. J.V.F. was involved in all the immunological assays, participated in the study design and data interpretation. J.R-G. carried out the processing of the 16S rRNA gene amplicon sequencing data and the subsequent multivariate analysis and functional predictions. J.H. carried out the LPS isolations, purity controls including gel staining and HEK-Blue™ cell assays and participated in all immunological assays. M.H. linked the obtained genome reconstructions in distinct samples by essential marker gene sequence homology, and S.N. carried out the sequence assembly and gene prediction. A.K. participated in biomolecular extractions and the bioinformatic assessment of metagenomic data. A.H.H. participated in the design of the study and storage of samples. C.d.B., L.B. and J.B. enrolled and consulted mothers and participated in sample and data collection. R.H. performed part of the metagenomic sequencing, P.M. carried out the functional profiling, and A.F.A. and C.S. performed the calculations of the fixation indices and intra-population diversities and participated in data interpretation. C.d.B. and P.W. conceived the study, participated in its design, performed data interpretation and coordinated the study. L.W., A.H-B., J.V.F. and P.W. wrote the manuscript. All authors read and approved the final manuscript.

## Additional information

**Competing interests:** The authors declare no competing interests.

