## [Peer Review File · Nature Communications]

Reviewers' comments:

Reviewer #1 (Remarks to the Author):

This very-well written paper is investigating the differences in the infant gut microbiome associated with mode of delivery (Cesarean section versus vaginal delivery). By using shotgun metagenomics, the authors describe the microbiome differences at multiple levels including the taxonomic composition, overall functional potential, and strain-level functional traits. Moreover, the study is supported by immunological assays (cytokine profiling), isolation of LPS from stool and tests in human cell lines. Overall the approach is sound and complete.

The main strengths of the paper are the combination of sequencing and functional profiling and the rigorous computational analysis, whereas the main limitation is the rather small sample size (12 mother-neonate pairs only 4 of which represent vaginal deliveries).

My main comments are listed below.

1. I think the sample size here is a bit small (12 mother-neonate pairs). Other studies of the infant gut microbiome are much larger, in the order of hundreds of infants such as PMID 28112736 and PMID 25974306. The problem of the small sample size is that some statistically significant differences in the functional potential could be driven by some specific strains enriched by chance in one of the considered groups. Can the authors test their conclusions using the data from PMID 28112736 and PMID 25974306? If their hypotheses are confirmed in these other two studies I think that the message would be much more statistically supported.

2. The main point of the paper is that birth mode (Cesarean against natural) influences the infant microbiome. However, other reports such as PMID 25974306 conclude that birth mode plays only a secondary role into the shaping of the gut microbiome. I think it is necessary that the authors directly deal with this partially conflicting theory, especially because this might again be related with the limited sample size.

Reviewer #2 (Remarks to the Author):

In this manuscript, Wampach et al test the hypothesis that microbes from mothers colonize their neonates, and then evaluate the metabolic functions encoded by the microbes that colonize the neonates. They compare the composition of neonates born by C-section vs. Vaginal delivery. The identity LPS, GAG degradation and other glycan degradation genes as being differentially abundant in VD neonates and find that stimulating primary human immune cells with LPS from VD neonate feces induces higher expression of TNF-alpha and IL18. These cytokines are also found to be more abundant in neonates born by VD compared to CSD. It does not appear that the authors controlled for potential confounders in this analysis, such as maternal exposure to antibiotics in the perinatal time period. The authors conclude that C-section influences mother-to-neonate microbe transmission and that this has immune-stimulatory effects with "likely effects" on human physiology later in life.

In general, the authors use appropriate and standard methods for sample collection from this small cohort, as well as appropriate methods for DNA extraction and data analysis. The method to remove "artifactual" reads is unusual based on my reading of work in this space, but as I am not an expert in neonate microbiome analysis, it is difficult for me to comment. Slightly older infant fecal samples typically have more than enough DNA, and thus are not low biomass. The

manuscript is, in general, well written, and the methods are well explained. The figures, though very dense, are of high quality.

Major concerns:

1) Recent manuscripts, notably Chu et al (Aagaard group) have put into question the model that bacterial strains are passed vertically from mother to child. Additionally re-analysis of existing data sets by Katie Pollard's group have done the same. This should be brought up in the introduction as this is a relevant, current, and very active and important debate in this field. Importantly, no data are provided regarding antibiotic exposure for the mothers or neonates. Maternal antibiotic exposure is probably a major confounder in these types of analyses, and should be explored. This should be possible as the authors have stated that they have a very rich and complete set of clinical metadata. Multivariable analysis are likely indicated to evaluate the impact of birth method on the questions at hand, especially as it relates to LPS production, for example.

2) The authors design a method that removes "artifactual" reads - this is useful for low biomass samples, but are fecal samples from neonates truly low biomass? Input DNA amounts for sequencing strategies using the Nextera prep, for example, require only 1ng of input DNA. I think the method the authors developed is reasonable, but it is interesting to note that most of the manuscripts that I have read on the topic of neonatal microbiome sequencing have not treated neonatal fecal samples as "low biomass". Does the removal of these "artifacts" result in dramatically different interpretations of the results?

Also, Regarding the choice of the artifact control sample, I found it interesting and unusual that Caco-2 cells were used - typically blank water controls are used for the controls, as mammalian cell culture likely is bacterially-contaminated. I find this choice rather unconventional.

3) In general, it would be nice if the authors pointed to the tools they use for "binning" (line 74), and identifying "strain-determining variant patterns" (line 75) in the actual manuscript. Also, how did they "reconstruct" genomes - they mention MEGAHIT in the methods, but it would be nice to cite this in the actual manuscript. Many of the references in the bioinformatic analysis section of the main manuscript appear to be incorrect or misplaced to me: for example, Ref 17 on line 74 appears incorrect (this referenced paper is not a binning tool as is implied by the location of the citation). Also, Ref 18 on line 75 appears to be incorrect. PhyloPhlAn is not a strain-determine variant pattern tool, as is implied by the location of the citation. In reading the methods, it is clear that the team engaged an experienced bioinformatician. It would be good if this bioinformatician carefully read over the main manuscript and checked all citations.

4) The inclusion of a significant proportion of infants who were small for gestational age (SGA) may have skewed results substantially. These are not healthy, usual, full-term births and are likely biological outliers. What is the justification/reasoning for inclusion of this group in the study? It was not clear to me.

5) For the LPS extraction, how did the authors account for samples where adequate LPS could not be extracted? Are we to conclude that LPS was absent? Did the authors measure the amount of LPS in the samples using methods such as targeted Mass spec? I am concerned that the size of the cohort studied and the limitations of not being able to extract LPS from all samples limits the generalizability of the findings presented. Also, why would babies born by VD have "higher immunostimulatory potential" than those born by CSD? A discussion of the expected impact of this observation would be helpful. Also, in the methods, it seems that of the 13 neonatal fecal samples, only 11 produced measurable quantities of LPS. In the main manuscript, I read this section as suggesting that 13 samples produced measurable quantities of LPS. Were samples that did not produce LPS from the VD, CSD or CSD+SGA cohorts? Why were only some of the total samples available for LPS extraction?

Given the very small n here, it seems that the results are depicted are of limited strength and may be very susceptible to over-interpretation.

Furthermore, based on Figure 4, it appears that Escherichia is highly abundance in all but one VD individual. Has this been described in other studies? Is this what is driving the LPS abundance in these samples? If so, this is worth discussing in the manuscript and putting into context for the readers. I would read the figure as Escherichia species being highly abundant in the neonatal microbiome of VD infants and that this may have consequences for LPS production. Is the E. coli that is present in these individuals the strain(s) that is from the mother? Based on Figure 3a, it does not appear so, as best as I can tell. So is one to conclude that VD neonates have a high abundance of E. coli that produces immunostimulatory LPS, but that the likely producer of this LPS (E. coli, which is abundant), is not vertically transmitted? If this interpretation is correct, I am not sure how the two main thrusts of the manuscript link to one another. Another area of confusion for me is EDFig 2C. In the neonatal stool from day 3, which I believe was used for the LPS extraction, I do not see many samples that have abundant Escherichia, as would be expected from Figure 4 - in fact, the samples that were chosen from the VD cohort for the LPS study appear to be the only two VD samples that have detectable E. coli. This may be skewing the results.

6) How was the LPS concentration and purity determined for each LPS sample that was extracted from feces determined? The effects that they saw may have to do largely with purity of the LPS as opposed to structural differences / immunogenicity of the LPS. I would be more convinced of the argument they are trying to make if each of the LPS samples had been subjected to mass spectrometry to evaluate purity, for example. I simply worry that the LPS extraction procedure is, in itself, very flawed and subject to error. And with such a small n, the results may not be terribly robust, even if intriguing.

Minor:

1) Not enough detail in the manuscript - how were differentially abundant functional categories from KEGG identified? I appreciate the strict space limitations, but some information regarding method should be included in the manuscript.

2) The authors refer to how the "early microbial functions in VD neonates reflected the mothers' functional microbiome profiles" on line 86-87, but they have not yet introduced any analysis of the mother's microbiome. This is confusing to me. Also, are they looking at the maternal vaginal microbiome or the maternal stool microbiome (both of which have been correlated with the neonatal microbiome). Did they look at the neonatal fecal microbiome and the maternal skin microbiome to compare functional potential? This would be suggested by Dominguez-Bello's paper from PNAS 2010.

3) The methods applied in Supplementary note 3 (to determine if birth method affected the microbiome) are not adequately explained and do not appear to be statistically driven. What is presented is not adequate to conclude that "the different feeding regimes of neonates did not explain the functional differences ..." on lines 94-95.

4) It seemed like the LPS biosynthesis gene was "cherry-picked" from a list of significantly differentially abundant genes. Also, what was the FDR cutoff? Why was this gene chosen? What were the most significantly differentially abundant genes?

5) GAG and other glycan metabolism is mentioned in the abstract but is only addressed in the supplement. If this is an important point, it should be included in the main manuscript. If not in the main manuscript, it should be left out of the abstract.

6) When cytokine levels were obtained from 31 neonates, it seems inappropriate to label the 14 additional infants as an "independent validation cohort" if the results are aggregated with the

discovery cohort. This is misleading to me.

7) The reference citation format appears to change at line 404. This is a small typographical issue.

Reviewer #3 (Remarks to the Author):

In this interesting research paper by Wampach et al, authors used a very robust bioinformatics approach to track specific bacterial strains from mothers to neonates during the perinatal period, taking into account levels of genetic diversity using well-established molecular ecology methods. This study has numerous strengths that are original and should be published: it combined taxonomic and functional profiling of the microbiome using reference independent metagenomics analysis, included controls and a bioinformatic approach that accounted for reagent and host derived contaminant DNA and removal of sequencing artifacts, and utilized sequencing data to inform the experimental parameters of an immune assay, which was informative of the potential interactions between LPS and the immune system. However, these strengths are not the main claim of this study. This clinical study was not designed nor was it shown to be powered to determine perinatal microbiome differences driven by mode of birth, and its results should not be interpreted as such. Below is a list of major and minor suggestions that should be incorporated in a new version of this manuscript.

Major comments

1. The sample number in this study is simply too low and there were no power calculations performed a priori to determine if it could identify difference between birth modes. Given that other small studies have failed to show microbiome differences driven by birth mode, it is hard to believe that this study was powered to assess the influence of birth mode, let alone the influence of low birth weight. I recommend that authors present power calculations based on previously published data to determine this. A good resource to perform these calculations for studies with multivariate data is MetSizeR. Clearly, the results from this study are in line with most other studies that have convincingly shown that mode of birth drives temporal neonatal composition and diversity. However, a recent study from the Xavier group (DOI: 10.1126/scitranslmed.aad0917) showed that while most CS births were associated with lower Bacteroides species abundance, about 20% of the VD babies displayed a similar pattern. With an N=4 per group (including 1 set of twins in the CSD groups) with only 2 valid time points (Day 1 only had 2 samples in 2 of the groups, so in my opinion should be excluded for comparisons), there is simply no statistical power to detect appropriate variance of a given population, especially considering the interindividual variability of the gut microbiome during the first days of human life.

2. Although differences between groups were detected using appropriate statistical tests, without confidence intervals and proper statistical tests to deconfound the effect of birth mode from other variables known to influence microbiome composition, it is not possible to properly assess whether birth mode truly drives these differences. Given that authors have access to 20 more samples (6 for which they had 16S data and 14 from a previous cohort that was collected in a similar way), the authors could use the 16S data and PICRUSt, at a minimum, to determine if other variables are also influencing the associations found between birth mode and microbiome composition. An excellent tool for this purpose is MsAsLin (<https://huttenhower.sph.harvard.edu/maaslin>). Variables to include in this model are gestational age, antibiotic use, dose, duration, time of meconium passage, if babies ingested colostrum, etc. In addition, authors should demonstrate that the samples used for metagenomics analysis are representative of the larger group of samples. For this, they could compare microbiome characteristics (beta or alpha diversity) as well as clinical variables (mode of birth, gestational age, etc.)

3. The results from the immune assay with extracted LPS from different neonatal samples are quite compelling but remain rather preliminary, and could be enhanced in several ways. As it stands, the purity of the LPS was not assessed, it is vaguely assumed that the majority of the cytokine response measured originated from E.coli LPS, and although it is mentioned in the text, it is not known if the differences in cytokine response are due to differences in LPS structure. Addressing at least 2 of these issues will significantly strengthen this paper. LPS purity can be determined by gel electrophoresis (comparing silver, protein and DNA staining, for example), or if available, mass spectrometry analysis. Differences in LPS structure (rough vs smooth, for example) can also be obtained through these methods. LPS purity can also be confirmed functionally by showing that the purified products did not activate Toll-like receptor 2 (TLR2), nuclear oligomerization domain 1 (NOD1), or NOD2 but did activate TLR4. Importantly, if the authors have more samples from other cohorts available, why not increase the number of samples from this assay? As for determining the origin of the LPS, could LPS genes not be assigned to the mOTUs given that the information is available? This is important as it will inform which taxa may or may not be relevant in stimulating immune cells during the first days of life.

4. Extended data figure 2. Change colour palette. Too many tones of blues and greens. Don't do colour coordination according to phyla.

Minor comments

1. Re. blinding in Life Sciences reporting summary: "As assigned study groups were predefined prior to delivery, blinding was irrelevant to our study." This is false. Many studies with predefined study groups are blinded and this adds strength to the study design. This study was simply non blinded, please change.

2. FIGURE 2: It is unclear how the DESEQ2 generated results are displayed in Figure 2. Deseq2 compares 2 groups. According to Figure 2a it looks that the comparison was made between VD samples and all CSD samples combined but figure legend and methods section describes two separate comparisons. Were the same 5 pathways differential in both comparisons or only when VD samples were compared to all CS babies?

3. FIGURE 4: The figure legend does not fully explain 4a. Specifically, if was only after I read the main text that I understood that the EU amounts were measured in the immune assay. I was confused and initially thought it was the LPS concentration used in the assay.

4. Line 87, vaginal or stool in mothers?

5. Line 94-96 briefly mention what statistical method was used (if any) to rule out the influence of feeding method

6. Lines 99-105 Where there any comparisons done between VD samples and each of the CSD groups. If so, this should be mentioned in this section. If not why separate the CSD groups?

7. Line 144, sources of microbial origin are too speculative, I suggest removing

8. Line 174 I suggest changing "more specifically" for "especially"

9. Line 202-203 Huge stretch here. There is no data on chronic diseases in this super small cohort. Also omit from the summary figure in extended data.

10. Lines 483-487 Unclear why bacterial DNA amount needed to be controlled in the LPS assay.

Please elaborate either here or in the Supplementary notes. These assays conditions will not mimic the amount of LPS that interacts with peripheral immune cells so I do not understand the reasoning behind this. Why not normalize by LPS units only? DNA quantification via qPCR of 16S RNA gene will vary depending on the number of 16S copies per cell so it is not the best way to quantify bacterial load. Further, this method will account for all bacterial cells, not just gram-negative (LPS containing cells). Flawed method.

11. Line 124 of Supplementary notes: I would change "corresponded" to correlated

12. The manuscript could use a brief discussion on the choice of adult blood DCs and how it may have differed from neonate peripheral immune cells. Neonatal immune cells are known produce immune responses similar to adults in some aspects but not others.

**Reviewer # 1**

**Overall summary**

**1.1.** This very-well written paper is investigating the differences in the infant gut
microbiome associated with mode of delivery (Caesarean section versus vaginal
delivery). By using shotgun metagenomics, the authors describe the microbiome
differences at multiple levels including the taxonomic composition, overall functional
potential, and strain-level functional traits. Moreover, the study is supported by
immunological assays (cytokine profiling), isolation of LPS from stool and tests in
human cell lines. Overall the approach is sound and complete.

The main strengths of the paper are the combination of sequencing and functional
profiling and the rigorous computational analysis, whereas the main limitation is the
rather small sample size (12 mother-neonate pairs only 4 of which represent vaginal
deliveries).

**Response:**

We appreciate the reviewer’s recognition of the quality and comprehensiveness of the
study. In light of the editor’s and reviewers’ comments, we have analyzed additional
16S rRNA gene amplicon and metagenomic sequencing data from collected samples
that cover the first 5 days *postpartum* (please refer to the table below for an overview
on the additional data that has been included in the revised manuscript;
**Supplementary Data 1**). Please also refer to our reply to **comment 1.2.** below for a
more detailed description of the additional samples collected as well as the additional
analyses performed. While the increased number of samples underscores the
statistical analyses, we want to emphasize that the main thrust of the manuscript is not
solely statistics-driven. Given their high degree of specificity, the presence of
transferred strains is relevant only on an individual, sample-to-sample basis in relation
to resolving mother-to-neonate transfer. In particular, using our high-quality data, we
are able to resolve the transfer of strains and linked functional repertoire on a
personalized level at the required level of specificity. This in turn is necessary to
address questions of mother-to-neonate transfer. Please also refer to our detailed
response to **comment 3.2.** for more information. Important to stress is that the results
from the analysis of the additional data support our original conclusions.

	Total number of samples	Total number of neonates	Study group composition
Initial dataset	130	18	7 VD, 6 CSD, 5 CSD + SGA
Updated dataset	176	33	15 VD, 13 CSD, 5 CSD + SGA

**Major comments**

**1.2.** I think the sample size here is a bit small (12 mother-neonate pairs). Other studies
of the infant gut microbiome are much larger, in the order of hundreds of infants such
as PMID 28112736 and PMID 25974306. The problem of the small sample size is

that some statistically significant differences in the functional potential could be
driven by some specific strains enriched by chance in one of the considered groups.
Can the authors test their conclusions using the data from PMID 28112736 and PMID
25974306? If their hypotheses are confirmed in these other two studies I think that the
message would be much more statistically supported.

**Response:**

We thank the reviewer for the relevant comment. While other studies with a focus on
the neonatal gut microbiome have involved apparent larger (albeit uneven for VD and
CSD) sample sizes, these studies have not involved the systematic collection and
appropriate preservation of paired mother and infant samples along the timeline of our
study or have not included very early neonatal samples. However, the collection of
such samples is essential to track differences between earliest microbiome
colonization in relation to delivery mode and to resolve potential vertical transmission
of specific strains and encoded functions from mothers to neonates. In other words,
only if the same sample types (i.e. stool) and collection time points (i.e. prior to
delivery for the mother and shortly *postpartum* for the neonate) are matched per
mother-neonate pair, one can objectively assess the vertical transfer of gut strains
from one mother to her respective neonate.

With reference to the studies cited by the reviewer in relation to potentially validating
our findings, PMID 28112736 presents only 2 out of 49 mother-infant pairs, for which
stool samples were collected from both mother and neonate shortly after birth. In both
pairs, infants were delivered vaginally, which makes the assessment of the delivery
mode effect on strain transfer impossible. On the other hand, PMID 25974306
presents 26 out of 98 mother-infant pairs for whom maternal stool samples were
collected at the day of delivery. In fact, previous analyses by Nayfach *et al.* (PMID
27803195) based on the data from PMID 25974306 were able to detect strain
transmission from mother to neonate in case of vaginal delivery. However, they were
unable to resolve any strains from the samples collected from CSD mother-neonate
pairs. Consequently, the effect of delivery mode in relation to strain transfer could not
be assessed properly. For our revised manuscript, our findings were put into context
with regards to these relevant previous studies especially within the extended text
limits of a *Nature Communications* article.

Introduction, manuscript lines 48 to 51: During vaginal birth, specific bacterial strains
are transmitted from mothers to infants³⁻⁶ and differences in microbial colonization in
neonates born by CSD have been identified⁷⁻¹⁰ as early as 3 days postpartum^{7,10}.

Introduction, manuscript lines 90 to 99: At the same time, several studies have
hypothesized that CSD impedes the vertical transfer of strains from mother to neonate
during delivery^{3,4,18,25}. In addition, although single nucleotide variants (SNVs) have
been tracked over time, no such studies have so far covered the earliest time points
after delivery (days 0-5) in well-matched mother-neonate pairs with respect to a direct

comparison of delivery modes. Consequently, there is a strong need for adequate
high-resolution metagenomic analyses capable of resolving the vertical transmission
of individual-specific strains and encoded functions from mothers to neonates on an
individual basis, while also supplementing observed *in silico* findings with further *in*
*vitro* validation experiments.

Discussion, manuscript lines 444 to 454: While previous studies have used analogous
analytical approaches (16S rRNA gene amplicon sequencing and metagenomics) to
resolve the early neonatal gut microbiome, they have not involved the systematic
collection and appropriate preservation of paired mother-neonate samples¹¹, they did
not specifically track vertical strain transfer^{11,18}, they did not include provisions for
the removal of artefactual sequences^{3,5,6,11,18}, they did not focus on the earliest time
points after delivery^{3,18}, nor did they analyse differences of functional potential
according to delivery mode^{3,5,6,11,18}. However, consideration of these factors is
essential to assess the effect of delivery mode on the earliest transfer of community
structure and function, subsequent microbiome colonization patterns and the resulting
implications for neonatal physiology.

In summary, the publicly available data is not appropriate for validating our findings.
In the vast majority of cases, the sample type and collection time point for both
mothers and neonates are not properly matched, which makes the assessment of strain
transfer and the linked transfer of functional potential impossible. Additionally,
although strain transfer was analyzed in previous studies, no effect of delivery mode
could be assessed due to a lack of neonates delivered by C-section.

In order to address the reviewer's valid concern regarding the overall sample size, we
have performed analyses of additional 16S rRNA gene amplicon and metagenomic
data from additional mother-neonate pairs (see table above under **comment 1.1**;
**Supplementary Data 1**). Moreover, we have performed additional analyses assessing
the physiological effects of the observed functional differences between CSD and VD
neonates (**Fig. 4a; Supplementary Fig. 11 and 12; Supplementary Data 12 and 13**).
The results of these analyses support our original conclusions, especially with regards
to the significantly increased relative abundance of Gram-negative bacteria in early
faecal samples of VD neonates as well as the significantly increased functional
potential of the earliest VD gut microbiome with respect to LPS biosynthesis.

Results, manuscript lines 252 to 258: To corroborate the apparent higher propensity of
the VD microbiome for LPS biosynthesis, we annotated the OTUs resulting from the
16S rRNA gene amplicon sequencing data according to their attributed Gram staining
information. Hereby, we observed that the gut microbiomes of VD neonates
harboured significantly higher relative abundances of Gram-negative bacteria at days
3 and 5 compared to CSD (\pm SGA) neonates (Wilcoxon rank-sum test, FDR-adjusted
$P = 1.7 \times 10^{-3}$ and $P = 4.0 \times 10^{-3}$ for day 3 and 5 respectively; Supplementary Fig.
3b).

**1.3.** The main point of the paper is that birth mode (Caesarean against natural)
influences the infant microbiome. However, other reports such as PMID 25974306
conclude that birth mode plays only a secondary role into the shaping of the gut
microbiome. I think it is necessary that the authors directly deal with this partially
conflicting theory, especially because this might again be related with the limited
sample size.

**Response:**

We thank the reviewer for this comment. The results of our study demonstrate that
early differences exist in the gut microbiomes of neonates born vaginally or via
caesarean section, and that these differences may impact early immune system
stimulation. Our findings do not conflict with the results of the study highlighted by
the reviewer as the authors of that study found, similar to our own results, that the gut
microbiome of vaginally delivered infants exhibits significantly greater resemblance
to the mothers' microbiome when compared to infants delivered by C-section (the
authors also found that the microbiome is shaped by nutrition; such differences are
however only apparent outside of the time window of our study). Our study adds
essential new elements to these earlier observations by describing the direct transfer
of specific strains from mother to infants in the context of vaginal delivery, the
differences in functional potential of the earliest microbiome conferred by these
strains and the potential physiological repercussions of these differences during the
first days of life. Thereby, our results do not sit at odds with previous observations but
expand on these and provide new mechanistic insights into the physiological
repercussions of a lack of transfer of specific microbiota from mother to infant in the
case of C-section. For the revised manuscript, we have included the reviewer's
suggested references in order to properly situate our study and results amongst the
currently partially conflicting results from other studies. In this context, we highlight
previous work, inconsistencies and general trends in the field of neonatal gut
microbiome colonization.

Introduction, manuscript lines 51 to 55: However, due to conflicting results, which
principally imply a negligible impact of delivery mode on the colonizing neonatal
microbiome in the gut¹¹, it remains unclear whether disruption of mother-to-infant
transmission of microbiota through CSD occurs and whether it affects human
physiology early on, with potentially persistent effects in later life.

Introduction, manuscript lines 65 to 76: While the majority of studies so far indicate
that delivery mode is the strongest factor determining early neonatal gut microbiome
colonization^{3,7-10,18}, these effects are either extenuated or largely absent in other
studies^{11,19}. Nevertheless, the possibility of microbial transfer from mother to neonate
during vaginal delivery cannot be excluded in studies which have reported a
negligible effect due to delivery mode¹¹. In this context, it is important to consider
that CSD may be performed as a result of underlying maternal or foetal medical

conditions (e.g. multiple gestation, foetal malpresentation or suspected foetal
macrosomia)²⁰ and can co-occur with other microbiome-influencing factors. More
specifically, CSD is most often accompanied by the administration of antibiotics to
mothers due to local health regulations or hospital practices (e.g. in case of a positive
screening of the mother for group B *Streptococcus*)²¹.

**Reviewer # 2**

**Overall summary**

**2.1.** In this manuscript, Wampach et al test the hypothesis that microbes from mothers
colonize their neonates, and then evaluate the metabolic functions encoded by the
microbes that colonize the neonates. They compare the composition of neonates born
by C-section vs. Vaginal delivery. The identity LPS, GAG degradation and other
glycan degradation genes as being differentially abundant in VD neonates and find
that stimulating primary human immune cells with LPS from VD neonate faeces
induces higher expression of TNF-alpha and IL18. These cytokines are also found to
be more abundant in neonates born by VD compared to CSD. It does not appear that
the authors controlled for potential confounders in this analysis, such as maternal
exposure to antibiotics in the perinatal time period. The authors conclude that C-
section influences mother-to-neonate microbe transmission and that this has immune-
stimulatory effects with “likely effects” on human physiology later in life.

In general, the authors use appropriate and standard methods for sample collection
from this small cohort, as well as appropriate methods for DNA extraction and data
analysis. The method to remove “artefactual” reads is unusual based on my reading of
work in this space, but as I am not an expert in neonate microbiome analysis, it is
difficult for me to comment. Slightly older infant faecal samples typically have more
than enough DNA, and thus are not low biomass. The manuscript is, in general, well
written, and the methods are well explained. The figures, though very dense, are of
high quality.

**Response:**

We appreciate the reviewer’s recognition of the main message of the study as well as
the overall high quality of the manuscript. In relation to the reviewer’s comment on
potential confounders, we have performed additional multivariate analyses (using the
multivariate tool MaAsLin), in relation to maternal antibiotics intake, day of sample
collection, gestational age, delivery mode and feeding regime. The results of these
additional analyses demonstrate that the maternal intake of antibiotics may have a
small effect on the neonatal gut microbiome composition but that the observed
fundamental changes in earliest community compositions are first and foremost due to
the delivery mode. For further information, please refer as well to our response to
**comment 2.2.** below.

Results, manuscript lines 194 to 202: In order to resolve the effect of delivery mode
from other potential contributing factors such as maternal antibiotic intake prior to
delivery, gestational age, feeding regime and sampling time point, differentially
abundant taxa for both 16S rRNA gene amplicon and metagenomic sequencing data
were determined separately using a multivariate additive general model approach
(MaAsLin³⁵). Taking into account the effects of the above-mentioned factors, delivery
mode was found to be the dominant driver of neonatal gut microbiome colonization,
with other measured factors having considerably less of an effect (Supplementary
Note 4; Supplementary Data 9).

Results, manuscript lines 260 to 267: A multivariate analysis (MaAsLin³⁵) was
performed to compare the functional profiles of CSD (\pm SGA) to VD neonates and for
both generated datasets (i.e. predicted KO functional categories based on 16S rRNA
gene amplicon sequencing data and annotated KOs based on metagenomic sequencing
data). Results from the multivariate analyses demonstrated that delivery mode was the
strongest determining factor in both datasets (i.e. predicted and metagenomic-based
KOs) for explaining differentially abundant genes (Supplementary Data 9).

Supplementary Information, Supplementary Note 4: The administration of antibiotics
to mothers prior to caesarean section delivery is often (e.g. in Luxembourg)
mandatory, resulting in delivery by caesarean section and the maternal intake of
antibiotics to be commonly coinciding factors. Additionally, vaginally delivering
mothers that are positively screened for group B *Streptococcus* infection receive
antibiotics prior to delivery to reduce the risk of infections in neonates. For these
reasons, most mothers were administered antibiotics in this study, for both delivery
modes. As other factors (e.g. day after delivery, gestational age, feeding regime, etc.)
have been suggested to impact neonatal gut microbiome colonization as well, we used
a multivariate analysis (MaAsLin³⁵) to identify differentially abundant taxa for both
metagenomic and 16S rRNA gene amplicon sequencing data (Supplementary Data 9).
After correcting for the respective effects of all the above-mentioned variables for the
metagenomic sequencing data, three mOTUs were associated with delivery mode
(CSD \pm SGA) and at the same time maternal antibiotics intake, namely *Bacteroides*
*fragilis* (for all conditions: $Q = 8.1 \times 10^{-3}$), *Bacteroides xylanisolvens* (for all
conditions: $Q = 8.1 \times 10^{-3}$) and *Parabacteroides merda* (for all conditions: $Q = 8.1 \times$
10^{-3}). In contrast, when correcting the different factors for the larger cohort screened
by 16S rRNA gene amplicon sequencing using MaAsLin³⁵, delivery mode had a
distinct driving effect on the earliest microbiome. More specifically, *Bacteroides* was
significantly decreased in CSD \pm SGA neonates ($Q = 2.6 \times 10^{-3}$ and $Q = 2.9 \times 10^{-2}$) as
well as Bacteroidaceae (CSD; $Q = 3.9 \times 10^{-2}$). One OTU of *Escherichia-Shigella* was
significantly decreased in CSD \pm SGA neonates (both $Q = 3.6 \times 10^{-2}$). Two OTUs
belonging to *Bacteroides* ($Q = 3.6 \times 10^{-2}$ and $Q = 3.9 \times 10^{-2}$) and two OTUs belonging
to *Bifidobacterium* ($Q = 1.6 \times 10^{-2}$ and $Q = 4.8 \times 10^{-2}$) were significantly decreased in
CSD neonates while one OTU belonging to *Staphylococcus* was significantly
increased in CSD neonates relative to VD ($Q = 2.3 \times 10^{-2}$). Feeding regime was found

to be a potential contributing factor of the relative abundances of *Trichococcus*
(formula feeding and mixed feeding regime; $Q = 3.6 \times 10^{-3}$ and $Q = 4.6 \times 10^{-2}$),
*Escherichia-Shigella* (mixed feeding regime; $Q = 1.3 \times 10^{-2}$) and one OTU belonging
to *Rothia* (formula feeding regime; $Q = 2.9 \times 10^{-4}$). The genus *Proteus* ($Q = 1.0 \times 10^{-2}$)
was associated with maternal antibiotics intake and multiple genera and OTUs were
associated with faecal samples collected at day 1 postpartum. Although we cannot
exclude an effect of maternal antibiotic exposure or minor effects of feeding regime
or collection time point on the taxonomic composition of the neonatal gut
microbiome, the main differences in microbial taxa in both datasets (16S rRNA gene
amplicon and metagenomic sequencing data) were clearly due to delivery mode.

An additional analysis to test for differentially abundant taxa associated with delivery
mode was performed using ANCOM⁷³. The results further confirmed the significantly
decreased relative abundances of *Bacteroides*, *Escherichia-Shigella*, *Bifidobacterium*
and *Parabacteroides* in CSD±SGA neonates, while *Staphylococcus* was significantly
increased in CSD±SGA neonate (all these results were based on the statistical
analyses based on Wilcoxon rank-sum tests, as well as the multivariate analyses). Our
results demonstrate that although several minor trends inside the earliest neonatal gut
microbiome were associated with distinct neonatal or maternal factors, the observed
fundamental changes in earliest community compositions were first and foremost due
to the delivery mode.

Discussion, manuscript lines 477 to 480: Our results based on both 16S rRNA gene
amplicon and metagenomic sequencing, and supported by multivariate analyses,
demonstrate that early differences exist in the gut microbiomes of neonates and that
these differences are predominantly driven by the mode of delivery.

Regarding the study design and employed methods, we would like to emphasize that
our study involved the systematic and careful collection of high-quality samples from
the first days of life. While indeed not routine, the failure of removing artefactual
reads may lead to serious flaws in interpretation, particularly when working on
samples containing a low microbial biomass. In the absence of appropriate controls,
sequences derived from contaminant taxa in reagents may be relatively prominent,
mask actual signals from taxa present and confound results regarding the presence of
actual taxa (doi:10.1038/nature.2014.16327; doi:10.1186/s12915-014-0087-z; doi:10.
1371/journal.pone.0110808; doi:10.1038/d41586-018-00664-8; our own recent work
on contaminant RNA in low-biomass samples: doi:10.1186/s12915-018-0522-7).
Failure to remove contaminants may in fact be partially responsible for the
conflicting results in earlier studies, especially in relation to apparent mother-to-
infant transfer. We are confident of the need for the applied methodological
workflow especially in relation to the removal of contaminant sequences and we
addressed this more specifically in the discussion of the revised manuscript.

Results, manuscript lines 153 to 180: To account for the presence of artefactual
sequences in the metagenomic data, we devised an additional, combined *in vitro* and
*in silico* strategy to identify and remove artefactual sequences from the metagenomic
data (Fig. 1a). For the *in vitro* part, DNA was extracted from a human gut epithelial
cell line using the same procedure as for the neonatal stool samples and diluted to the
levels of DNA extractable from the collected low-biomass samples (Methods). The
choice of human DNA as a negative control was based on the following criteria: (i)
the inability to generate a sequencing library from blank water control samples due to
the inherent very low amounts of DNA (these are typically below the threshold for
library construction); (ii) the ability to clearly differentiate signal (in the titration
series: human sequences) from artefacts (non-human sequences); microbial DNA was
not chosen as the homology between contaminant and *bona fide* sequences may have
confounded delineation; (iii) the removal of human sequences is common practice
when performing metagenomic analyses on human samples and appropriate methods
exist to distinguish between human and microbial sequences *in silico*; (iv) the
blinding of the variability originating from the laboratory environment or sequencing
facility due to the nature of the samples (i.e. human control samples were treated with
the exact same reagents as the faecal study samples). Our *in silico* workflow for the
identification and removal of artefacts from metagenomic data (Fig. 1b) first clusters³²
contigs from the artefact control samples and the study samples together
(Supplementary Fig. 1a). It subsequently removes contigs from study samples that
cluster with the artefactual contigs, i.e. that fall into the same bin (Supplementary
Note 1). After subsequent filtering steps and the successful removal of artefactual
contigs from all study samples, we observed differences in the number of removed
reads according to sample type (Supplementary Fig. 1b; Supplementary Data 2).
Based on this essential data curation step, sequences from *Achromobacter*
*xylooxidans* or *Burkholderia* spp. taxa were for example identified and subsequently
eliminated from the *bona fide* metagenomic data.

Discussion, manuscript lines 464 to 477: As earliest neonatal gut microbiome samples
are naturally of low biomass, the accurate identification and removal of potential
artefactual sequences is essential. In the absence of appropriate controls, sequences
derived from contaminant taxa in reagents may be relatively prominent, thereby
masking actual signals and confounding results regarding in particular the transfer of
taxa and functions from mothers to neonates. In our study, adequate controls were
included and putative artefactual reads removed based on a combined *in vitro* and *in*
*silico* workflow. In order to reach the required specificity (and thereby resolution) to
unambiguously address the question of vertical transmission of microbial community
structure and function from mother to neonate, the use of curated, high-resolution
metagenomic sequencing data, rather than solely performing 16S rRNA gene
amplicon sequencing, is imperative. More specifically, the applied methodological
approach, allows the highly specific tracking of individual microbial functions and
strains from mother to neonates on an individual basis.

**Major comments**

**2.2.** Recent manuscripts, notably Chu et al (Aagaard group) have put into question the
model that bacterial strains are passed vertically from mother to child. Additionally
re-analysis of existing data sets by Katie Pollard's group have done the same. This
should be brought up in the introduction, as this is a relevant, current, and very active
and important debate in this field.

Importantly, no data are provided regarding antibiotic exposure for the mothers or
neonates. Maternal antibiotic exposure is probably a major confounder in these types
of analyses, and should be explored. Multivariable analyses are likely indicated to
evaluate the impact of birth method on the questions at hand, especially as it relates to
LPS production, for example.

**Response:**

We thank the reviewer for this point and agree that the field of neonatal colonization
is currently actively discussing some contradictory findings. In accordance with the
reviewer's suggestion, we have discussed the previous work, inconsistencies and
general trends in the introduction of the revised manuscript. Apart from the issues
surrounding the impact of contamination on observed early colonization patterns (see
also our detailed response to **comment 2.1.** above), another important consideration
for inconsistent results regarding potential vertical transmission from mother to infant
and early microbial colonization is the fact that the methods used to date (particularly
16S rRNA gene amplicon sequencing) do not afford the required specificity (and
thereby resolution) to unambiguously address this question. In contrast, our
methodological approach based on metagenomic sequencing allows the highly
specific tracking of strains from mother to neonates on an individual basis, which is
necessary to unambiguously link specific organisms from specific mother-neonate
pairs. Nevertheless, it is important to stress that the work by Chu et al. (highlighted by
the reviewer) made observations which go in the same direction as our results, e.g.
that meconium samples harbor OTUs that originate from the maternal gut
(doi:10.1038/nm.4272). As these observations were based on 16S rRNA gene
amplicon sequencing, it is important to highlight that we were able to expand on these
previous suggestions and provide detailed results from corresponding high-resolution
metagenomic sequencing data. More specifically, we were able to perform strain-
tracking from mothers to infants on a case-by-case basis and assess differences at the
level of the functional potential.

Introduction, manuscript lines 83 to 88: Apart from confounding factors, the methods
and study designs employed over the past years may in part explain some of the
conflicting results regarding the effect of delivery mode on the early gut microbiome.
Notably, taxonomic profiling based on rRNA gene amplicon sequencing does not
offer sufficient resolution to assess the direct effect of the delivery mode at the level
of strain transmission, which is expected to be a determinant of succession.

C-section and maternal antibiotics intake are commonly coinciding factors, also in
multiple previous studies. In Luxembourg, the maternal administration of antibiotics
in case of C-section delivery is mandatory. Additionally, vaginally delivering mothers
that were positively screened for group B *Streptococcus* infection receive antibiotics
prior to delivery to reduce the risk of neonatal infection. Collectively, this reflects
why most study participants were administered antibiotics in the framework of this
study and for either delivery mode. According to the additional multivariate analyses,
which we have performed using MaAsLin, delivery mode was the most determinant
driver of the neonatal gut microbiome composition. However, when considering the
metagenomic sequencing data, three mOTUs, namely *Bacteroides fragilis*,
*Bacteroides xylanisolvens* and *Parabacteroides merda*, were associated with both
factors, delivery mode (CSD±SGA) and maternal antibiotics intake. Although we
cannot exclude an effect of maternal antibiotic exposure on the taxonomic
composition of the neonatal gut microbiome, the main differences in microbial taxa in
both datasets (16S rRNA gene amplicon and metagenomic sequencing data) are
clearly due to delivery mode. Please also refer to the manuscript passages highlighted
above under **comment 2.1**.

**2.3.** The authors design a method that removes “artefactual” reads - this is useful for
low biomass samples, but are faecal samples from neonates truly low biomass? I think
the method the authors developed is reasonable, but it is interesting to note that most
of the manuscripts that I have read on the topic of neonatal microbiome sequencing
have not treated neonatal faecal samples as “low biomass”. Does the removal of these
“artefacts” result in dramatically different interpretations of the results?

Also, Regarding the choice of the artefact control sample, I found it interesting and
unusual that Caco-2 cells were used - typically blank water controls are used for the
controls, as mammalian cell culture likely is bacterially-contaminated. I find this
choice rather unconventional.

**Response:**

We appreciate the reviewer’s points on the need for removing artefactual sequences in
case of working on low biomass samples. Given the extended length limitations of
*Nature Communications* (in contrast to the original *Nature* Letter format), we have
expanded on our rationale for the removal of artefactual reads in the revised
manuscript. Briefly, according to our earlier work (doi:10.3389/fmicb.2017.00738),
neonatal stool samples from days 1-5 contain between 0.0001 - 10 ng of microbial
DNA per mg of stool. As discussed in response to the reviewer’s **comment 2.1**.
above, low microbial biomass samples are prone to overrepresentation of artefactual
sequences, which may explain some of the inconsistencies between previous studies.
According to our own results, the removal of artefactual sequences is necessary to
resolve actual signals. More specifically, in our datasets, sequences from
*Achromobacter xylosoxidans* or *Burkholderia* taxa were found to be prominent in

both low biomass samples and negative controls and these stem from contaminated
reagents (doi: 10.1186/s12915-014-0087-z; doi: 10.1007/s10096-016-2644-6). Our
protocols allow for the diligent removal of such sequences, which is necessary to
resolve actual microbial community compositions and track strains from mothers to
neonates.

Given that the influence of artefactual sequences depends on the amount of DNA in
the original sample, we performed titrations with human DNA to assess the overall
complement of contaminant sequences analogous to previous work
(doi:10.3389/fmicb.2017.00738). Please also refer to the manuscript passages
highlighted above under **comment 2.1**.

The removal of artefactual sequences was performed across all of the analyzed
samples in a consistent manner and this removal was independent of the human
sequences, which were generated to identify artefactual sequences. We do not fully
understand the reviewer's comment about likely contamination of the mammalian cell
culture. Nevertheless, we would like to reassure the reviewer by stressing that the
Caco-2 cells from which the human DNA was extracted were maintained in DMEM
containing pen/strep. Furthermore, the presence of *Mycoplasma* is routinely
monitored in our cell culture lab. Based on these facts and as the metagenomic
sequencing did not include any *Mycoplasma* sequences, we can confidently exclude a
bacterial contamination of the mammalian cell culture and are confident that our
protocol is entirely robust for identifying and removing artefactual sequences.

Material and methods, manuscript lines 708 to 712: Given that the Caco-2 cells were
cultured in the presence of 1% penicillin–streptomycin, that the routine surveys for
*Mycoplasma* were negative, and that the metagenomic sequencing data did not
include any *Mycoplasma* sequences, any bacterial contamination of the mammalian
cell culture could be confidently excluded.

**2.4.** In general, it would be nice if the authors pointed to the tools they use for
“binning” (line 74), and identifying “strain-determining variant patterns” (line 75) in
the actual manuscript. Also, how did they “reconstruct” genomes - they mention
MEGAHIT in the methods, but it would be nice to cite this in the actual manuscript.
Many of the references in the bioinformatic analysis section of the main manuscript
appear to be incorrect or misplaced to me: for example, Ref 17 on line 74 appears
incorrect (this referenced paper is not a binning tool as is implied by the location of
the citation). Also, Ref 18 on line 75 appears to be incorrect. PhyloPhlAn is not a
strain-determine variant pattern tool, as is implied by the location of the citation. In
reading the methods, it is clear that the team engaged an experienced
bioinformatician. It would be good if this bioinformatician carefully read over the
main manuscript and checked all citations.

**Response:**

We appreciate the reviewer highlighting the importance for providing full
methodological details. All the methodological details were outlined in the Materials
and Methods section and in accordance with the requirements of *Nature*
*Communications*, a detailed Materials and Methods section now forms an integral part
of the manuscript. Also, we incorporated additional methodological steps that are
important for the understanding into the revised main text where necessary. There
were indeed two wrong citations in the original manuscript in relation to the
bioinformatic analyses, which have been corrected in the revised manuscript.

**2.5.** The inclusion of a significant proportion of infants who were small for gestational
age (SGA) may have skewed results substantially. These are not healthy, usual, full-
term births and are likely biological outliers. What is the justification/reasoning for
inclusion of this group in the study? It was not clear to me.

**Response:**

We thank the reviewer for the comment on infants that were born small for gestational
age (SGA). As detailed in the original manuscript, SGA neonates were included in the
study as this condition typically coincides with caesarean delivery. In addition,
neonates born SGA have an elevated propensity for developing metabolic disorders
during childhood or adulthood, and this elevated risk has been linked to changes in
the gut microbiome (doi:10.1186/2049-2618-2-38). Given that CSD may be linked to
SGA and that early changes to the microbiome may be at the origin of later-life
conditions, SGA infants were explicitly included to compare patterns observed in
them against non-SGA CSD neonates. Although the original manuscript included
references to the rationale, this was not explicit due to length limitations. In
accordance with the reviewer's comment, we have included additional explanations
on the rationale for including neonates born SGA in the revised manuscript.

Introduction, manuscript lines 76 to 81: Although CSD is not associated with
improved health outcomes, being born small for gestational age (SGA) frequently
coincides with CSD (i.e. more than 50% of all SGA neonates)^{22,23}. SGA neonates
have an elevated propensity for developing metabolic disorders during childhood or
adulthood, which has been associated with alterations to the gut microbiome^{22,24}, and
may be linked to the elevated rate of CSD in this population.

Discussion, manuscript lines 428 to 432: Based on all analyses, no differences in
taxonomical compositions or functional potentials were apparent when comparing
CSD and CSD+SGA neonates, suggesting that the impact of delivery mode was a
stronger determinant for neonatal gut microbiome colonization than the SGA status.

**2.6.** For the LPS extraction, how did the authors account for samples where adequate
LPS could not be extracted? Are we to conclude that LPS was absent? Did the authors

measure the amount of LPS in the samples using methods such as targeted Mass spec?
I am concerned that the size of the cohort studied and the limitations of not being able
to extract LPS from all samples limits the generalizability of the findings presented.
Also, why would babies born by VD have “higher immunostimulatory potential” than
those born by CSD? A discussion of the expected impact of this observation would be
helpful. Also, in the methods, it seems that of the 13 neonatal faecal samples, only 11
produced measurable quantities of LPS. In the main manuscript, I read this section as
suggesting that 13 samples produced measurable quantities of LPS. Were samples that
did not produce LPS from the VD, CSD or CSD+SGA cohorts? Why were only some
of the total samples available for LPS extraction?

**Response:**

We appreciate the reviewer’s reflections on the isolated LPS fractions and thank the
reviewer for the suggested additional experiments to strengthen the study. The fact
that sufficient LPS could not be isolated from samples can be explained by the limited
amounts of faecal samples that were obtainable. This does not mean that LPS was
absent in the samples collected from these neonates at the specific time points. Given
that the amounts of sample material which can be collected from neonates on days 1-5
were very limited and that large parts of the limited sample material was used for
DNA extraction, sufficient sample material was simply not left for LPS extraction
(the method requires 150 mg of material to result in sufficient LPS for downstream
analyses). Consequently, no data was obtainable for these samples. The amount of
LPS was quantified using state-of-the-art methods and we have now included
additional information on the characterization of the LPS fractions in the revised
manuscript (see also our detailed responses to **comment 3.4.** below). Although we
investigated analysis of LPS by mass spectrometry as per the reviewer’s suggestion,
such an analysis was not possible due to the very high amount of LPS needed, which
is not easily obtainable for early neonatal stool samples. Given the fact that LPS
moieties are highly diverse, such analyses would have not provided any conclusive
qualitative information and accurate quantification via chromatography coupled to
mass spectrometry would have also not been possible. In contrast, we have applied
additional, in our opinion more relevant, assays involving agarose gel electrophoresis
and reporter cell lines to further characterize the obtained LPS fractions (see also our
detailed response to **comment 2.9.** below).

In order to address the reviewer’s concern regarding the generalizability of the results,
we have included data from additional samples in the revised manuscript, which,
along with the more detailed characterizations of the LPS fractions, support our
original conclusions. Collectively, we now present isolated LPS mixtures from faecal
samples collected at day 3 postpartum from a total of 16 neonates (7 VD, 7 CSD, 2
CSD + SGA). The apparent “higher immunostimulatory potential” of the early
microbiome in VD neonates is, according to the results in the original manuscript and
supported by the new data, due to the higher numbers of Gram-negative bacteria and
the enrichment of the LPS biosynthesis pathway in the gut microbiome of VD

neonates. In order to clarify this point, we have included an additional section in the
revised manuscript. The reason for only some samples producing measurable
quantities of LPS has been described and clarified in the revised manuscript. Please
also refer to revised text passages cited under **comment 1.2**.

Results, manuscript lines 298 to 303: Notably, in the case of vaginal delivery,
multiple strains of Gram-positive bacteria (e.g. *Bifidobacterium*) were transferred
from mother to neonate (Fig. 3a; transmission in 71% of all VD neonates, 0% in CSD
\pm SGA on days 3 and 5), as well as Gram-negative bacteria (e.g. Bacteroidetes; Fig.
3a; transmission in 79% of all VD neonates, 0% in CSD and 20% in CSD+SGA on
596 days 3 and 5).

Results, manuscript lines 353 to 362: As LPS forms part of the outer membrane of
Gram-negative bacteria, the attributed Gram staining information of microorganisms
directly corresponds to their propensity to synthesize LPS. Importantly, LPS is a
highly potent innate immune activator that is recognized by the Toll-like receptor
(TLR) 4. The earliest VD gut microbiome exhibited an enrichment in the microbial
LPS biosynthesis pathway (Fig. 2a and Fig. 3h) as well as in Gram-negative taxa,
which were frequently transmitted from the mother (Fig. 3a). This observation is
supported by the 16S rRNA gene amplicon sequencing data (Supplementary Fig. 3b).
Consequently, an apparent higher microbial synthesis of LPS likely results in an
increased immunostimulatory potential of the developing gut microbiome.

Discussion, manuscript lines 502 to 512: Independent of the precise mechanism of
strain transfer, we observed that several functional pathways were significantly under-
represented in CSD neonates, while these were in turn enriched in VD neonates and
linked to vertically transmitted strains, in particular the LPS biosynthesis pathway
(Fig. 2a). LPS, an outer surface membrane component of Gram-negative bacteria,
promotes the secretion of pro-inflammatory cytokines and thereby sits at the interface
of the earliest gut microbiome colonization and neonatal immune priming. Following
the apparent enrichments in LPS biosynthesis in VD neonates due to higher amounts
of Gram-negative bacteria, the subsequent extraction and quantification of LPS from
neonatal stool and stimulation of primary human immune cells therewith
demonstrated a reduced immunostimulatory potential of the earliest gut microbiome
in CSD neonates.

**2.7.** Given the very small n here, it seems that the results are depicted are of limited
strength and may be very susceptible to over-interpretation.

**Response:**

We analyzed the data from faecal samples from additional samples and have included
the results in the revised manuscript. The analyses from the additional samples
support the original results and conclusions. Please also refer to **comment 1.2** for

additional details.

**2.8.** Furthermore, based on Figure 4, it appears that *Escherichia* is highly abundance
in all but one VD individual. Has this been described in other studies? Is this what is
driving the LPS abundance in these samples? If so, this is worth discussing in the
manuscript and putting into context for the readers. I would read the figure as
*Escherichia* species being highly abundant in the neonatal microbiome of VD infants
and that this may have consequences for LPS production. Is the *E. coli* that is present
in these individuals the strain(s) that is from the mother? Based on Figure 3a, it does
not appear so, as best as I can tell. So is one to conclude that VD neonates have a high
abundance of *E. coli* that produces immunostimulatory LPS, but that the likely
producer of this LPS (*E. coli*, which is abundant), is not vertically transmitted? If this
interpretation is correct, I am not sure how the two main thrusts of the manuscript link
to one another. Another area of confusion for me is Fig2C. In the neonatal stool from
643 day 3, which I believe was used for the LPS extraction, I do not see many samples
that have abundant *Escherichia*, as would be expected from Figure 4 - in fact, the
samples that were chosen from the VD cohort for the LPS study appear to be the only
two VD samples that have detectable *E. coli*. This may be skewing the results.

**Response:**

We thank the reviewer for this detailed comment. The apparent difference in the
abundance of *Escherichia coli* across the different VD neonates (Figure 4) was
highlighted in the original manuscript. According to our results, *E. coli* must indeed
be important for early LPS-based immunostimulation in many neonates but not all.
According to our results (Figure 3a), other Gram-negative bacteria, which contribute
to the isolated LPS mixtures and were transmitted from mothers to VD neonates, may
occupy the same role (Figure 3h). For the revision of the manuscript, we have
performed additional measurements, in particular qPCR analyses to quantify *E. coli*
abundances (see paragraph below) that allow us to validate these findings. The
additional data has been included in the manuscript to clarify this point. In the revised
manuscript, we have discussed the apparent importance of *E. coli* as well as other
Gram-negative bacteria with respect to LPS-mediated immune stimulation early on in
life. Accordingly, we have included a supplementary note on the potential effect of *E.*
*coli* abundance and linked immunostimulatory potential in the revised manuscript.

Supplementary Information, Supplementary note 6: LPS from VD neonate C117, in
whom the microbiome displayed lower presence of Gram-negative bacteria based on
16S rRNA gene amplicon sequencing data and the lowest amounts of *E. coli*
according to qPCR measurements, also had the lowest immunostimulatory potential
among all VD neonates, and triggered a negligible cytokine response. Although a
higher *E. coli* abundance appeared to coincide with an increased immunostimulatory
potential of the isolated LPS, the proportion of *E. coli* alone was not sufficient to
explain the lack of immunostimulative effects in CSD (\$\pm\$ SGA) neonates. More

specifically, in five cases of faecal samples collected from CSD+SGA, the absolute
 abundance of *E. coli* was at least 25-fold increased when compared to the faecal
 sample collected from VD neonate C007, which triggered an immune response (Fig.
 4a). However, the isolated LPS from these five samples did not result in any
 considerable immune response in terms of TNF- α production. Additionally, in CSD
 neonate C121, the microbiome was depleted of *E. coli*, while the extracted LPS
 fraction still triggered an immune response. At the same time, CSD+SGA neonate
 C119 was depleted of *E. coli* but had a high proportion of Gram negative bacteria
 overall, while the LPS extract only triggered a minimal immune response in some of
 the MoDCs from adult donors. Collectively, these observations indicate that the
 composition of the isolated LPS fractions has an important role in their activity as
 well.

As Figure 4 was based on samples for which no metagenomic data was available at
 that time, the proportion of *E. coli* was derived from the 16S rRNA gene amplicon
 sequencing data and not metagenomic sequencing data. We have now performed
 additional qPCR analyses for assessing the actual proportions of *E. coli* per sample,
 which are included in the Figure 4 of the revised manuscript. According to the
 additional data, we have 7 faecal samples from VD neonates collected on day 3 for
 which 5 out of 7 include an increased proportion of *E. coli* (resolved based on mOTU
 analysis as well). Therefore, the presence of *E. coli* in samples collected from VD
 neonates are unlikely to skew the results but represent a common trait in the gut
 microbiome of VD neonates at day 3. These results have been included and discussed
 in the revised manuscript.

**Figure 4 | Cytokine measurements in monocyte-derived dendritic cells after**
 **stimulation with isolated LPS from neonatal stool and in neonatal plasma. a,**

Lipopolysaccharide (LPS) was isolated from faecal samples collected on day 3
postpartum from neonates in vaginal delivery (VD), caesarean-section delivery (CSD)
and CSD with small for gestational age (SGA) status (CSD+SGA) groups, and
incubated for 24 h with human monocyte-derived dendritic cells (MoDCs) isolated
from a total of 12 adult donors. Exact numbers of donors used per sample are given in
the plot. Positive control: isolated LPS from *E. coli* overnight culture. Neonates C115
and C116 are twins. **b**, Plasma levels of TNF- α and IL-18 in samples collected at day
3 after birth from VD and CSD (\pm SGA) neonates. Comparison by Wilcoxon rank-
sum test with multiple testing adjustment; *false discovery rate (FDR)-adjusted P
<0.05 and **(FDR)-adjusted $P <0.01$. Circles correspond to neonates with
metagenomic data, crosses represent neonates without metagenomic data.

**2.9.** How was the LPS concentration and purity determined for each LPS sample that
was extracted from faeces determined? The effects that they saw may have to do
largely with purity of the LPS as opposed to structural differences / immunogenicity
of the LPS. I would be more convinced of the argument they are trying to make if
each of the LPS samples had been subjected to mass spectrometry to evaluate purity,
for example. I simply worry that the LPS extraction procedure is, in itself, very
flawed and subject to error. And with such a small n , the results may not be terribly
robust, even if intriguing.

**Response:**

We appreciate the reviewer's concern regarding the purity of the extracted LPS
fractions. As already partially discussed in response to **comment 2.6.**,
chromatography followed by mass spectrometric analysis could not be performed as
high amounts of LPS are needed which we could not obtain because neonatal faecal
samples are low biomass. Additionally, such analyses would not provide any
conclusive qualitative information and accurate quantification via chromatography
coupled to mass spectrometry would also not be possible. Nevertheless, to address the
reviewer's concerns, we have performed additional quantitative and qualitative
characterizations of the LPS fractions (including from samples from additional
neonates) using the assays detailed below. In accordance to the reviewers' comment
and **comment 3.4.**, we have included the results of additional analyses in the
manuscript.

The concentration of LPS was determined using a state-of-art ELISA-based endotoxin
detection assay (Endolisa; # 609033, Hyglos GmbH, Germany). Detailed qualitative
assessments of the LPS fractions were performed using standard agarose gel
electrophoresis (to exclude the presence of DNA contamination) combined with
Coomassie (to exclude protein contamination) as well as silver staining (to visualize
LPS). To assess the immunogenicity as well as the purity of the LPS fractions, we
have included the results of additional analyses in the revised manuscript including:
(i) stimulation of primary dendritic cells with additional purified LPS, and (ii)

stimulation of specific HEK blue reporter cell lines overexpressing, either hTLR4
(#HKB-HTLR4, InvivoGen), hTLR2 (#HKB-HTLR2, InvivoGen) or hNOD1
(#HKB-HNOD1, InvivoGen) or hNOD2 (#HKB-HNOD2, InvivoGen). The results of
these additional experiments demonstrate that: (i) the protocol for LPS extraction and
purification results in pure LPS fractions when compared to commercially available,
pure LPS; (ii) no immunological stimulation of dendritic cells was observed which
would be attributable to DNA; (iii) if LPS was used at a defined concentration for all
samples, only hTLR4 was activated and not hTLR2, nor hNOD1 or hNOD2; (iv) if
isolated LPS was used at sample-specific concentrations to mimic realistic *in vivo*
conditions, LPS mixtures were still relatively pure and comparable to commercially
available LPS (Sigma) as hNOD1 and hNOD2 were in both cases not activated. In
samples that were highly enriched in LPS also hTLR2 was activated, which was also
the case for commercially available, pure LPS. Finally, in case a minimal amount of
endotoxin unit (EU) of LPS was used to stimulate dendritic cells, the TNF- α response
did not correlate with a linearly increasing amount of both LPS and TNF- α ,
suggesting that indeed the composition of the different isolated LPS mixtures does
play a role in the immune response. Based on the immunogenicity of the purified LPS
fractions, additional factors may be at play, however the detailed elucidation thereof
goes beyond the scope of this study. In this context, we would like to stress again that
the present work represents the first study to show a difference in
immunostimulatory potential of the earliest gut microbiome between VD and CSD
neonates and during a critical window of opportunity for immune system priming.
More detailed future work will be necessary to elucidate the different molecular
factors involved in immune system priming. With regards to the reviewers'
comments, we have now included an additional supplementary note and figures on all
additional experiments that were conducted to properly assess the purity and
immunogenicity of the isolated LPS mixtures from neonatal stool samples.

[revised manuscript text omitted]

although some unknown microbial products may play a stimulatory role in the high
yield LPS fractions (1 ng of standard LPS and an average of 2.9 ng of LPS isolated
from VD neonates; Supplementary Fig. 12). Consequently, the composition of the
different isolated LPS fractions played an important role in the subsequently triggered
immune response.

Supplementary Information, Supplementary Note 5: To assess the LPS purity, we
performed detailed characterizations of the obtained LPS fractions. First, we used
agarose gel electrophoresis to exclude the presence of DNA contamination
(Supplementary Fig. 11a) with the subsequent extraction of DNA from excised
agarose bands and measurement of TNF- α in the supernatant of MoDCs upon
stimulation (Supplementary Fig. 11b). Our results confirmed that DNA contamination
did not considerably contribute to the observed immune activation of MoDCs.
Second, we used polyacrylamide gel electrophoresis combined with Coomassie
staining in order to successfully exclude the presence of proteins (Supplementary Fig.
11c). Third, we visualized LPS using polyacrylamide gel electrophoresis followed by
silver staining (Supplementary Fig. 11d).

To further assess the immunogenicity as well as the purity of the LPS fractions, we
performed additional stimulation assays using specific HEK-Blue™ reporter cell lines
which overexpressed either hTLR4, hTLR2, hNOD1 or hNOD2 (Supplementary Fig.
12a to d). In summary, when LPS was used at a defined concentration for all samples,
only hTLR4 was activated and not hTLR2, nor hNOD1 or hNOD2; if isolated LPS
was used at sample-specific concentrations to mimic realistic *in vivo* conditions, LPS
mixtures were still relatively pure and comparable to commercially available LPS
(Sigma) as hNOD1 and hNOD2 were in both cases not activated. In samples that were
highly enriched in LPS also hTLR2 was activated, which was also the case for the
commercially available, pure LPS. Finally, in case a minimal amount of endotoxin
unit (EU) of LPS was used to stimulate dendritic cells, the TNF- α response did not
correlate with a linearly increasing amount of both LPS and TNF- α , suggesting that
indeed the composition of the different isolated LPS mixtures does play the dominant
role in the observed immune responses.

**Supplementary Figure 11 | Purity assessment of isolated LPS fractions from**
 **neonatal stool.** **a**, Agarose gel electrophoresis with Ethidium bromide staining to
 visualize the presence of DNA contamination in LPS fractions that were isolated from
 neonatal faecal samples. Agarose bands at the highlighted sizes were cut and DNA
 was isolated. **b**, Stimulation of human monocyte-derived dendritic cells (MoDCs)
 from adult donor 9 with extracted DNA samples obtained from (a). Immune reaction
 was measured by levels of TNF- α in the supernatant. Untreated MoDCs were used as
 negative control, while MoDC stimulation with 15 endotoxin units (EU) of LPS
 isolated from an overnight culture of *Escherichia coli* strain K-12 (sub-strain
 MG1655) were used as positive control. **c**, Poly-acrylamide gel electrophoresis
 combined with Coomassie staining to visualize protein contamination. **d**, Poly-
 acrylamide gel electrophoresis combined with silver staining to visualize LPS
 presence.

**Supplementary Figure 12 | Immune stimulation assays of reporter cell lines after**
 **stimulation with isolated LPS from neonatal stool.** Read-outs of HEK-Blue cell
 lines overexpressing respectively one of the membrane receptors hTLR4 (a), hTLR2
 (b), NOD1 (c) or NOD2 (d) were stimulated with LPS fractions isolated from
 neonatal faecal samples. a, The hTLR4 receptor only recognizes LPS. Positive
 control: ultrapure LPS (5 µg). b, The hTLR2 receptor recognizes peptidoglycan,
 lipoteichoic acid and lipoprotein from gram-positive bacteria, lipoarabinomannan
 from mycobacteria, and zymosan from yeast cell wall. Positive control: Pam3CSK4
 (1 µg). c, The NOD1 receptor binds to bacterial molecules containing the D-glutamyl-
 meso-diaminopimelic acid (iE-DAP) moiety. Positive control: TriDAP (10 µg). d,
 The NOD2 receptor recognizes bacterial molecules (peptidoglycans) and stimulates
 an immune reaction. Positive control: Murabutide (10 µg). OD: optical density.
 Technical duplicates were done per condition and error bars reflect the standard
 deviation between duplicates.

**Minor comments**

We thank the reviewer for the valuable input. We address the important minor
 comments in the following sections but all the reviewer's minor comments have been
 comprehensively addressed in the revised version of the manuscript.

**2.10.** Not enough detail in the manuscript - how were differentially abundant

functional categories from KEGG identified? I appreciate the strict space limitations,
but some information regarding method should be included in the manuscript.

**Response:**

We appreciate the reviewer's concern over the apparent lack of methodological
details outside the material and methods section of the manuscript. Differentially
abundant functional categories from KEGG were identified using the Deseq2 package
using the R statistical package, while differentially abundant pathways were detected
through pathway enrichment analysis using a custom R script according to
doi:10.1038/nmicrobiol.2016.180. Due to previous space limitations, we refrained
from adding too many details on the methods into the main text of the manuscript.
However, we have now included as much methodological information as needed in
the revised main text.

**2.11.** The authors refer to how the “early microbial functions in VD neonates reflected
the mothers’ functional microbiome profiles” on line 86-87, but they have not yet
introduced any analysis of the mother’s microbiome. This is confusing to me. Also,
are they looking at the maternal vaginal microbiome or the maternal stool microbiome
(both of which have been correlated with the neonatal microbiome). Did they look at
the neonatal faecal microbiome and the maternal skin microbiome to compare
functional potential? This would be suggested by Dominguez-Bello’s paper from
PNAS 2010.

**Response:**

The analysis of maternal gut microbiome was first mentioned in the introductory
paragraph, but we agree with the reviewer that an additional mention of the different
maternal samples that were analyzed should be added earlier in the main text. In our
study, we assessed the maternal vaginal and stool microbiome through metagenomic
sequencing, however the earliest gut functional potential of VD neonates resembled
significantly more the functional potential of the maternal gut than vaginal
microbiome, which is why Figure 2 only includes the maternal gut microbiome data.
We have referred to this this more precisely in the revised manuscript.

Results, manuscript lines 206 to 224: To assess whether the apparent taxonomic
differences between the gut microbiomes of VD and CSD neonates are reflected at the
level of functional potential, we used the metagenomic sequencing data to compare
functional profiles of all neonates to the gut microbial potential of their respective
mothers. We also compared the CSD (\pm SGA) microbiota at day 3 and day 5
postpartum to those of VD neonates. The correlations of the functional profiles of the
neonatal gut microbiome to the respective maternal vaginal microbiome were lower
(median rho 0.06 for day 1, 0.37 for day 3 and 0.44 for day 5) than the correlation
between neonatal and maternal gut microbiomes (median rho 0.29 for day 1, 0.59 for
939 day 3 and 0.62 for day 5). While the correlations of the functional profiles of the

940 neonatal gut microbiome and the maternal vaginal microbiome did not differ
significantly between delivery modes, early microbial functions in VD neonates better
reflected the mothers' functional gut microbiome profiles compared to CSD (\pm SGA)
neonates (Fig. 2a; Wilcoxon rank-sum test, FDR-adjusted $P = 4.1 \times 10^{-3}$ for day 3).
CSD (\pm SGA) neonates lacked most functions at day 3 compared to VD neonates
(Supplementary Fig. 4-9), while some appeared at day 5 (Fig. 2a). Notably, neonatal-
maternal correlations between community-wide functional potentials of the gut
microbiomes at days 1, 3 and 5 postpartum were higher for VD than for CSD (\pm
SGA) (Fig. 2b; Wilcoxon rank-sum test, FDR-adjusted $P = 6.0 \times 10^{-3}$ for day 3 and P
$= 1.8 \times 10^{-2}$ for day 5).

As the composition and diversity of the skin microbiome is highly dependent on body
site, it would be improbable to get a complete assessment of skin-to-neonate strain
transfer, which is why we focused on the two microbially rich body sites that are
known for taking an important role in neonatal gut colonization (i.e. maternal gut and
vaginal microbiomes). With regards to the reviewer's comment, we have included our
reasoning in the revised manuscript.

Results, manuscript lines 136 to 139: For each mother-neonate pair, we sampled
microbiomes of maternal body sites, which are indicated to be important in relation to
neonatal gut colonization (collection of stool and vaginal swabs; Methods) less than
24 h before delivery.

**2.12.** The methods applied in Supplementary note 3 (to determine if birth method
affected the microbiome) are not adequately explained and do not appear to be
statistically driven. What is presented is not adequate to conclude that "the different
feeding regimes of neonates did not explain the functional differences ..." on lines
94-95.

**Response:**

We thank the reviewer for the valuable comment. Although it may not be completely
excluded that the milk feeding regimen had an effect on the neonatal microbiome at
972 day 5 after birth, according to the multivariate statistical analyses, delivery mode was
973 the main factor at the origin of observed effects at day 3 after birth. These additional
974 results are discussed in the revised manuscript.

Supplementary Information, Supplementary Note 4: Feeding regime was found to be
a potential contributing factor of the relative abundances of *Trichococcus* (formula
feeding and mixed feeding regime; $Q = 3.6 \times 10^{-3}$ and $Q = 4.6 \times 10^{-2}$), *Escherichia-*
*Shigella* (mixed feeding regime; $Q = 1.3 \times 10^{-2}$) and one OTU belonging to *Rothia*
(formula feeding regime; $Q = 2.9 \times 10^{-4}$).

**2.13.** It seemed like the LPS biosynthesis gene was "cherry-picked" from a list of

significantly differentially abundant genes. Also, what as the FDR cutoff? Why was
this gene chosen? What were the most significantly differentially abundant genes?

**Response:**

The enriched microbial pathway ‘LPS biosynthesis’ comprises several genes, which
are statistically significantly different between CSD and VD. The pathway of LPS
biosynthesis was specifically chosen for further validation because (i) this pathway
harbors the potential to be closely involved in the earliest priming of the neonatal
immune system during the crucial window of opportunity in early neonatal life and
early exposure to LPS may have persisting effects on the later health status. (ii)
Established methods exist for isolating LPS and performing informative experiments.
In the revised manuscript, we have included a supplementary note on the other
differentially abundant functional pathways.

Results, manuscript lines 240 to 250: Other important microbial metabolic pathways,
which were enriched with differentially abundant genes between VD and CSD,
included flagellar assembly (Fig. 2a; hypergeometric test, FDR-adjusted $P = 4.9 \times$
10^{-12}), bacterial chemotaxis (Fig. 2a; hypergeometric test, FDR-adjusted $P = 1.5 \times$
10^{-2}), cationic antimicrobial peptide (CAMP) resistance (Fig. 2a; hypergeometric test,
FDR-adjusted $P = 4.0 \times 10^{-3}$), two-component system (Fig. 2a; hypergeometric test,
FDR-adjusted $P = 2.5 \times 10^{-5}$) and ABC transporters (Fig. 2a; hypergeometric test,
FDR-adjusted $P = 1.3 \times 10^{-4}$). Notably, all pathways also showed higher relative gene
abundances in VD compared to CSD (\pm SGA) neonates except for the ABC
transporter pathway (Fig. 2c; Wilcoxon rank-sum test, FDR-adjusted $P = 4.1 \times 10^{-3}$,
3.8×10^{-2} , 2.2×10^{-4} , 2.1×10^{-2} , respectively).

Discussion, manuscript lines 544 to 573: Apart from LPS biosynthesis, other
pathways that were significantly enriched in the gut microbiome of VD neonates
included genes involved in membrane transport, i.e. ATP-binding cassette (ABC)
transporters. On the one hand this may reflect the adaptation of the colonizing
microbiome of VD neonates to the gut environment through enhanced nutrient intake.
On the other hand, associated ABC transporter proteins for both Gram-positive and
Gram-negative bacteria were previously shown to be immunogenic⁵¹, which could
suggest an implication in the activation of the neonatal immune system. Additionally,
enrichments in pathways relating to bacterial motility were observed. This included
the two-component system pathway, which is an important mediator of signal
transduction, flagellar assembly and bacterial chemotaxis. More specifically, these
pathways are essential for bacterial motility in response to external stimuli and
consequently competition with other members of the gut microbiome⁵². Additionally,
flagellin, the main structural component of the flagellum, is an effective stimulator of
innate immunity⁵³ and promotes mucosal immunity through the activation of TLR5
(ref 54). Another functional pathway that is potentially interacting with the human
immune system early on is resistance to cationic antimicrobial peptides (CAMP).
While resistance to antimicrobial peptides has been found in all major commensal

phyla of the human gut and across all members of the phylum Bacteroidetes, this
pathway is essential to evade detection by the human immune system through the
modification of the microbial LPS structure⁵⁵. In the context of our study, an
enrichment in CAMP resistance may prevent the predominantly colonizing gut
bacteria (i.e. Bacteroidetes) from being recognized by the immune system and
subsequently removed from the VD neonatal gut. Future studies are needed to assess
whether the gut microbiome of VD neonates harbours more modified LPS moieties
linked to CAMP resistance and what the subsequent immunostimulatory effects of
altered LPS structures may be on the neonatal immune system. In accordance with the
observation of an apparent enrichment in flagellar biosynthesis, bacterial chemotaxis,
CAMP resistance, other microbiota-derived molecular factors, apart from LPS, may
be involved in immune system priming.

The FDR cut-off was set to 0.05. We have additionally included the statistics per KO
in the supplementary data of the revised manuscript (Supplementary Data 10).

**2.14.** GAG and other glycan metabolism is mentioned in the abstract but is only
addressed in the supplement. If this is an important point, it should be included in the
main manuscript. If not in the main manuscript, it should be left out of the abstract.

**Response:**

We appreciate the reviewer's observation and agree that the section on GAG and
other glycans was referred to in the abstract but not discussed in the main text due to
the space limitations for the original format (*Nature* Letter format). Following the
analysis of additional metagenomic sequencing data, we now find additional
microbial pathways that are enriched in VD neonates compared to CSD and that could
also be highly relevant for the neonatal host organism. We have revised the abstract as
well as reported and discussed these pathways in the revised manuscript. For more
information, please also refer to the reply to **comment 2.13.** above.

**2.15.** When cytokine levels were obtained from 31 neonates, it seems inappropriate to
label the 14 additional infants as an "independent validation cohort" if the results are
aggregated with the discovery cohort. This is misleading to me.

**Response:**

We agree with the reviewer and have changed this accordingly in the revised
manuscript.

**2.16.** The reference citation format appears to change at line 404. This is a small
typographical issue.

**Response:**

To avoid any confusion when adding a reference number to a number in the text (as
for software version numbers), we decided to use the chosen style. However, in order
to stick to a uniform reference style, we have changed the citation style throughout the
revised manuscript.

**Reviewer #3**

**Overall summary**

**3.1.** In this interesting research paper by Wampach et al, authors used a very robust
bioinformatics approach to track specific bacterial strains from mothers to neonates
during the perinatal period, taking into account levels of genetic diversity using well-
established molecular ecology methods. This study has numerous strengths that are
original and should be published: it combined taxonomic and functional profiling of
the microbiome using reference independent metagenomics analysis, included
controls and a bioinformatic approach that accounted for reagent and host derived
contaminant DNA and removal of sequencing artefacts, and utilized sequencing data
to inform the experimental parameters of an immune assay, which was informative of
the potential interactions between LPS and the immune system. However, these
strengths are not the main claim of this study. This clinical study was not designed nor
was it shown to be powered to determine perinatal microbiome differences driven by
mode of birth, and its results should not be interpreted as such. Below is a list of
major and minor suggestions that should be incorporated in a new version of this
manuscript.

**Response:**

We appreciate the reviewer's assessment of the quality and importance of the work.
As discussed above and in the following comment (see responses to **comments 1.2.**
**and 3.2.**), we have included results from additional samples from additional
individuals in the revised version of the manuscript. The results from the additional
analyses support the conclusions in the original manuscript.

**Major comments**

**3.2.** The sample number in this study is simply too low and there were no power
calculations performed a priori to determine if it could identify difference between
birth modes. Given that other small studies have failed to show microbiome
differences driven by birth mode, it is hard to believe that this study was powered to
assess the influence of birth mode, let alone the influence of low birth weight. I
recommend that authors present power calculations based on previously published
data to determine this. A good resource to perform these calculations for studies with
multivariate data is MetSizeR. Clearly, the results from this study are in line with
most other studies that have convincingly shown that mode of birth drives temporal
neonatal composition and diversity. However, a recent study from the Xavier group
(DOI: 10.1126/scitranslmed.aad0917) showed that while most CS births were

associated with lower *Bacteroides* species abundance, about 20% of the VD babies
displayed a similar pattern. With an N=4 per group (including 1 set of twins in the
CSD groups) with only 2 valid time points (Day 1 only had 2 samples in 2 of the
groups, so in my opinion should be excluded for comparisons), there is simply no
statistical power to detect appropriate variance of a given population, especially
considering the interindividual variability of the gut microbiome during the first days
of human life.

**Response:**

We agree with the reviewer that the number of study participants is comparatively
low in relation to previous studies (see response to **comment 1.2.**). Nevertheless, it is
important to emphasize important unique aspects of our study: (i) paired mother-
infant samples, (ii) samples from the earliest time points, and (iii) high-resolution,
contaminant-free metagenomic data. We want to stress again that the main point of
the manuscript is not statistics-driven, as the presence of strains, given their high
degree of specificity, is valid only on an individual, sample-to-sample basis. Single
nucleotide variants are highly specific and the presence of the exact same gut strains
in mothers and their vaginally delivered neonates demonstrates that transfer of strains
happens from mothers to neonates, with vaginal delivery being the most probable
mode of vertical transfer. The fact that considerably less maternal enteric strains were
found in neonates that were delivered by C-section, provides evidence that delivery
through C-section impedes this vertical transfer. Our high-resolution approach of
strain tracking in addition to statistical tests on the functional profiles (which are
statistically significant; see also our response to **comment 3.3.** below) is more robust
and to date unprecedented with regards to the earliest neonatal gut microbiome.
Nevertheless, we have included additional metagenomic sequencing data, as well as
16S rRNA gene sequencing data from additional neonates, which support our high-
resolution, strain-specific tracking results. Please also refer to **comment 1.2.** for
additional details.

With specific regards to the study from the Xavier group, we want to stress that their
conclusions on the low relative abundance levels of *Bacteroides* were (i) based on a
disproportionate distribution of study participants regarding delivery mode (4 C-
section delivered against 35 vaginally delivered infants), which compromises their
statistical power, (ii) were based on 16S rRNA gene amplicon sequencing, (iii) did
not include paired mother-neonate pairs, and (iv) did not focus on earliest time points
after birth. In our case, as the tracking of highly specific strains from mother to
neonate is considered on a case-by-case basis, our main conclusions are valid in light
of inter-individual variability of the gut microbiome during the first days of human
life.

**3.3.** Although differences between groups were detected using appropriate statistical
tests, without confidence intervals and proper statistical tests to deconfound the effect

of birth mode from other variables known to influence microbiome composition, it is
not possible to properly assess whether birth mode truly drives these differences.
Given that authors have access to 20 more samples (6 for which they had 16S data
and 14 from a previous cohort that was collected in a similar way), the authors could
use the 16S data and PICRUSt, at a minimum, to determine if other variables are also
influencing the associations found between birth mode and microbiome composition.
An excellent tool for this purpose is MsAsLin
(<https://huttenhower.sph.harvard.edu/maaslin>). Variables to include in this model are
gestational age, antibiotic use, dose, duration, time of meconium passage, if babies
ingested colostrum, etc. In addition, authors should demonstrate that the samples used
for metagenomics analysis are representative of the larger group of samples. For this,
they could compare microbiome characteristics (beta or alpha diversity) as well as
clinical variables (mode of birth, gestational age, etc.)

**Response:**

We thank the reviewer for this constructive comment. In accordance with the
reviewer's suggestion, we have included additional 16S rRNA gene amplicon
sequencing data in the revised manuscript (see also our response to **comment 1.2.**).
Furthermore, in accordance with the reviewer's suggestion to determine whether other
variables are potentially influencing the associations found between delivery mode
and microbiome composition, we have indeed used the tool MaAsLin by including
variables such as delivery mode, feeding regime, gestational age and maternal
antibiotics intake. The major trends that were highlighted in the initial manuscript
were thereby still explained by delivery mode [i.e. higher relative abundance of
*Staphylococcus* or lower levels in *Bacteroides* in CSD (\pm SGA) neonates]. Please also
refer to our reply to **comment 2.1.**

In accordance with the reviewer's suggestion to infer functional profiles based on the
extensive 16S rRNA gene amplicon data, we also used the tool PanFP
(<https://github.com/srjun/PanFP>) and observed similar trends to the pathways that we
detected based on metagenomic sequencing. In order to predict the functional profiles
of microbial communities based on our 16S rRNA gene amplicon data, we have
included additional analyses in the revised manuscript that are based on the tool
PanFP (doi:10.1186/s13104-015-1462-8). While PICRUSt presumes a closed-
reference OTU picking strategy, which results in a strong dependency on the
completeness of the reference database, PanFP is highly compatible with the open-
reference strategy of NG-Tax (doi:10.12688/f1000research.9227.1) that we used for
processing the 16S rRNA gene amplicon data. An additional paragraph on the
outcome of these analyses has been included in the revised manuscript.

Results, manuscript lines 258 to 276: Additionally, the relative abundances of 7,000
KO functional categories were predicted using PanFP³⁷ based on the extensive 16S
rRNA gene amplicon data (Supplementary Data 11). A multivariate analysis
(MaAsLin³⁵) was performed to compare the functional profiles of CSD (\pm SGA) to

VD neonates and for both generated datasets (i.e. predicted KO functional categories
based on 16S rRNA gene amplicon sequencing data and annotated KOs based on
metagenomic sequencing data). Results from the multivariate analyses demonstrated
that delivery mode was the strongest determining factor in both datasets (i.e. predicted
and metagenomic-based KOs) for explaining the differentially abundant genes
(Supplementary Data 9). Whilst not statistically significant, the trends for the
predicted microbial pathways obtained with PanFP were largely concordant with the
enriched pathways in VD neonates based on the differential analysis of the
metagenomic data. Nevertheless, predictions of functional potentials based on 16S
rRNA gene amplicon sequencing data are likely unreliable as a significant fraction of
the gut microbiome (i.e. up to 40%) is represented by microorganisms without a
sequenced isolate genome³⁸. In contrast, the metagenomic data, through resolving the
actual functional gene complement, allows a detailed comparison of the functional
potential of the earliest gut microbiomes, as well as the tracking of individual-specific
single-nucleotide variants (SNVs).

In order to assess whether the samples used for metagenomic analysis were
representative of the larger group of samples in the initial manuscript, we included a
supplementary note and supplementary figures on specific microbiome
characteristics, including beta and alpha diversity, in the original manuscript.
According to this information, the presented metagenomic data is representative of the
larger group of samples.

Results, manuscript lines 186 to 194: The 16S rRNA gene amplicon and the
metagenomic sequencing data, which was generated for a subset of mother-neonate
pairs, showed highly similar succession trends in terms of diversity, evenness and
richness measures (Supplementary Fig. 2a&b; Supplementary Note 2). The taxonomic
profiles derived from the 16S rRNA gene amplicon and metagenomic sequencing
were highly correlated (Supplementary Fig. 3a). The differences in taxonomic profiles
according to delivery mode reflected results from previous studies, notably the higher
relative abundance in *Bacteroides* and *Parabacteroides* and lower levels in
*Staphylococcus* in VD neonates at days 3 and 5 postpartum^{7,10} (Supplementary Data 6
to 8; Supplementary Note 3).

Supplementary Information, Supplementary Note 3: To identify differences between
the birth modes (and SGA status) and the different collection time points postpartum,
we performed Wilcoxon rank-sum tests (without multiple-alignment adjustments) on
the sum-normalized relative abundances of metagenomic and 16S rRNA gene
amplicon sequencing data (Supplementary Data 7 and 8). Although several
statistically significant differences were found, we only considered genera and OTUs
that showed the same trends for both CSD and CSD+SGA neonates and had a P-
value below 0.05. In VD neonates, compared to both CSD and CSD+SGA, the
metagenomic data resolved a higher relative abundance of the species *Bacteroides*
*dorei/vulgatus* (day 5 VD vs CSD, $P = 2.5 \times 10^{-2}$ and VD vs CSD+SGA, $P = 1.2 \times$

10^{-2}), as well as the specific *Bacteroides dorei/vulgatus* mOTU (day 5 VD vs CSD, P
$= 1.4 \times 10^{-2}$ and VD vs CSD+SGA, $P = 6.7 \times 10^{-3}$). Based on the 16S rRNA gene
amplicon sequencing data, significantly higher relative abundances of the genus
*Bacteroides* were observed in VD neonates at days 3 and 5 (day 3, $P = 7.8 \times 10^{-3}$ and
$P = 1.2 \times 10^{-2}$; day 5, $P = 1.2 \times 10^{-3}$ and $P = 5.1 \times 10^{-3}$, respectively). The genus
*Parabacteroides* was significantly increased ($P = 4.2 \times 10^{-3}$ and $P = 2.8 \times 10^{-2}$),
while *Rothia* was found to be significantly decreased in VD neonates ($P = 1.8 \times 10^{-2}$
and $P = 2.2 \times 10^{-4}$). A total of 10 OTUs assigned to the genus *Staphylococcus* were
significantly increased in CSD±SGA neonates, while OTU 243 assigned to
*Bacteroides* was significantly increased in VD neonates at days 3 and 5 (day 3, $P =$
2.3×10^{-2} and $P = 2.5 \times 10^{-2}$; day 5, $P = 2.5 \times 10^{-3}$ and $P = 9.4 \times 10^{-3}$, respectively).

**3.4.** The results from the immune assay with extracted LPS from different neonatal
samples are quite compelling but remain rather preliminary, and could be enhanced in
several ways. As it stands, the purity of the LPS was not assessed, it is vaguely
assumed that the majority of the cytokine response measured originated from *E. coli*
LPS, and although it is mentioned in the text, it is not known if the differences in
cytokine response are due to differences in LPS structure. Addressing at least 2 of
these issues will significantly strengthen this paper. LPS purity can be determined by
gel electrophoresis (comparing silver, protein and DNA staining, for example), or if
available, mass spectrometry analysis. Differences in LPS structure (rough vs smooth,
for example) can also be obtained through these methods. LPS purity can also be
confirmed functionally by showing that the purified products did not activate Toll-like
receptor 2 (TLR2), nuclear oligomerization domain 1 (NOD1), or NOD2 but did
activate TLR4.
Importantly, if the authors have more samples from other cohorts available, why not
increase the number of samples from this assay?
As for determining the origin of the LPS, could LPS genes not be assigned to the
mOTUs given that the information is available? This is important as it will inform
which taxa may or may not be relevant in stimulating immune cells during the first
1274 days of life.

**Response:**

We appreciate the reviewer's recognition of the compelling nature of the
immunological data presented and thank him/her for the suggested additional
experiments to strengthen the study. To address the reviewer's comments, we have
now conducted additional experiments including additional LPS extractions from
additional samples from early neonates. More specifically, in the revised manuscript
we have included results reflecting the purity of the isolated LPS and on the
specificity of immunological responses. Please also refer to our detailed response to
**comment 2.9.** above. Bearing in mind these additional experiments, we have
addressed both suggestions of the reviewer regarding assessment of LPS purity and
immunogenicity. The results of these additional analyses underscore our earlier

results and thereby further strengthen our original conclusions.

For determining the origin of LPS, we annotated the OTUs resulting from the 16S
rRNA gene sequencing data according to their Gram staining characteristics using the
recorded microbial attributes from NCBI ([http://www-ab2.informatik.uni-
tuebingen.de/megan/taxonomy/microbialattributes.zip](http://www-ab2.informatik.uni-tuebingen.de/megan/taxonomy/microbialattributes.zip)). More precisely, the Gram
staining information of microorganisms directly corresponds to their propensity of
producing LPS, therefore the direct assignment of OTUs being either Gram positive
or negative is robust, while identifying LPS gene functions in the complement of
reconstructed genomes provides comprehensive results only for the most dominant
taxa, i.e. those with almost complete genome reconstructions. On a genome-scale
level, we identified which taxa may be relevant in stimulating immune cells during
the first days of life in the initial manuscript and this information was represented by
the colored spokes in the Circos plots in Figure 3. Based on these observations, we
suggested that strains that were transferred from mother to VD but not CSD neonate
were significantly enriched in genes that are involved in LPS biosynthesis. To clarify
these points, we have included additional text passages in the revised manuscript.
Please refer to the cited text passages given under **comment 1.2.** and **2.6.**

**3.5.** Extended data figure 2. Change color palette. Too many tones of blues and
greens. Don't do color coordination according to phyla.

**Response:**

We thank the reviewer for this constructive comment. We have tried several color
combinations for this supplementary figure but concluded that the current color
palette based on phyla assignment was the most informative for the visual assessment
of the neonatal gut microbiome composition. For a more detailed evaluation, the
complete mOTU table obtained by metagenomic sequencing is provided in the
supplementary data (Supplementary Data 3).

**Minor comments**

We thank the reviewer for their valuable input. We comprehensively addressed all the
reviewer's minor comments in the revised version of the manuscript.

**3.6.** Blinding in Life Sciences reporting summary: "As assigned study groups were
predefined prior to delivery, blinding was irrelevant to our study." This is false. Many
studies with predefined study groups are blinded and this adds strength to the study
design. This study was simply non-blinded, please change.

**Response:**

We thank the reviewer and agree with this comment. We have made a corresponding
change to the reporting summary.

**3.7. FIGURE 2:** It is unclear how the DESEQ2 generated results are displayed in
Figure 2. Deseq2 compares 2 groups. According to Figure 2a it looks that the
comparison was made between VD samples and all CSD samples combined but figure
legend and methods section describes two separate comparisons. Were the same 5
pathways differential in both comparisons or only when VD samples were compared
to all CS babies?

**Response:**

We thank the reviewer for this relevant comment. Indeed, as DESeq2 compares two
groups, we made comparisons based on VD vs CSD and VD vs CSD+SGA.
Subsequently, we combined all significantly differential KOs from both tests for
which the FDR-adjusted P value of the Wald test was <0.05 for at least one
comparison and for which the directionality of change in both comparisons was the
same. We have made corresponding changes to the revised manuscript.

Material and methods, manuscript lines 836 to 844: Differential analysis of KO
abundance, comparing VD to CSD and VD to CSD+SGA with a linear model, which
considered the different collection time points containing at least 1,000 KOs (days 3
and 5) as covariates, was performed with the R package ‘DESeq2’ version 1.10.1³⁶.
KOs were considered significantly differentially abundant in VD and CSD (\pm SGA) if
the FDR-adjusted P value of the Wald test was <0.05 for at least one comparison
(CSD versus VD or CSD+SGA versus VD) and the directionality of change in both
comparisons was the same. Differentially abundant pathways were detected through
pathway enrichment analysis using a custom R script⁵⁹.

Results, manuscript lines 226 to 232: We detected a total of 1,697 functional
categories from the Kyoto Encyclopedia of Genes and Genomes (KEGG) Orthology
(KO) database that were differentially abundant in the combined comparisons of the
gut microbiome of CSD and CSD+SGA neonates to VD neonates and were
presenting the same directionality of log fold change. We used the R package
‘DESeq2’³⁶ with a linear model, which considered the different collection time points
containing at least 1,000 KOs (days 3 and 5) as covariates (Fig. 2a; Supplementary
Data 10).

Figure legend, manuscript lines 1320 to 1332: Heatmap of relative abundance of gut
microbial orthologous gene groups with significant differential abundances in
neonates born by vaginal delivery (VD) compared to either caesarean-section delivery
(CSD) or CSD with small for gestational age (SGA) status (CSD+SGA) groups and
having the same direction of log₂ fold change (calculated with the R package
‘DESeq2’³⁶ and indicated by the color key; false-discovery-rate (FDR)-adjusted P
<0.05).

**3.8. FIGURE 4:** The figure legend does not fully explain 4a. Specifically, it was only

after I read the main text that I understood that the EU amounts were measured in the
immune assay. I was confused and initially thought it was the LPS concentration used
in the assay.

**Response:**

We thank the reviewer for pointing this out. In fact, the endotoxin units (EU) given in
Figure 4a (now **Supplementary Fig. 13** in the revised manuscript) were estimated
based on the quantification of the LPS samples prior to the immunostimulatory assay.
No LPS amounts were measured in the immune assay. We have revised the
manuscript and figure legends to make this clearer.

Figure legend, manuscript lines 1367 to 1380: Lipopolysaccharide (LPS) was isolated
from faecal samples collected on day 3 postpartum from neonates in groups of vaginal
delivery (VD), caesarean-section delivery (CSD) and CSD with small for gestational
age (SGA) status (CSD+SGA), and incubated for 24 h with human monocyte-derived
dendritic cells (MoDCs) isolated from a total of 12 adult donors. MoDCs were
stimulated the with the exact same LPS volume that was extractable from the same
initial amount of faecal material from each neonate sample (Methods).

**3.9.** Line 87, vaginal or stool in mothers?

**Response:**

We thank the reviewer for pointing this out. We clarified this part in the revised
manuscript.

Results, manuscript lines 206 to 219: To assess whether the apparent taxonomic
differences between the gut microbiomes of VD and CSD neonates are reflected at the
level of functional potential, we used the metagenomic sequencing data to compare
functional profiles of all neonates to the gut microbial potential of their respective
mothers. We also compared the CSD (\pm SGA) microbiota at day 3 and day 5
postpartum to those of VD neonates. The correlations of the functional profiles of the
neonatal gut microbiome to the respective maternal vaginal microbiome were lower
(median rho 0.06 for day 1, 0.37 for day 3 and 0.44 for day 5) than the correlation
between neonatal and maternal gut microbiomes (median rho 0.29 for day 1, 0.59 for
1407 day 3 and 0.62 for day 5). While the correlations of the functional profiles of the
1408 neonatal gut microbiome and the maternal vaginal microbiome did not differ
significantly between delivery modes, early microbial functions in VD neonates better
reflected the mothers' functional gut microbiome profiles compared to CSD (\pm SGA)
neonates (Fig. 2a; Wilcoxon rank-sum test, FDR-adjusted $P = 4.1 \times 10^{-3}$ for day 3).

**3.10.** Line 94-96 briefly mention what statistical method was used (if any) to rule out
the influence of feeding method

**Response:**
We thank the reviewer for this comment. In the revised manuscript, we have included
a detailed multivariate analysis using MaAsLin. For more details, please refer to our
reply to **comments 2.1.** and **2.2.**

**3.11.** Lines 99-105 Where there any comparisons done between VD samples and each
of the CSD groups. If so, this should be mentioned in this section. If not why separate
the CSD groups?

**Response:**
We thank the reviewer for this comment. Indeed, comparisons between VD and CSD
and VD and CSD+SGA were done and were largely matching independent of the
SGA status. As no discernible differences were observed based on Fig. 4a between
CSD and CSD+SGA, we chose to combine both groups in order to increase the
statistical power.

Results, manuscript lines 237 to 240: In order to increase the statistical power and as
comparisons between VD and CSD and VD and CSD+SGA were largely matching
independent of additional SGA status, we combined both groups (CSD and
CSD+SGA; Fig. 2b and c).

**3.12.** Line 144, sources of microbial origin are too speculative, I suggest removing

**Response:**
We thank the reviewer for the suggestion. We have revised the manuscript and cited
previous studies that observed or suggested strain transfer from other environments to
the neonate.

Discussion, manuscript lines 485 to 487: Consequently, the gut of CSD neonates is
most likely colonized by strains derived from other sources, such as breast milk, skin
or saliva, as suggested in previous studies⁴⁴⁻⁴⁶.

**3.13.** Line 174 I suggest changing “more specifically” for “especially”

**Response:**
We thank the reviewer for the suggestion and we have revised the manuscript
accordingly.

Results, manuscript lines 401 to 404: The levels of all measured cytokines, and
especially of TNF- α and IL-18, were higher in culture supernatants from MoDCs
treated with LPS from VD neonates (Supplementary Figure 13; Supplementary Data
16; Supplementary Note 6).

**3.14.** Line 202-203 Huge stretch here. There is no data on chronic diseases in this
super small cohort. Also omit from the summary figure in extended data.

**Response:**

We thank the reviewer for pointing this out. We agree that the previous statement was
too suggestive in the given context. We have now included a separate paragraph based
on previous studies, which support our results. We have also removed the summary
figure.

Discussion, manuscript lines 522 to 542: Our study highlights differences in
immunostimulatory potential of the earliest gut microbiome according to delivery
mode. This occurs during a critical window of immune system priming. Notably,
alterations to early immune system stimulation may be linked to the higher propensity
of CSD infants to develop chronic diseases in later life². For example, previous
studies focusing on environmental exposure in early life have suggested that the
exposure to Gram-negative bacteria and/or environmental endotoxins (such as LPS)
could confer protective effects towards allergy development^{48,49}. In this context, the
LPS biosynthesis pathway harbours the potential to be closely involved in the priming
of the neonatal immune system and the subsequent tolerance towards the colonizing
gut microbiome during a most critical window in early neonatal life¹²⁻¹⁴, with
potential persisting effects on the later health status. Using a mouse model, it has been
shown that strongly immunostimulatory LPS can contribute to the protection from
immune-mediated diseases such as diabetes⁵⁰ and that disruption of host-commensal
interactions in early-life can lead to persistent defects in the development of specific
immune subsets¹². Based on additional cytokine measurements in neonatal plasma,
VD neonates displayed higher levels of IL-18 and TNF- α , thereby indicating a link
between the immunostimulatory potential of microbial LPS in the gut and the overall
immune status of the neonatal host early on. Investigations of the longer-term
consequences of these differences between CSD and VD neonates will be necessary
to assess their possible impact on the development of chronic diseases in later life.

**3.6** Lines 483-487 Unclear why bacterial DNA amount needed to be controlled in the
LPS assay. Please elaborate either here on in the Supplementary notes. These assays
conditions will not mick the amount of LPS that interacts with a peripheric immune
cells so I do not understand the reasoning behind this. Why not normalize by LPS
units only? DNA quantification via qPCR of 18S RNA gene will vary depending on
the number of 16S copies per cell so it is not the best way to quantify bacterial load.
Further, this method will account for all bacterial cells, not just gram-negative (LPS
containing cells). Flawed method.

**Response:**

The rationale for using the amount of bacterial DNA was to compare the same
bacterial load for each sample and, thus, assess if a distinct sample will induce a

distinct immune response. As the bacterial loads in early meconium samples were in
general lower in CSD compared to VD, we thereby aimed to minimize the likelihood
that an observed effect is not due to a lower bacterial presence in the CSD meconium.
However, we also agree with the reviewer that the current normalization does not
reflect the real situation. Therefore, we have included additional immunological
assays in the revised manuscript in which we stimulate primary immune cells with the
exact same “LPS volume” extracted from the exact same starting faecal mass for each
individual sample. Using both methods, we observed the same effects (Fig. 4a and
Supplementary Fig. 13). With respect to additional samples that were used, as well as
the specific reporter cell lines (comments 2.6., 2.9. and 3.4.), we thus show that the
immunological stimulatory potential of neonatal faecal samples is much higher in
case of vaginal delivery compared to C-section delivery.

Results, manuscript lines 392 to 401: In order to reflect the *in vivo* situation as closely
as possible, we stimulated the MoDCs with the exact same LPS volume that was
obtainable from the same initial amount of faecal material from each neonate sample
and subsequently measured levels of the LPS-inducible cytokine TNF- α in the
supernatants using an ELISA assay (Fig. 4a; Supplementary Data 13). In parallel, a
panel of additional cytokines was measured using an approach for quantifying and
normalizing the employed LPS fractions (Methods). This was based on a maximum
stimulation of MoDCs with 100 Endotoxin Units (EU) of LPS in order to mimic the
amount of LPS an immune cell may encounter within a given neonatal sample
(Supplementary Figure 13; Supplementary Data 14 and 15).

11. Line 124 of Supplementary notes: I would change “corresponded” to correlated

**Response:**

We agree with the reviewer and have adapted the manuscript accordingly.

Supplementary Information, supplementary note 6: The observed levels of TNF- α
mostly correlated with the relative abundance of Gram-negative bacteria (Fig. 4a,
Supplementary Fig.3b).

12. The manuscript could use a brief discussion on the choice of adult blood DCs and
how it may have differed from neonate peripheral immune cells. Neonatal immune
cells are known produce immune responses similar to adults in some aspects but not
others.

**Response:**

We thank the reviewer for this constructive comment. We included details on the
choice of adult blood DCs to the revised manuscript.

Material and methods, manuscript lines 879 to 885: Human neonatal dendritic cells
(DCs) were previously shown to be competent in MHC class I antigen processing and
presentation to the same extent than adult DCs⁷⁹. Most importantly, the NF- κ B-
dependent pathway in TLR-4 signalling is intact in neonatal MoDCs as they produce
pro-inflammatory cytokines upon LPS stimulation, while adult and neonatal DCs are
both able to produce comparable levels of TNF- α , IL-6 and IL-8 in response to LPS⁸⁰.

REVIEWERS' COMMENTS:

Reviewer #1 (Remarks to the Author):

The authors performed a substantial amount of work and additional experiments for this revised manuscript. The new data is strong and all the crucial issues pointed out on the original submission seem to be fixed. One of the strongest points of the work is the increased vertical transmission of microbes in vaginally delivered infants compared to delivery via C-section.

I still have some remarks for the authors.

- The paper is convincing on the fact that vaginal deliver impacts the infant microbiome. The authors state that this was quite controversial, but I partially disagree with this. What was controversial is whether the effect on the microbiome of vaginal versus C-section delivery is long-lasting and whether it is stronger than other conditions including breast-feeding versus formula-feeding. Because in this study the infants are not followed for months and breast feeding is not discussed in depth, these questions cannot be answered. I think the authors should be a bit more careful when discussing the effect of delivery mode on the microbiome, because there are not really many controls for other conditions (again feeding regime).

- The introduction now is very comprehensive. However, it is also redundant and repeated in several passages. I feel the introduction should be shortened a bit and repeated concepts minimized. Also the discussion is very verbose.

- Is multiple hypothesis testing correction applied on the identification of 1,697 differential KEGG KOs at line 226?

- Figure 3A. It is difficult to see the vertical transmission events because of the use of a color gradient (shade of orange). More distinct colors should be adopted here, and in general the figure is not very intuitive. In the legend, it is not clear what "taxon without link" means

- line 212-220: microbiomes from different body sites of the mother and infants are here compared using correlation. Giving the large differences in the overall structure of the microbiome across body sites and hosts, I don't think correlation is the correct statistical tool to use.

Reviewer #2 (Remarks to the Author):

In this manuscript, Wampach et al collected stool samples from mothers and matched neonates. They perform shotgun sequencing and analyze the gene-level differences between the stool microbiomes of vaginally born babies vs. C-section born babies. They find that genes encoding LPS synthesis are enriched in the vaginally born babies - and that the LPS extracted from these two stool types, when tested on primary human immune cells, elicits differences in cytokine release. The LPS extracted from the Vaginally born babies appears to be more immunostimulatory. Thus, the authors conclude that C-section disrupts transfer of specific microbial strains that are highly immunostimulatory, and that this may impact "neonatal immune system priming".

Overall, the manuscript is much improved from its prior version. A notable strength of

this manuscript is the collection and evaluation of very early post-natal samples (day 0-5) and the strong effort to characterize the functional impact of LPS on the immune system. The manuscript, as it stands, is a bit lengthy (especially in the discussion section), but is well carried out and interesting. I offer the following suggestions for improvement.

Major suggestions:

#. The authors now allude to the existing controversy in evaluating the impact of birth-mode on microbial "transmission". I propose that it might be better for the authors to steer clear of the transmission question and to simply report the results in the context of vaginally born and CS born babies having different microbiomes (which may be the consequence of confounding factors other than just birth mode - e.g. antibiotic exposure).

#. The title suggests that birth mode is causal of strain-conferred gut microbiome functions and immunostimulatory potential - I think the title should be adjusted to reflect the fact that the authors identify an association, as opposed to a causal link between birth mode and these outcomes.

#. Are there any other notable differences between the VD and CSD babies? Or were these all elective CSD babies - which would be the best comparator.

#. I find the LPS aspect of the paper the most novel and exciting - but it is only briefly focused on in the manuscript. Also, is there a reason that only a subset of the total number of subjects was included in the LPS part of the study? Lastly, is there a reason the day 3 sample was selected (as opposed to the day 5 sample, which may have been higher biomass)?

Minor suggestions:

#. Would remove the phrase "After data curation" in the abstract - it is vague.

#. line 76 - I am not exactly sure what the authors are referring to when they state "Although CSD is not associated with improved health outcomes...". Also - are the a

#. Line 86 - would call this 16S rRNA gene amplicon sequencing.

#. Line 132 - for what proportion of samples could sufficient DNA be extracted for downstream metagenomic analysis?

#. The majority of the analyses seem to focus on CSD +/- SGA compared to VD; I applaud the authors for including CSD -SGA controls - but am surprised that they chose not to analyze the three groups (CSD -SGA, CSD + SGA and VD) separately, as this would be a much stronger analysis that lacked the likely confounding factors contributed by SGA.

#. Line 422, 458 - calling the approach they employed "artefact-free" seems hyperbolic.

#. line 490 - typo - "form" instead of "from"

#. Only 16 mother/neonate pairs were collected; yet, additional neonates were studied - the methods section could be more clear as to how those additional neonates were recruited to the study.

#. I presume the authors used standard V4 primers for 16S sequencing. Please state this, if so.

#. What is the "optimized low-quantity DNA library preparation kit" that was used?

#. Why were some of the metagenomic libraries (C105, 109, 110 and 119) prepared using another (presumably) kit?

#. The method for removing artifactual reads relies on contig assembly and alignment. What was the contig size cutoff used for this method? If the contig size cutoff is >read length, this might remove low abundance contaminants that are not present at a high enough abundance to assemble into contigs - thus compromising the effectiveness of this contamination removal effort.

#. line 854 - Please provide some additional detail in text regarding the "further purification" that was used to obtain the LPS.

#. Line 856 - why was LPS from the three aliquots of stool pooled - presumably because each extraction resulted in a very small amount of LPS. I would have very much liked to see this assay done in replicate (starting with the stool), as I am not familiar with the variability of LPS extraction efficiency.

#. What proportion of all metagenomic reads were assigned a potential function? I ask because it seems surprising that 2% of all metagenomic reads are classified as part of a 2-component system, for example. That seems rather high.

#. Figure 3 is very complex and does not communicate a strong, focused story. I wonder if the authors could simplify the figure (by, perhaps, moving some of the panels to the supplement)?

#. It is interesting (Fig 4) that one of the CSD+SGA neonates has a high a high proportion of Gram neg organisms and a very low amount of TNF-alpha release in the MoDC assay. What do the authors make of this? Perhaps this was in the discussion and I missed it?

Reviewer #3 (Remarks to the Author):

After reviewing this manuscript again, I am quite impressed with the amount of attention and added work that the authors have given to the reviewers' comments and suggestions. While the previous manuscript was already strong, the authors acknowledged the rightful arguments made by all reviewers and submitted a revised paper that appropriately addressed all of the major points. The authors: added samples to the study (for microbiome analysis and immune assays), controlled for confounding variables known to influence microbiome structure, confirmed purity of LPS, performed the LPS assay under alternative conditions that normalized LPS concentrations across samples, provided added evidence that samples used for metagenomics were representative of all cohort samples, and limited the discussion to their findings, without associating to future health outcomes that were not assessed in their study. Truly, the authors should be commended for their effort. The result is an outstanding contribution to the field that in my view, confirms that mode of delivery impacts the infant microbiome, both taxonomically, functionally and in relation to the immune consequence to the infant host. I would seriously question who would continue to debate on this

issue after the findings presented by Wampach et al.

One comment I would like to leave with the authors, which does not impact this paper but may be helpful in future studies of a similar nature, is in reference to the answer in their rebuttal letter

(Lines 1126-1228):

"We want to stress again that the main point of the manuscript is not statistics-driven, as the presence of strains, given their high degree of specificity, is valid only on an individual, sample-to-sample basis."

Their study argues that mode of delivery impacts infant microbiome structure and performed several analysis comparing two or more groups. All studies that compare variables of interest are subjected to statistics. Thus, all of these studies must be statistics-driven. I have learned this not as a biostatistician but after analyzing many human microbiome studies, where controlling for possibly confounding co-variates is the only valid method to link a microbiome feature with a variable of interest. There are enough studies available on the effect of antibiotics on infant microbiome, for example, to help direct future studies in terms of study design and number of samples necessary.

Manuscript reference number: NCOMMS-17-19613B

Responses to the reviewers' comments

N.B.: The original remarks by the reviewers are provided below in boxes whereas the authors' responses are listed in black type and cited passages from the manuscript are listed in blue.

Reviewer 1

1.1. The authors performed a substantial amount of work and additional experiments for this revised manuscript. The new data is strong and all the crucial issues pointed out on the original submission seem to be fixed. One of the strongest points of the work is the increased vertical transmission of microbes in vaginally delivered infants compared to delivery via C-section.

We thank the reviewer for his/her positive evaluation of the efforts that were put into improving the manuscript.

1.2. The paper is convincing on the fact that vaginal deliver impacts the infant microbiome. The authors state that this was quite controversial, but I partially disagree with this. What was controversial is whether the effect on the microbiome of vaginal versus C-section delivery is long-lasting and whether it is stronger than other conditions including breast-feeding versus formula-feeding. Because in this study the infants are not followed for months and breast feeding is not discussed in depth, these questions cannot be answered. I think the authors should be a bit more careful when discussing the effect of delivery mode on the microbiome, because there are not really many controls for other conditions (again feeding regime).

We thank the reviewer for this comment. Indeed, we agree with the reviewer that the current discussions in the field of the neonatal microbiome do not only focus on the question of potentially long-lasting effects of delivery mode but also on the diverse set of factors which may affect the neonatal gut microbiome. In the introduction of the revised manuscript, we now more explicitly mention one specific publication (Chu et al. 2017), which claimed that delivery mode had no effect on the microbiome structure and which thereby sparked the

aforementioned controversy regarding the impact of delivery mode on the neonatal microbiome. Although our multivariate analysis showed that delivery mode was the strongest driver in our study, we do agree that our timeframe is limited. In this sense, our manuscript will most likely serve as a model for future metagenomic studies that will need to include more mother-neonate pairs. Additionally, these future studies will also be able to evaluate the specific effects of certain conditions such as feeding regime after the timeframe of the first 5 days after delivery. We have revised the manuscript to be more careful with our phrasing and have additionally adapted the title.

1.3. The introduction now is very comprehensive. However, it is also redundant and repeated in several passages. I feel the introduction should be shortened a bit and repeated concepts minimized. Also the discussion is very verbose.

We thank the reviewer for pointing this out. We have revised the introduction accordingly.

1.4. Is multiple hypothesis testing correction applied on the identification of 1,697 differential KEGG KOs at line 226?

We thank the reviewer for his/her comment. Yes, these tests were performed with multiple testing correction by controlling false discovery rate according to Benjamin and Hochberg (1995).

1.5. Figure 3A. It is difficult to see the vertical transmission events because of the use of a color gradient (shade of orange). More distinct colors should be adopted here, and in general the figure is not very intuitive. In the legend, it is not clear what "taxon without link" means

We thank the reviewer for pointing this out. We aimed to make the figures as clear and intuitive as possible to support our findings. Therefore, we tested many different options that allowed us to include all the current information into one graph. As several different methods for shared taxa between mother and neonate and strain identification were applied, the current version was the clearest in conferring all necessary information to the reader. We agree that gradient colours might be sometimes difficult to assess, however in this case it reflects best the fact that the darker the colour is, the more evidence was found to prove that strain transfer

between mother and neonate gut occurred. Additionally, as the taxa are already highlighted by distinct colours, we wanted to refrain from another panel of distinct colours and opted for an orange colour gradient instead.

A ‘taxon without link’ in this context described a taxon that was found in the maternal samples, but not found to be shared between mother and neonate based on all of the applied levels of analyses (e.g. neither the same taxon was identified, nor was the same strain detected in both mother and neonate gut microbiomes). We have revised the legend within Figure 3A to clarify this point.

Manuscript, figure legends, lines 1681 to 1684: The level of evidence of transmission is indicated by the shading colour, with darker shading for stronger evidence. A taxon without link describes a taxon that was found in the maternal samples, but not shared between mother and neonate.

1.6. line 212-220: microbiomes from different body sites of the mother and infants are here compared using correlation. Giving the large differences in the overall structure of the microbiome across body sites and hosts, I don't think correlation is the correct statistical tool to use.

We thank the reviewer for pointing this out. In the revised manuscript, we have additionally assessed the functional potentials of the distinct microbiome samples (vaginal swab and neonatal stool samples) using a principal coordinate analysis (PCoA) representing the ordination of their respective Jensen–Shannon distances. Based on these results, a large fraction of the vaginal swab microbiomes cluster together and represent a common microbial community structure. However, as both vaginal swab and early neonatal stool samples were marked by generally low numbers of identified KOs, some of the maternal vaginal microbiomes clustered together with neonatal microbiomes, making a clear distinction impossible. Therefore, we decided that a subsequent statistical analysis using correlation was valid in this specific case. We have included the PCoA plot and the matching plots representing the Jensen-Shannon distances comparing neonatal samples to maternal gut and vaginal swab samples as Supplementary Figure 4 and discuss the figure in the main manuscript.

Supplementary information, legend of figure 4: Assessment of functional potential across sample types and between maternal and neonatal samples. **a**, Principal coordinate analysis of Jensen-Shannon divergence was generated for the functional data of all different sample types. Lines connect samples which originated from the same neonate according to the order of sampling. **b**, Jensen-Shannon divergences of the functional profiles of the neonatal gut microbiomes to those of the respective maternal gut microbiomes. **c**, Jensen-Shannon divergences of the functional profiles of the neonatal gut microbiomes to those of the respective maternal vaginal microbiomes. Samples are coloured according to birth mode and small for gestational age (SGA) status. VD, delivery; CSD, caesarean-section delivery; M, maternal faeces; MV, maternal vaginal swab; N, neonatal faeces collected at day1 / day 3 / day 5. Boxplots: centre line – median, bounds – first and third quartile, whisker $\leq 1.5 \times$ interquartile range.

Manuscript, results lines 271 to 277: To assess whether the apparent taxonomic differences between the gut microbiomes of VD and CSD neonates are reflected at the level of functional potential, we used the metagenomic sequencing data to calculate Jensen-Shannon divergences for all samples (Supplementary Fig. 4a). Overall, comparison of the functional profiles of all neonates to the gut microbial potential of their respective mothers highlighted that the neonatal gut microbiota were more divergent from the maternal vaginal microbiota than the corresponding gut microbiota (Supplementary Fig. 4a, b & c).

Reviewer 2

2.1. In this manuscript, Wampach et al collected stool samples from mothers and matched neonates. They perform shotgun sequencing and analyze the gene-level differences between the stool microbiomes of vaginally born babies vs. C-section born babies. They find that genes encoding LPS synthesis are enriched in the vaginally born babies - and that the LPS extracted from these two stool types, when tested on primary human immune cells, elicits differences in cytokine release. The LPS extracted from the Vaginally born babies appears to be more immunostimulatory. Thus, the authors conclude that C-section disrupts transfer of specific microbial strains that are highly immunostimulatory, and that this may impact “neonatal immune system priming”.

Overall, the manuscript is much improved from its prior version. A notable strength of this manuscript is the collection and evaluation of very early post-natal samples (day 0-5) and the strong effort to characterize the functional impact of LPS on the immune system. The manuscript, as it stands, is a bit lengthy (especially in the discussion section), but is well carried out and interesting. I offer the following suggestions for improvement.

We thank the reviewer for re-evaluating the manuscript and for kindly acknowledging the efforts that were put into the revision.

Major suggestions

2.2. The authors now allude to the existing controversy in evaluating the impact of birth-mode on microbial “transmission”. I propose that it might be better for the authors to steer clear of the transmission question and to simply report the results in the context of vaginally born and CS born babies having different microbiomes (which may be the consequence of confounding factors other than just birth mode - e.g. antibiotic exposure).

We thank the reviewer for his/her comment. Mention of the controversy regarding transmission according to delivery mode was in fact suggested by reviewer 1 in the previous revision phase. We agreed that this question should not go unmentioned, since our approach not only addresses differential analyses of gut microbiome structure and function according to delivery mode, but through integration of strain-level information points to differences in vertical strain transmission (see also doi: 10.1016/j.chom.2018.06.007 and doi: 10.1016/j.chom.2018.06.005). Furthermore, we want to highlight that an extensive multivariate analysis had identified the delivery mode as the main driver of the neonatal gut microbiome structure during the earliest days after birth.

2.3. The title suggests that birth mode is causal of strain-conferred gut microbiome functions and immunostimulatory potential - I think the title should be adjusted to reflect the fact that the authors identify an association, as opposed to a causal link between birth mode and these outcomes.

We thank the reviewer for the thoughtful suggestion and have changed the title in accordance with the editor's suggestion to 'Birth mode is associated with earliest strain-conferred gut microbiome functions and immunostimulatory potential'.

2.4. Are there any other notable differences between the VD and CSD babies? Or were these all elective CSD babies - which would be the best comparator.

We thank the reviewer for pointing this out. Indeed, the data is provided in the Supplementary Data 1: out of the CSD neonates for whom the high-resolution metagenomics approach was applied, only 1 was born by emergency C-section (C101), while all others were born by elective C-section.

2.5. I find the LPS aspect of the paper the most novel and exciting - but it is only briefly focused on in the manuscript. Also, is there a reason that only a subset of the total number of subjects was included in the LPS part of the study? Lastly, is there a reason the day 3 sample was selected (as opposed to the day 5 sample, which may have been higher biomass)?

We thank the reviewer for highlighting our work on the LPS extraction and stimulations assays. As highlighted in the manuscript (lines 1147 to 1150), only a subset of stool samples contained sufficient stool material (150 mg) for extracting measurable LPS concentrations. Additionally, some stool samples, although presenting the desired biomass (150 mg), did not yield any measurable LPS concentrations. The reason for this was either because the bacterial biomass in these samples was generally low or because some of these samples were just not rich enough in Gram-negative bacteria.

Furthermore, we chose to extract LPS from the neonatal stool samples of day 3 after delivery as both, the functional data (Figure 2a) and the cytokine data from plasma (Figure 4b), reflected the most significant differences according to delivery mode at this specific day of life. By selecting day 3, we were able integrate matching microbiome, LPS and plasma data. This rationale is now more clearly presented in the manuscript.

Manuscript, results, lines 502 to 517: Based on our data, the microbial composition differed most strongly in VD and CSD neonates on day 3 postpartum and, thereby, may critically

affect the developing immune system at this time¹². We isolated LPS from faecal samples from day 3 with sufficient biomass from 16 neonates (7 VD, 7 CSD, 2 CSD+SGA; Supplementary Data 12; Methods).

Minor suggestions

2.6. Would remove the phrase “After data curation” in the abstract - it is vague.

We thank the reviewer for his/her suggestion and have adapted the abstract accordingly.

2.7. line 76 - I am not exactly sure what the authors are referring to when they state “Although CSD is not associated with improved health outcomes...”.

We agree with this reviewer that this sentence was not clear and have removed the passage from the manuscript in line 139.

2.8. Line 86 - would call this 16S rRNA gene amplicon sequencing.

We agree with the reviewer and have changed this term accordingly.

2.9. Line 132 - for what proportion of samples could sufficient DNA be extracted for downstream metagenomic analysis?

While a total of 75 samples (including maternal vaginal swabs and stool and neonatal stool samples) were collected and extracted from the 16 mother-neonate pairs, a total of 65 samples yielded enough DNA to reliably proceed with metagenomic sequencing (87%). For only 2 out of these 65 samples a high proportion of the resulting data represented contaminant sequencing reads and were thus excluded from further analyses (samples MV_C100 and V1_C100). We have mentioned this in the revised manuscript, lines 973 to 975.

2.10. The majority of the analyses seem to focus on CSD +/- SGA compared to VD; I applaud the authors for including CSD -SGA controls - but am surprised that they chose not

to analyze the three groups (CSD -SGA, CSD + SGA and VD) separately, as this would be a much stronger analysis that lacked the likely confounding factors contributed by SGA.

We thank the reviewer for raising this comment. Indeed, the initial analyses were done by keeping the samples from CSD and CSD+SGA neonates as separate groups and comparing both independently to the microbiome characteristics of VD neonates. However, on each level of analysis, we observed that there were no pronounced differences between the CSD and CSD+SGA profiles, while both groups showed the same kind of differences compared to the VD microbiomes. Therefore, we report both groups together as CSD±SGA. We have clarified this aspect in the revised manuscript.

Manuscript, results lines 356 to 359: As comparisons between VD and CSD as well as VD and CSD+SGA were largely matching independent of SGA status, we combined both groups (CSD and CSD+SGA) to increase statistical power (CSD±SGA).

2.11. Line 422, 458 - calling the approach they employed “artefact-free” seems hyperbolic.

We agree with the reviewer and suggest to change ‘artefact-free’ to ‘artefact-curated’ throughout the entire manuscript.

2.12. line 490 - typo - “form” instead of “from”.

Thank you for this comment. We have corrected this typo.

2.13. Only 16 mother/neonate pairs were collected; yet, additional neonates were studied - the methods section could be more clear as to how those additional neonates were recruited to the study.

We agree with the reviewer that this aspect could be better explained in the manuscript. We have now added another sentence to the methods section.

Manuscript, methods lines 899 to 903: From the 33 neonates that were recruited into the study, the gut microbiome of 15 (Supplementary Data 1) had previously been characterised

using a combination of 16S rRNA gene amplicon sequencing and quantitative real-time PCR⁷. For a subset of neonates, the mother was sampled additionally.

2.14. I presume the authors used standard V4 primers for 16S sequencing. Please state this, if so.

We thank the reviewer for pointing this out and have specified this in the revised manuscript.

Manuscript, methods lines 960 to 963: All DNA samples (along with 8 controls) underwent standard amplicon sequencing of the V4 region of 16S rRNA genes using primers 515F-GTGCAGCMGCCGCGGTAA and 805R-GACTACHVGGGTATCTAATCC at the Center for Analytical Research and Technology–Groupe Interdisciplinaire de Génoprotéomique Appliquée (CART-GIGA; Liège, Belgium).

2.15. What is the “optimized low-quantity DNA library preparation kit” that was used?

We thank the reviewer for his/her comment. The library preparation was performed by the service provider GATC Biotech using validated standard procedures modified to optimize the method for automated library preparation. These protocols are proprietary to GATC Biotech and were not released to the authors. We referred to the fact that this sequencing was performed by GATC Biotech in the manuscript, lines 966 to 970.

2.16. Why were some of the metagenomic libraries (C105, 109, 110 and 119) prepared using another (presumably) kit?

During the revision of the manuscript, we additionally sequenced samples from 4 more mother-neonate pairs (C105, C109, C110, C119). At that time, our in-house sequencing platform was well established and experienced with low biomass samples so we chose to do the metagenomic sequencing in-house in order to have a shorter turn-around time. Although we could not use the exact same protocol or library preparation kit as the previous sequencing facility, the quality after in-house sequencing was at least as high as for the initial sequencing of the first samples of the 12 mother-neonate pairs.

2.17. The method for removing artifactual reads relies on contain assembly and alignment. What was the contig size cutoff used for this method? If the contig size cutoff is >read length, this might remove low abundance contaminants that are not present at a high enough abundance to assemble into contigs - thus compromising the effectiveness of this contamination removal effort.

We thank the reviewer for his/her comment. Indeed, we observed that the putative artefactual sequences were almost exclusively very small fragments that could not be assembled effectively or for which no genome reconstruction could be obtained. Therefore, as a defined contig size cutoff would have not taken into consideration the many low abundance contaminant reads, we opted to have no cut-off for any of the low biomass samples (neonatal samples and vaginal swab samples). More specifically, as the removal of putative artefactual sequences relies on the simultaneous binning of contigs from both study sample and contamination control samples, the cut-off for visualizing and considering contigs was set to >0 instead of the usual cut-off of >1000 nt (which was kept for the complex samples from maternal stool). Notably, the artefact removal was not only based on assembled contigs, but identified putative contaminant sequences were also removed from the original sample reads, which further highlights the need of properly identifying, assessing and removing putative artefactual contigs and reads from low biomass microbiome data.

Manuscript, methods lines 1002 to 1005: After removing the rRNA sequences from the contigs⁵⁶, we performed joint binning of control cell-culture contigs with each of the samples' contigs individually using VizBin²⁹ without any length cut-off.

2.18. line 854 - Please provide some additional detail in text regarding the “further purification” that was used to obtain the LPS.

We now added more detail in the text, specifying the nature of the “further purification”.

Manuscript, methods lines 1150 to 1159: To maximise yields, LPS was purified from three aliquots of 50 mg of each neonatal faecal sample using the hot phenol–water method⁷³ and further purification was performed using a modified phenol re-extraction protocol⁷⁴.

2.19. Line 856 - why was LPS from the three aliquots of stool pooled - presumably because each extraction resulted in a very small amount of LPS. I would have very much liked to see this assay done in replicate (starting with the stool), as I am not familiar with the variability of LPS extraction efficiency.

LPS extraction was much more efficient if it was done on small quantities of stool samples (e.g. 50 mg) compared to LPS extraction done on bigger quantities of stool samples (e.g. 150 mg). Therefore, three small aliquots were used and pooled afterwards to have enough LPS to work with. The variability of LPS extraction efficiency using the method described within the manuscript was quite low as we could observe that for the control samples, where we extracted LPS from pure *E. coli* cultures (each culture with an OD600 of 0.5), we could obtain similar amounts of LPS (please refer to the table below).

LPS in EU/ml (Replicate A)	LPS in EU/ml (Replicate B)	LPS in EU/ml (Replicate C)	LPS in EU/ml (Replicate D)
10,610.07	8,018.01	7,923.36	9,363.41

LPS was extracted from a pure *E.coli* overnight culture grown in LB medium. Based on OD600 reading, cultures were diluted to an OD600 of 0.5 and 1.5 ml of each culture was taken for LPS extraction. Bacterial suspensions were centrifuged at 10,600 g for 10 minutes, the supernatants were discarded and pellets were used for LPS extraction. LPS concentration was quantified using an ELISA-based endotoxin detection assay (Endolisa; # 609033, Hyglos GmbH, Germany).

2.20. What proportion of all metagenomic reads were assigned a potential function? I ask because it seems surprising that 2% of all metagenomic reads are classified as part of a 2-component system, for example. That seems rather high.

We thank the reviewer for pointing this out. 77 % of the curated metagenomic reads mapped to genes that were assigned a potential function (standard deviation 13 %) in the neonate stool samples displayed in the figure. This is now mentioned in the manuscript, lines 1019 to 1022.

2.21. Figure 3 is very complex and does not communicate a strong, focused story. I wonder if the authors could simplify the figure (by, perhaps, moving some of the panels to the supplement)?

We thank the reviewer for his/her suggestion. Although we agree that Figure 3 contains much information, removing some of the panels would not seem as intuitive to us, since the order of the panels follows the main results text, while giving concrete visual examples for both a CSD and VD case. We would therefore prefer not to move any of the information that supports the main message of the manuscript to the supplement.

2.22. It is interesting (Fig 4) that one of the CSD+SGA neonates has a high proportion of Gram neg organisms and a very low amount of TNF-alpha release in the MoDC assay. What do the authors make of this? Perhaps this was in the discussion and I missed it?

We thank the reviewer for this observation. Indeed, we were also intrigued by this case and had already previously addressed this in the supplementary information, note 6. We think that this specific case could point to the importance of the composition of the LPS rather than the overall abundance being crucial for immune system stimulation.

Reviewer 3

3.1. After reviewing this manuscript again, I am quite impressed with the amount of attention and added work that the authors have given to the reviewers' comments and suggestions. While the previous manuscript was already strong, the authors acknowledged the rightful arguments made by all reviewers and submitted a revised paper that appropriately addressed all of the major points. The authors: added samples to the study (for microbiome analysis and immune assays), controlled for confounding variables known to influence microbiome structure, confirmed purity of LPS, performed the LPS assay under alternative conditions that normalized LPS concentrations across samples, provided added evidence that samples used for metagenomics were representative of all cohort samples, and limited the discussion to their findings, without associating to future health outcomes that were not assessed in their study. Truly, the authors should be commended for their effort. The result is an outstanding contribution to the field that in my view, confirms that mode of delivery impacts the infant

microbiome, both taxonomically, functionally and in relation to the immune consequence to the infant host. I would seriously question who would continue to debate on this issue after the findings presented by Wampach et al.

We are deeply thankful for the honest evaluation of our revised manuscript and for the overall positive feedback.

3.2. One comment I would like to leave with the authors, which does not impact this paper but may be helpful in future studies of a similar nature, is in reference to the answer in their rebuttal letter (Lines 1126-1228):

“We want to stress again that the main point of the manuscript is not statistics-driven, as the presence of strains, given their high degree of specificity, is valid only on an individual, sample-to-sample basis.”

Their study argues that mode of delivery impacts infant microbiome structure and performed several analysis comparing two or more groups. All studies that compare variables of interest are subjected to statistics. Thus, all of these studies must be statistics-driven. I have learned this not as a biostatistician but after analyzing many human microbiome studies, where controlling for possibly confounding co-variates is the only valid method to link a microbiome feature with a variable of interest. There are enough studies available on the effect of antibiotics on infant microbiome, for example, to help direct future studies in terms of study design and number of samples necessary.

We thank the reviewer for this very thoughtful comment. As the domain of microbiome research and especially the analysis of shotgun metagenomic sequencing is still maturing, we see this study as a methodological model for further studies. We hope to be able to perform additional studies with larger sample sizes to determine the effects of different factors on the neonatal gut microbiome in the future. In this context, the presented data will prove helpful to perform the corresponding power analyses.